# Neural Regenerative Stochastic Differential Equation: Dropout Scheme for Neural Differential Equations

## Abstract

Neural Differential Equations (NDEs) are an excellent tool for modeling continuous-time (stochastic) dynamics, effectively handling challenges such as irregular observations, missing values, and noise. Despite their advantages, there is a lack of regularization techniques in the NDE framework, particularly those like dropout, which have been successfully implemented in other discrete neural networks, making them susceptible to overfitting. To address this research gap, we introduce Neural Regenerative Stochastic Differential Equation (NRSDE), based on alternating renewal processes, as a universally applicable regularization technique for NDEs. Our study reveals that NRSDE can effectively represent a continuous approximation of neural networks that randomly deactivate some neurons during training, similar to dropout, thereby enhancing the robustness and generalization capabilities of NDEs. Through extensive experiments, we demonstrate that NRSDE outperforms existing regularization methods for NDEs and can be applied to all existing NDE models, significantly improving their performance across various deep learning tasks, including time series classification and image classification.

## 1 Introduction

Neural Differential Equations (NDEs) have gained significant attention in recent years due to their ability to model continuous-time dynamical systems by integrating differential equations with neural networks (Chen et al., 2018; Rubanova et al., 2019). The continuous-time nature of NDEs allows for more precise modeling of temporal dynamics than discrete neural network architectures and has found applications in various domains, including physics (Greydanus et al., 2019), finance (Yang et al., 2023), and a variety of fields dealing with irregularly-sampled data and missing values (Rubanova et al., 2019; Kidger et al., 2020; Oh et al., 2024).

Despite their remarkable successes, NDEs are prone to overfitting like other deep learning models, particularly when training data is limited or the model complexity is too high (Oh et al., 2024). Therefore, regularization techniques for NDEs are crucial in improving their generalization and robustness. Recent studies have explored various regularization techniques for NDEs such as Neural Stochastic Differential Equations (Neural SDEs) by injecting noise into Neural Ordinary Differential Equations (Neural ODEs) (Tzen and Raginsky, 2019; Kong et al., 2020; Liu et al., 2020; Oganesyan et al., 2020), variants of Neural SDEs under distribution shifts (Oh et al., 2024), temporal regularization technique (Ghosh et al., 2020), and kinetic energy regularization for Neural ODE-based generative models (Finlay et al., 2020).

While these approaches have made progress in regularizing NDE models, several aspects remain largely unexplored compared to well-established regularization techniques developed for discrete neural networks. In particular, although dropout, one of the most efficient and widely used regularization techniques in deep learning, has been extensively studied for conventional neural networks, its application to NDEs remains limited. Liu et al. (2020) made an important first step in incorporating the dropout mechanism into NDEs using jump diffusion processes and SDEs with various noise types. While this approach introduces novel ideas, it encounters theoretical and practical challenges in accurately capturing dropout. One key issue is that the continuous dynamics they propose, when discretized, fail to align with the actual behavior of dropout in discrete time. Furthermore, the

dynamic nature of continuous models raises new questions around determining dropout rates and adjusting them in real-time to maintain model stability. Additionally, the need for rescaling the model's outputs when applying dropout in NDEs remains an open problem, further complicating its integration.

To address this gap, we propose *Neural Regenerative Stochastic Differential Equations* (NRSDEs), a dropout scheme tailored for NDEs. NRSDEs leverage the concept of alternating renewal processes (Ross, 1995; Cox, 1962; Pham-Gia and Turkkan, 1999) to model the dropout process as a regenerative phenomenon, where the system alternates between active and inactive (due to dropout) states randomly. Then, we redefine the dropout rate in the continuous-time setting, and establish the connection between the dropout rate and the intensity parameters of the alternating renewal process. We also study a scaling method for consistency between outputs in training and test phases, which is important for the practical implementation of NRSDEs. Lastly, we conduct comprehensive series of experiments on various benchmark datasets and NDE models to evaluate the effectiveness of NRSDEs. Our numerical results demonstrate that NRSDE outperforms existing regularization methods for NDEs and can be applied to any type of NDE model, significantly improving the generalization ability and robustness of NDEs, achieving state-of-the-art results across various tasks.

## 2 PRELIMINARIES

**Notations.** Let $(\Omega, \mathcal{F}, \mathbb{P})$ be a probability space. For a $\mathbb{R}^d$-valued vector $\mathbf{x}$, denote the $i$-th component of $\mathbf{x}$ by $x^{(i)}$ for $i = 1, \ldots, d$. For $\lambda \in \mathbb{R}^+$, $\text{Exp}(\lambda)$ represents the exponential distribution with rate parameter $\lambda$. The rate parameter is often referred to as the intensity parameter in the context of stochastic processes. For two matrices $\mathbf{A}$ and $\mathbf{B}$ of the same size, $\mathbf{A} \circ \mathbf{B}$ represents a matrix obtained by element-wise multiplication of $\mathbf{A}$ and $\mathbf{B}$.

### 2.1 PROBLEM STATEMENT

Let $\{\mathbf{z}_0(t)\}_{0 \leq t \leq T}$ be a $d_z$-dimensional continuous-time dynamical (stochastic) process. In particular, we consider $\mathbf{z}_0(t)$ as a latent process used for various tasks such as prediction, classification, and regression. Let $\mathbf{x}$ denote the $d_x$-dimensional input data, and $\zeta : \mathbb{R}^{d_x} \to \mathbb{R}^{d_z}$ is an affine function with parameter $\theta_\zeta$.

To represent the underlying process $\mathbf{z}_0(t)$, Chen et al. (2018) proposed a Neural ODE as the solution of the following ordinary differential equation

$$\frac{d\mathbf{z}_0(t)}{dt} = \gamma(t, \mathbf{z}_0(t); \theta_\gamma) \quad \text{with } \mathbf{z}_0(0) = \zeta(\mathbf{x}; \theta_\zeta),$$

where $0 \leq t \leq T$, $\gamma(\cdot; \cdot; \theta_\gamma)$ is a neural network parameterized by $\theta_\gamma$, which is inspired by the following residual connections in ResNet:

$$\mathbf{Z}_0(k + 1) = \mathbf{Z}_0(k) + \gamma(\mathbf{Z}_0(k); \theta_k), \tag{1}$$

where $\mathbf{Z}_0(k)$ represents the hidden state of ResNet at the $k$-th layer. See Sander et al. (2022) for a detailed discussion of the relationship between ResNets and Neural ODEs.

In (Tzen and Raginsky, 2019; Liu et al., 2020; Kong et al., 2020), Neural ODEs have been extended to Neural SDEs for describing continuous-time *stochastic* latent processes, which are governed by the following stochastic differential equation:

$$d\mathbf{z}_0(t) = \gamma(t, \mathbf{z}_0(t); \theta_\gamma) \, dt + \sigma(t, \mathbf{z}_0(t); \theta_\sigma) \, d\mathbf{W}(t) \quad \text{with } \mathbf{z}_0(0) = \zeta(\mathbf{x}; \theta_\zeta), \tag{2}$$

where $0 \leq t \leq T$, $\mathbf{W}(t)$ is a $d_z$-dimensional Brownian motion[1], and neural networks $\gamma(\cdot, \cdot; \theta_\gamma)$ and $\sigma(\cdot, \cdot; \theta_\sigma)$ are drift and diffusion functions parameterized by $\theta_\gamma$ and $\theta_\sigma$, respectively. Neural SDEs learn stochastic paths generated by Gaussian noise to improve the generalization and robustness of the models.

On the other hand, when it comes to preventing neural networks from overfitting, the most successful and powerful regularization technique in deep learning models is dropout, which randomly deactivates certain neurons during training (Srivastava et al., 2014). Therefore, to further improve the performance of Neural Differential Equations (NDEs), one naturally pose the following question:

---

[1]The dimension of Brownian motion can be arbitrary chosen. Here, we set to be $d_z$ in our experiments.

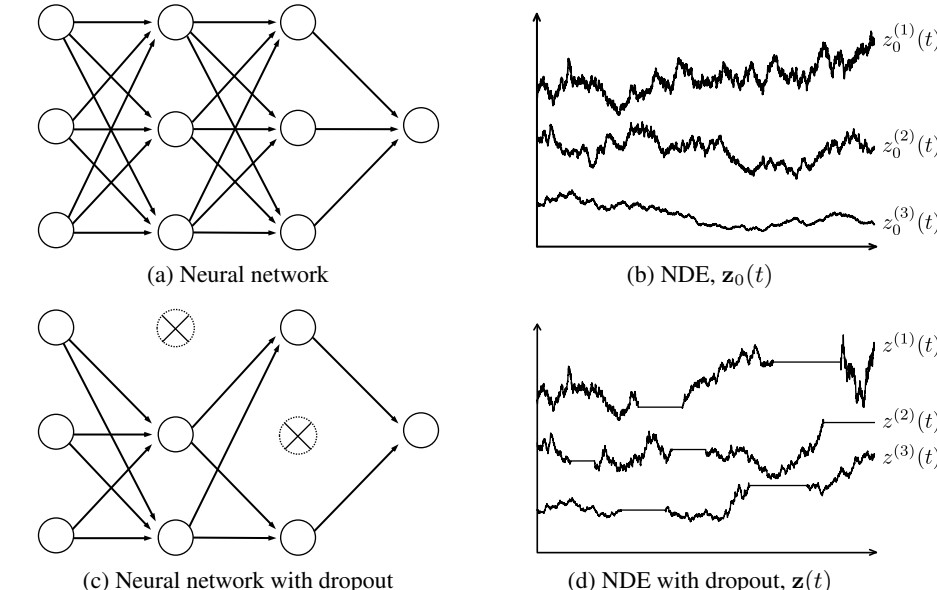

Figure 1: Illustration of dropout in discrete neural networks and continuous-time latent processes: (a) discrete neural network, (b) NDE, (c) neural network with dropout, (d) NDE with dropout.

> **Q.** How can we incorporate the mechanism of dropout into the NDE framework?

The answer to this question lies in developing a suitable continuous-time stochastic process that approximates the following ResNet with dropout:

$$\mathbf{Z}(k+1) = \mathbf{Z}(k) + \gamma(\mathbf{Z}(k); \theta_k) \circ \xi_k, \tag{3}$$

where $\mathbf{Z}(k)$ represents the hidden state of ResNet with dropout at the $k$-th layer 4and $\mathbb{P}(\xi_k^{(i)} = 0) = 1 - \mathbb{P}(\xi_k^{(i)} = 1) = p$, with $\xi_k \in \mathbb{R}^{d_z}$. Note that $\xi_k^{(i)}$ refers to the $i$-th component of $\xi_k$. Compared to Equation 1, Equation 3 introduces the Bernoulli variable $\xi_k$, which controls whether the state of the hidden layer at step $k+1$ evolves from its previous state. Specifically, when $\xi_k^{(i)} = 0$, the corresponding hidden state remains unchanged, whereas when $\xi_k^{(i)} = 1$, the hidden state is updated as in the standard ResNet.

To provide a clearer illustration, Figure 1 provides a visual comparison of a standard neural network, a neural network with dropout, and their continuous-time counterparts. First, consider a standard neural network without dropout, as shown in Figure 1a. When dropout is applied, some neurons are temporarily removed from the network, as shown in Figure 1c. Now, consider the solution $\mathbf{z}_0(t)$ of an NDE, which represents the continuous counterpart of the discrete neural network shown in Figure 1b. From this, a new continuous-time latent process $\mathbf{z}(t)$ can be constructed, where $\mathbf{z}(t)$ evolves like $\mathbf{z}_0(t)$ in its active state and temporarily pauses its evolution during its inactive state; See Figure 1d. This serves as the continuous analogue of Equation 3, effectively incorporating the dropout mechanism into the NDE framework.

Our key insight is that the behavior of dropout in neural networks in continuous time can be interpreted as a stochastic process where the system alternates between periods of active (evolution) and inactive (pause) states over random time intervals. This dynamics can be naturally modeled using an alternating renewal process, which effectively capture the random dynamics of dropout in continuous time. By leveraging this framework, we provide a more robust and theoretically sound regularization technique for NDEs. In addition, we address important follow-up questions: i) what is the proper definition of the dropout rate in continuous-time settings? ii) how can the dropout rate be controlled throughout training? and iii) how can the model's outputs be rescaled during the test phase to maintain consistency?

## 2.2 ALTERNATING RENEWAL PROCESS

Before presenting our proposed NRSDE, we briefly review the alternating renewal process, which is a key concept in our methodology. This class of regenerative stochastic processes is commonly used to model systems that alternate between two states over time. It has found wide applications in fields such as operations research, queueing theory, and reliability engineering (Stanford, 1979; Heath et al., 1998; Pham-Gia and Turkkan, 1999; Birolini, 1974). For a more detailed explanation, we refer to Appendix A.

Consider a dynamical system that alternates between two (active and inactive) states. Let $\{X_n\}_{n \geq 1}$ be the sequence of i.i.d. random variables with a distribution $G$ representing the lengths of time that the system is in active state, and let $\{Y_n\}_{n \geq 1}$ be the sequence of i.i.d. random variables with a distribution $H$ representing the lengths of time that the system is in inactive state. Assume that $X_n$ and $Y_m$ are independent for any $n \neq m$. However, $X_n$ and $Y_n$ (for the same index $n$) are not necessarily independent. Moreover, a renewal is defined as the alternation between active and inactive states where the length of each renewal, $\{X_n + Y_n\}_{n \geq 1}$, has a common distribution $F$. Then, the renewal process associated with $\{X_n, Y_n\}_{n \geq 1}$ is called an *alternating renewal process*. In particular, we focus on an exponential alternating renewal process where $G$ and $H$ are exponentially distributed.

For $t \geq 0$, let $N(t)$ denote the number of renewals in the interval $[0, t]$, i.e.,

$$N(t) = \sum_{n=1}^{\infty} \mathbf{1}_{\{X_n + Y_n \leq t\}}. \tag{4}$$

So, $\{N(t)\}_{t \geq 0}$ is the counting process. With $S_0 = 0$, define $S_n$ by for each $n \geq 1$,

$$S_n = \sum_{i=1}^{n} T_i, \tag{5}$$

where $T_{2i-1} := X_i$ and $T_{2i} := Y_i$. That is, $S_{2n}$ indicates the arrival time of the $n$-th renewal. Similarly, $S_{2n-1}$ denotes the start time of the $n$-th inactive state, which lasts for a duration of $Y_n$ until it alternates to the next active state. Moreover, at time $t$, the system is in the active state if $S_{2n-2} \leq t < S_{2n-1}$, and in the inactive state if $S_{2n-1} \leq t < S_{2n}$.

## 3 THE PROPOSED DROPOUT METHOD FOR NDES

In this section, we present the proposed dropout scheme for NDEs, Neural Regenerative Stochastic Differential Equation (NRSDE). We then redefine the concept of dropout rate to fit the continuous-time setting and introduce a method for tuning the hyperparameters of NRSDE. Finally, we discuss the scaling factor of dropout used during the test phase.

### 3.1 NEURAL REGENERATIVE STOCHASTIC DIFFERENTIAL EQUATION (NRSDE)

Recall that $\mathbf{z}_0(t) \in \mathbb{R}^{d_z}, 0 \leq t \leq T$ be the solution of a NDE. Note that we can use any type of NDEs such as Neural ODE, Neural CDE, and Neural SDE for modeling $\mathbf{z}_0(t)$. However, to provide a more general explanation of NDEs, we assume that $\mathbf{z}_0(t)$ is represented as a Neural SDE satisfying Equation 2.

Define $\mathbf{X_n} = \left( \{X_n^{(1)}\}_{n \geq 1}, \ldots, \{X_n^{(d_z)}\}_{n \geq 1} \right)$ and $\mathbf{Y_n} = \left( \{Y_n^{(1)}\}_{n \geq 1}, \ldots, \{Y_n^{(d_z)}\}_{n \geq 1} \right)$ where $\{X_n^{(i)}\}$ are i.i.d. exponential random variables with rate $\lambda_1$ and $\{Y_n^{(i)}\}$ are i.i.d. exponential random variables with rate $\lambda_2$ for $i = 1, \ldots, d_z$. Assume that $\{X_n^{(i)}\}_{n \geq 1}$ and $\{X_n^{(j)}\}_{n \geq 1}$ are mutually independent for all $i$ and $j$. Then, for $i = 1, \ldots, d_z$, $N^{(i)}(t)$ and $S_n^{(i)}$ can be defined as in Equation 4 and Equation 5. Given $\mathbf{z}_0(t)$, let $\mathbf{z}(t)$ denote its alternating renewal process associated with $\mathbf{X_n}$ and $\mathbf{Y_n}$[2]. Here, $X_n^{(i)}$ and $Y_n^{(i)}$ represent the sequences of random times that the $i$-th component of $\mathbf{z}(t)$ remains in the active and inactive states, respectively, as described in Section 2.2. We refer to this

---

[2]That is, $z^{(i)}(t)$, each component of $\mathbf{z}(t)$, is an exponential alternating renewal process of $z_0^{(i)}(t)$ associated $\{X_n^{(i)}, Y_n^{(i)}\}_{n \geq 1}$.

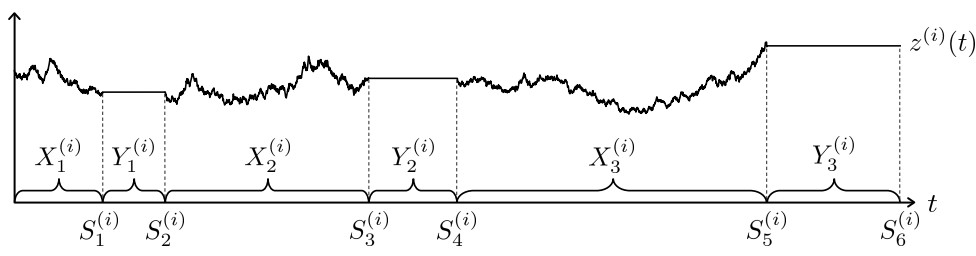

Figure 2: Illustration of $i$-th component of NRSDE $\mathbf{z}(t)$

new process $\mathbf{z}(t)$ as *Neural Regenerative Stochastic Differential Equation (NRSDE) of* $\mathbf{z}_0(t)$. More specifically, $z^{(i)}(t)$ is given by for $S_{2k}^{(i)} \leq t < S_{2k+1}^{(i)}$ and $k = 0, 1, 2, \ldots$:

$$\mathrm{d}z^{(i)}(t) = \gamma^{(i)}(t, \mathbf{z}(t); \theta_\gamma)\,\mathrm{d}t + \sum_{j=1}^{d_z} \sigma^{(i,j)}(t, \mathbf{z}(t); \theta_\sigma)\,\mathrm{d}W^{(j)}(t), \quad \text{with } z^{(i)}(t) = z^{(i)}(S_{2k}^{(i)}),$$

and $z^{(i)}(t) = z^{(i)}(S_{2k+1}^{(i)})$ for $S_{2k+1}^{(i)} \leq t < S_{2k+2}^{(i)}$ with $z^{(i)}(0) = z_0^{(i)}(0)$. Equivalently, $z^{(i)}(t)$ can be written in integral form: for $S_{2k}^{(i)} \leq t < S_{2k+1}^{(i)}$ and $k = 0, 1, 2, \ldots$,

$$z^{(i)}(t) = z^{(i)}(S_{2k}^{(i)}) + \int_{S_{2k}^{(i)}}^{t} \gamma^{(i)}(s, \mathbf{z}(s); \theta_\gamma)\,\mathrm{d}s + \int_{S_{2k}^{(i)}}^{t} \sum_{j=1}^{d_z} \sigma^{(i,j)}(s, \mathbf{z}(s); \theta_\sigma)\,\mathrm{d}W^{(j)}(s)$$

and $z^{(i)}(t) = z^{(i)}(S_{2k+1}^{(i)})$ for $S_{2k+1}^{(i)} \leq t < S_{2k+2}^{(i)}$ with $z^{(i)}(0) = z_0^{(i)}(0)$.

The construction of $\mathbf{z}(t)$ is intuitively explained as follows. For each $i = 1, \ldots, d_z$, when $S_{2k}^{(i)} \leq t < S_{2k+1}^{(i)}$, i.e., $z^{(i)}(t)$ is in the active state, $z^{(i)}(t)$ and the original latent process $z_0^{(i)}(t)$ share the same stochastic differential equation. On the other hand, during $S_{2k+1}^{(i)} \leq t < S_{2k+2}^{(i)}$, i.e., the inactive state, the evolution of $z^{(i)}(t)$ is temporarily deactivated due to dropout and remains at the fixed value $z^{(i)}(S_{2k+1}^{(i)})$ until the active state is renewed. See Figure 2 for an illustrative example of an NRSDE trajectory $\mathbf{z}(t)$.

## 3.2 DROPOUT RATE

In discrete neural networks, the dropout technique has a hyperparameter called the dropout rate $p$. Specifically, if the dropout rate is $p$, then each neuron has a probability of being deactivated for each training iteration. However, in the continuous setting of NDEs, the traditional concept of the dropout rate cannot be directly applicable because each neuron in NDEs represents the value of a continuous-time process at a specific time rather than a countable entity. Therefore, the concept of the dropout rate needs to be redefined to fit the continuous version.

Among possible candidates[3], we adopt the concept of instantaneous availability at the terminal time $T$, which represents the probability that the renewal process under consideration is in the active state at $T$, denoted by $A(T) := \mathbb{P}(\{\mathbf{z}(T) \text{ is in the active state}\})$. This approach is not only computationally efficient but aligns well with the traditional meaning of the dropout rate in discrete neural networks. Hence, in NRSDEs, the dropout rate $p$ is defined as $p = 1 - A(T)$, the probability that the NRSDE is in the inactive state at $T$.

NRSDEs have two intensity parameters $\lambda_1$ and $\lambda_2$, which determine the proportion of time spent in the active (or inactive) state over the total length $T$. The following theorem provides a formula for the dropout rate $p$ in terms of two parameters of NRSDE, $\lambda_1$ and $\lambda_2$.

---

[3]One may consider representing the dropout rate in continuous time as the proportion of the average inactive time to the total time. However, for the sake of analytical tractability, we choose to use instantaneous availability.

**Theorem 3.1.** *Define* $\mathbf{X_n} = \left( \{X_n^{(1)}\}_{n \geq 1}, \ldots, \{X_n^{(d_z)}\}_{n \geq 1} \right)$ *with* $\{X_n^{(i)}\} \stackrel{i.i.d.}{\sim} Exp(\lambda_1)$ *and* $\mathbf{Y_n} = \left( \{Y_n^{(1)}\}_{n \geq 1}, \ldots, \{Y_n^{(d_z)}\}_{n \geq 1} \right)$ *with* $\{Y_n^{(i)}\} \stackrel{i.i.d.}{\sim} Exp(\lambda_2)$. *Let* $\{\mathbf{z}_0(t)\}_{0 \leq t \leq T}$ *be the original latent process and* $\mathbf{z}(t)$ *be its NRSDE associated with* $\mathbf{X_n}$ *and* $\mathbf{Y_n}$. *Then, the dropout rate* $p \in (0,1)$ *is determined by*

$$p = \frac{\lambda_1}{\lambda_1 + \lambda_2} \left( 1 - e^{-(\lambda_1 + \lambda_2)T} \right). \tag{6}$$

The proof for Theorem 3.1 can be found in Appendix B. Given the desired level of dropout $p$, there are infinitely many possible pairs $(\lambda_1, \lambda_2)$ that satisfy Equation 6. To determine $(\lambda_1, \lambda_2)$, we impose an additional condition on the expected number of renewals of $\mathbf{z}(t)$ in the interval $[0, T]$, denoted as $m := \mathbb{E}[N^{(i)}(T)]$[4]. In other words, $m$ represents the average number of repetitions of the active and inactive states in the interval $[0, T]$. Consequently, the following corollary offers a principled way to tune the intensity parameters $\lambda_1$ and $\lambda_2$ in NRSDE given the dropout rate $p$ and the expected number of renewals $m$.

**Corollary 3.2.** *Let* $p$ *be the dropout rate and* $m$ *be the expected number of renewals of* $\mathbf{z}(t)$ *in the interval* $[0, T]$. *Given* $p \in (0,1)$ *and* $m > 0$, $\lambda_1$ *and* $\lambda_2$ *of NRSDE can be determined by solving the following system of nonlinear equations:*

$$\begin{cases} p = \dfrac{\lambda_1}{\lambda_1 + \lambda_2} \left( 1 - e^{-(\lambda_1 + \lambda_2)T} \right), \\ m = \dfrac{\lambda_1 \lambda_2}{\lambda_1 + \lambda_2} T - \dfrac{\lambda_1 \lambda_2}{(\lambda_1 + \lambda_2)^2} \left( 1 - e^{-(\lambda_1 + \lambda_2)T} \right). \end{cases} \tag{7}$$

*In particular, for large* $T$, $\lambda_1$ *and* $\lambda_2$ *can be approximated by*

$$\lambda_1 \approx \frac{m}{(1-p)\,T}, \quad \lambda_2 \approx \frac{m}{p\,T}.$$

The proof for Collorary 3.2 can be found in Appendix B. While the traditional dropout scheme has a single hyperparameter, the dropout rate $p$, the continuous-time setting requires two hyperparameters, $p$ and $m$, to characterize NRSDE. Nevertheless, our sensitivity analysis summarized in Appendix F.3 indicates that the performance of NRSDE is not sensitive to $m$, which reduces the time and effort required for hyperparameter tuning. Moreover, a detailed explanation of the parameters $(p, m)$ for the trajectories of NRSDEs is provided in Figure 11 of Appendix F.

### 3.3 SCALING FACTOR IN TEST PHASE

Another important procedure when applying dropout in discrete neural networks is to use the full neural network without dropout during the test phase. This creates a difference between the network's outputs during training and testing, necessitating scaling to calibrate this discrepancy. For example, during the test phase, the weights of neurons are scaled by $1 - p$ to ensure that the average output of the network is consistent between training and testing.

Following the mechanism of dropout in discrete neural networks, the dropout rate in NRSDE is set to 0 to eliminate inactivate states during the test phase. Therefore, suitable scaling is necessary to match the average outputs during training and test phases. That is, one needs to determine a scaling factor $\mathbf{c}$ such that $\mathbb{E}[\mathbf{z}(T)] = \mathbb{E}[\mathbf{z}_0(T)] + \mathbf{c}$. Then, we use the scaled output, $\mathbf{z}_0(T) + \mathbf{c}$, for various inference tasks. We explicitly note that the scaling factor is not a tunable or learnable parameter but rather a fixed value specific to each data point. Since $\mathbf{c}$ cannot be obtained analytically, we employ Monte Carlo simulation which might seem to introduce additional computational burden. However, we emphasize that the computational overhead due to Monte Carlo simulation is incurred *only* during the test phase, and therefore does not significantly increase the overall computational cost of NRSDE. For numerical experiments related to this, please refer to Section 4.2.

Moreover, the accuracy of estimating $\mathbf{c}$ depends on the number of samples used in the Monte Carlo simulation, which can affect the performance of NRSDE. The impact of sample size is investigated in Section 4.2, which reveals that 5–10 samples are sufficient to obtain stable results. Algorithm 1 summarizes the whole process of NRSDE.

---

[4]Note that $\mathbb{E}[N^{(1)}(T)] = \mathbb{E}[N^{(2)}(T)] = \cdots = \mathbb{E}[N^{(d_z)}(T)]$ since $\{X_n^{(i)}\}_{n \geq 1}$ and $\{Y_n^{(i)}\}_{n \geq 1}$ have common distributions $\text{Exp}(\lambda_1)$ and $\text{Exp}(\lambda_2)$, respectively, for all $i$.

---

**Algorithm 1** Training and Testing with NRSDE

---

**Input**: training data $\{\mathbf{x}_i, \mathbf{y}_i\}_{i=1}^{N_{\text{train}}}$, testing data $\{\mathbf{x}_i^\dagger\}_{i=1}^{N_{\text{test}}}$, time interval $[0, T]$, hyperparameters $(p, m) \in [0, 1) \times \mathbb{R}^+$, number of epochs $N_{\text{epochs}}$
**Initialize**: Network parameters $\theta = [\theta_\zeta, \theta_\gamma, \theta_\sigma, \theta_{\text{MLP}}]$

**Training Phase**
1: **for** epoch $= 1$ to $N_{\text{epochs}}$ **do**
2:    **for** each $(\mathbf{x}, \mathbf{y})$ in $\{\mathbf{x}_i, \mathbf{y}_i\}_{i=1}^{N_{\text{train}}}$ **do**
3:       $\mathbf{z}(0) = \zeta(\mathbf{x}; \theta_\zeta)$
4:       $\mathbf{z}(T) = \text{NRSDE\_Solver}(\gamma_\theta, \sigma_\theta, \mathbf{z}(0), [0, T], (p, m))^5$          ▷ See Appendix D.1
5:       $\mathbf{y}_{\text{pred}} = \text{MLP}(\mathbf{z}(T); \theta_{\text{MLP}})$
6:       Compute loss $\mathcal{L}(\mathbf{y}_{\text{pred}}, \mathbf{y})$
7:    **end for**
8:    Compute gradients $\nabla_\theta \mathcal{L}$ and update parameters $\theta$ using optimizer
9: **end for**

**Test Phase**
1: **for** each $\mathbf{x}^\dagger$ in $\{\mathbf{x}_i^\dagger\}_{i=1}^{N_{\text{test}}}$ **do**
2:    $\mathbf{z}(0) = \zeta(\mathbf{x}^\dagger; \theta_\zeta)$
3:    $\mathbf{z}_0(T) = \text{SDE\_Solver}(\gamma_\theta, \sigma_\theta, \mathbf{z}(0), [0, T])$
4:    $\mathbf{z}(T) = \text{NRSDE\_Solver}(\gamma_\theta, \sigma_\theta, \mathbf{z}(0), [0, T], p, m)$
5:    Estimate $\mathbf{c} = \mathbb{E}[\mathbf{z}(T)] - \mathbb{E}[\mathbf{z}_0(T)]$ using Monte Carlo simulation
6:    $\mathbf{y}_{\text{pred}} = \text{MLP}(\mathbf{z}_0(T) + \mathbf{c}; \theta_{\text{MLP}})$
7: **end for**
8: **return** predicted outputs $\{\mathbf{y}_{\text{pred}}\}$

---

### 3.4 WHY ARE ALTERNATING RENEWAL PROCESSES MORE ADEQUATE THAN JUMP DIFFUSION PROCESSES FOR MODELING DROPOUT?

Liu et al. (2020) made an interesting attempt to represent the stochastic nature of dropout using random jumps through a jump diffusion process. However, as we discuss in Appendix G.1, this approach encounters some limitations in accurately capturing the dropout mechanism. Specifically, the discretization form of the jump diffusion process they proposed differs from Equation 3, leading to a mismatch in describing the true dynamics of dropout. In contrast, the discretization of NRSDE aligns exactly with Equation 3, providing a more precise representation of the dropout mechanism.

Beyond theoretical alignment, alternating renewal processes offer significant practical advantages. First, as shown in our numerical experiments in Section 4, jump diffusion dropout only improves performance at very low dropout rates $p$, typically less than 0.1 (as discussed in Appendix E.1). This behavior deviates from traditional dropout and often requires additional time-consuming tuning. On the other hand, NRSDE provides robust performance improvements across a wide range of typical dropout rates. Second, NRSDE consistently outperforms jump diffusion dropout in various settings. Lastly, NRSDE is universally applicable to any variant of NDEs, unlike jump diffusion dropout, as detailed in Appendix G.2.

## 4 EXPERIMENTS

We perform numerical experiments using real-world datasets to evaluate the effectiveness of the proposed dropout method (NRSDE) in NDEs. First, we compare the proposed method with several regularization methods in Section 4.1. Then, we assess the performance of the proposed method across various NDEs in Section 4.2. We utilize time-series datasets (SmoothSubspace (Huang et al., 2016), ArticularyWordRecognition (Wang et al., 2013), ERing (Wilhelm et al., 2015), RacketSports, Speech Commands (Warden, 2018), and PhysioNet Sepsis (Reyna et al., 2019)) and image datasets (CIFAR-100, CIFAR-10 (Krizhevsky and Hinton, 2009), STL-10 (Coates et al., 2011), and SVHN (Netzer et al., 2011)). For NRSDEs, we search for optimal hyperparameters among the dropout rate $p \in [0.1, 0.2, 0.3, 0.4, 0.5]$ and the expected number of renewals $m \in [5, 10, 50, 100]$. Further details of experimental settings and results are summarized in Appendix D and Appendix E, respectively. The source code can be accessed at `https://bit.ly/4gMUDQk`.

---

$^5$For brevity, we denote $\gamma(\cdot, \cdot; \theta_\gamma)$ and $\sigma(\cdot, \cdot; \theta_\sigma)$, respectively, simply as $\gamma_\theta$ and $\sigma_\theta$.

### 4.1 Superior Performance Over Existing Regularization Methods

We compare the proposed dropout method (NRSDE) with existing regularization methods for NDEs, using Neural ODE as the baseline. Specifically, we consider **Dropout for Drift Network** and **Dropout for MLP Classifier**, where conventional dropout (Srivastava et al., 2014) is applied to the drift network and the MLP classifier, respectively. Additionally, we also consider **Dropout of Liu et al. (2020)**, and **STEER** (Ghosh et al., 2020). We compute the performance of both with and without dropout noise during the test phase for Dropout of Liu et al. (2020), following their experimental setup. TTN refers to the use of dropout noise not only during training but also during the test phase.

Table 1 shows the classification accuracy of various regularization methods on four time series datasets. We have highlighted **the best** methods in the results table. NRSDE demonstrates superior performance compared to existing regularization methods for NDEs.

Table 1: Accuracy of various regularization methods on time series classification

| Regularization Methods | SmoothSubspace | ArticularyWordRecognition | ERing | RacketSports |
|---|---|---|---|---|
| Baseline (Neural ODE) | 0.569 (0.040) | 0.859 (0.005) | 0.839 (0.018) | 0.565 (0.065) |
| Dropout for Drift Network | 0.594 (0.016) | 0.862 (0.014) | 0.844 (0.031) | 0.598 (0.045) |
| Dropout for MLP Classifier | 0.600 (0.067) | 0.860 (0.006) | 0.856 (0.078) | 0.554 (0.076) |
| Dropout of Liu et al. (2020) | 0.617 (0.043) | 0.871 (0.054) | 0.861 (0.064) | 0.609 (0.031) |
| Dropout of Liu et al. (2020)+TTN | 0.606 (0.018) | 0.876 (0.025) | 0.878 (0.025) | 0.598 (0.076) |
| STEER (Ghosh et al., 2020) | 0.578 (0.056) | 0.871 (0.036) | 0.850 (0.029) | 0.592 (0.054) |
| NRSDE (ours) | **0.639 (0.018)** | **0.882 (0.040)** | **0.884 (0.025)** | **0.625 (0.044)** |

Table 2 shows the classification performance of various regularization methods on four image datasets. We follow the dataset selection used in previous studies on regularization methods. The performance of each method is evaluated in terms of top-5 accuracy for CIFAR-100 and top-1 accuracy for CIFAR-10, STL-10, and SVHN. NRSDE demonstrates improvement in performance over other regularization methods.

Table 2: Performance of various regularization methods on image classification

| Regularization Methods | CIFAR-100 | CIFAR-10 | STL-10 | SVHN |
|---|---|---|---|---|
| Baseline (Neural ODE) | 74.475 (1.181) | 73.870 (0.820) | 70.650 (0.688) | 91.348 (0.440) |
| Dropout for Drift Network | 75.850 (0.367) | 74.865 (1.710) | 70.787 (0.197) | 91.671 (0.353) |
| Dropout for MLP Classifier | 75.005 (0.575) | 75.095 (0.685) | 70.723 (0.376) | 92.327 (0.098) |
| Dropout of Liu et al. (2020) | 76.083 (0.502) | 74.987 (0.350) | 71.097 (0.242) | 91.568 (0.413) |
| Dropout of Liu et al. (2020)+TTN | 76.013 (0.276) | 75.015 (0.503) | 70.931 (0.286) | 91.730 (0.518) |
| STEER (Ghosh et al., 2020) | 75.567 (0.559) | 74.668 (0.821) | 70.994 (0.259) | 91.501 (0.328) |
| NRSDE (ours) | **76.470 (0.480)** | **76.877 (0.615)** | **71.833 (0.334)** | **92.381 (0.083)** |

### 4.2 Universal Applicability in NDEs

We investigate whether the proposed dropout method (NRSDE) can demonstrate consistent improvements across various NDEs. To evaluate this, we conduct experiments on both time series and image classification tasks. Notably, the time series classification tasks considered here involve challenges such as irregular observations, label imbalance, and missing values where naive Neural ODE and Neural SDE models tend to struggle (Oh et al., 2024).

For time series classification tasks, we consider the following NDE models: Neural ODEs (**GRU-ODE** (De Brouwer et al., 2019), and **ODE-RNN** (Rubanova et al., 2019)), Neural CDEs (**Neural CDE** (Kidger et al., 2020) and **ANCDE** (Jhin et al., 2023)), Neural SDEs (**Neural LSDE**, **Neural LNSDE**, and **Neural GSDE** (Oh et al., 2024)). Note that Neural LSDE, Neural LNSDE, and Neural GSDE are state-of-the-art models for the time series classification tasks under consideration. Furthermore, we apply the proposed dropout method to these NDE models to create their corresponding NRSDEs. This allows us to investigate whether the proposed dropout method contributes to performance improvement for NDEs.

Table 3 shows classification accuracy of Neural CDE, ANCDE, Neural LSDE, Neural LSDE, Neural LNSDE, Neural GSDE, and their NRSDEs on Speech Commands. GRU-ODE and ODE-RNN are excluded due to their failure in training (Kidger et al., 2020). For PhysioNet Sepsis, Table 4 shows the AUROC of various NDE models and their NRSDEs with and without Observation Intensity (OI). Moreover, we mark "∗" and "∗∗" to represent the statistical significance of the differences between NDE models with and without dropout using two-sample $t$-tests, where $p$-values are less than 0.05 and 0.01, respectively. As shown in Table 3 and Table 4, the proposed dropout scheme improves the performance of all NDE models, and most of these improvements are statistically significant. In particular, we emphasize that the performance of the current state-of-the-art models—Neural LSDE, Neural LNSDE, and Neural GSDE— can be further improved with proposed dropout method.

Table 4: AUROC on PhysioNet Sepsis

| Models | Dropout | Test AUROC | |
|--------|---------|------------|--|
| | | OI | No OI |
| GRU-ODE | X | 0.852 (0.010) | 0.771 (0.024) |
| | O | **0.877 (0.004)**∗∗ | **0.805 (0.009)**∗ |
| ODE-RNN | X | 0.874 (0.016) | 0.833 (0.020) |
| | O | **0.895 (0.003)**∗ | **0.842 (0.002)** |
| Neural CDE | X | 0.909 (0.006) | 0.841 (0.007) |
| | O | **0.912 (0.005)** | **0.860 (0.001)**∗∗ |
| ANCDE | X | 0.900 (0.002) | 0.823 (0.003) |
| | O | **0.908 (0.004)**∗∗ | **0.843 (0.007)**∗∗ |
| Neural LSDE | X | 0.909 (0.004) | 0.879 (0.008) |
| | O | **0.928 (0.003)**∗∗ | **0.894 (0.005)**∗∗ |
| Neural LNSDE | X | 0.911 (0.002) | 0.881 (0.002) |
| | O | **0.930 (0.001)**∗∗ | **0.890 (0.005)**∗∗ |
| Neural GSDE | X | 0.909 (0.001) | 0.884 (0.002) |
| | O | **0.928 (0.002)**∗∗ | **0.890 (0.002)**∗∗ |

Table 3: Accuracy on Speech Commands

| Models | Dropout | Test Accuracy |
|--------|---------|---------------|
| Neural CDE | X | 0.910 (0.005) |
| | O | **0.945 (0.001)**∗∗ |
| ANCDE | X | 0.760 (0.003) |
| | O | **0.793 (0.007)**∗∗ |
| Neural LSDE | X | 0.927 (0.004) |
| | O | **0.933 (0.001)**∗ |
| Neural LNSDE | X | 0.923 (0.001) |
| | O | **0.932 (0.002)**∗∗ |
| Neural GSDE | X | 0.913 (0.001) |
| | O | **0.930 (0.002)**∗∗ |

For image classification tasks, we consider **Neural ODE** (Chen et al., 2018) and **Neural SDEs** (Tzen and Raginsky, 2019; Liu et al., 2020; Kong et al., 2020) (Neural SDE with additive noise and Neural SDE with multiplicative noise). Other variants of NDEs are not suitable for image classification since the concept of a controlled path is designed for handling time series data.

Table 5 presents the performance of Neural ODE, Neural SDE (additive), and Neural SDE (multiplicative) with and without dropout on the four image datasets. The experimental results suggest that proposed dropout method consistently enhances the performance of all NDEs across all datasets with statistical significance.

Table 5: Classification performance on four image datasets using Neural ODE and Neural SDEs

| Model | Dropout | CIFAR-100 | CIFAR-10 | STL-10 | SVHN |
|-------|---------|-----------|----------|--------|------|
| Neural ODE | X | 74.475 (0.581) | 73.870 (0.820) | 70.650 (0.688) | 91.348 (0.440) |
| | O | **76.470 (0.480)**∗∗ | **76.877 (0.615)**∗∗ | **71.833 (0.334)**∗∗ | **92.381 (0.083)**∗∗ |
| Neural SDE (additive) | X | 74.878 (0.328) | 74.457 (0.910) | 70.666 (0.354) | 91.839 (0.255) |
| | O | **76.872 (0.442)**∗∗ | **77.102 (0.225)**∗∗ | **71.650 (0.253)**∗∗ | **92.418 (0.153)**∗∗ |
| Neural SDE (multiplicative) | X | 75.317 (0.338) | 74.670 (0.875) | 70.463 (0.485) | 91.419 (0.577) |
| | O | **76.947 (0.458)**∗∗ | **76.654 (0.696)**∗∗ | **71.484 (0.591)**∗ | **92.434 (0.149)**∗∗ |

## 4.3 Sensitivity and Computational Cost Analysis for Scaling Factor c

NRSDE uses MC simulations to estimate the scaling factor **c**. To evaluate this process, we conduct experiments to perform a sensitivity analysis and analyze the associated computational cost. First, we analyze the performance of the proposed method with varying numbers of MC samples by applying it to models such as Neural CDE, ANCDE, Neural LSDE, Neural LNSDE, and Neural GSDE on the Speech Commands dataset. The experimental results show that the accuracy of the models stabilizes once the number of MC samples exceeds 5, indicating that the scaling factor **c** is accurately estimated with 5 or more samples. We confirm a similar observation on the image dataset CIFAR-100, and the

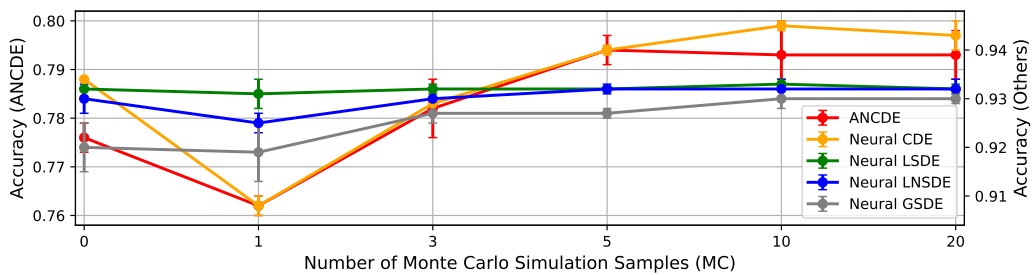

Figure 3: Performance with Different Numbers of MC Simulation Samples on Speech Commands.

detailed results are provided in the Appendix F.2. Second, Table 6 reports the computation time for cases without dropout and with **c** estimated using 5 and 10 MC samples. While MC increases the computational overhead of the proposed method, the additional computational cost remains within an acceptable range, as MC simulations are used only during the test phase, requiring inference on the trained model.

Table 6: Computation time comparison on Speech Commands (time in seconds per epoch)

| Dropout | MC | Neural CDE | ANCDE | Neural LSDE | Neural LNSDE | Neural GSDE |
|---------|-----|------------|-------|-------------|--------------|-------------|
| X | - | 25.560 (0.259) | 53.264 (0.157) | 19.416 (0.147) | 19.532 (0.109) | 19.776 (0.052) |
| O | 5 | 27.127 (0.293) | 58.943 (0.300) | 21.595 (0.163) | 21.682 (0.169) | 22.458 (0.154) |
|   | 10 | 32.172 (0.329) | 75.486 (0.250) | 31.567 (0.147) | 30.034 (0.096) | 31.234 (0.227) |

## 4.4 CONTINUOUS-TIME MODELING

While our paper focuses on modeling the continuous learning representation of neural networks with dropout, the proposed NRSDE framework can also be effectively applied to regenerative processes. To demonstrate this, we conduct an experiment using the Total Agent Call Time dataset. This dataset represents the cumulative call time of multiple call center agents recorded over time and is characterized by periods of inactivity (no calls). Based on this dataset, we design a forecasting task to estimate the total cumulative call time across all agents. As shown in Table 4, the proposed method achieves significantly lower Test Root Mean Squared Error (RMSE) compared to Neural ODE and Neural SDE. Furthermore, Neural ODE and Neural SDE methods fail to adequately capture the characteristics of regenerative processes, highlighting their limitations for such tasks.

Figure 4: Comparison of Test RMSE on Total Agent Call Time data

| Model | Test RMSE |
|-------|-----------|
| Neural ODE | 809.5 (439.3) |
| Neural SDE | 528.7 (375.0) |
| NRSDE | 203.7 (40.3) |

## 5 CONCLUSION

In this study, we introduced Neural Regenerative Stochastic Differential Equations (NRSDEs), a dropout technique tailored for NDEs to enhance the generalization and robustness of NDEs. The key idea is to construct an alternating renewal process, which alternates between active and inactive states for random time periods, to represent the continuous version of dropout neural networks. Within the proposed framework, we extended the definition of the dropout rate to a continuous-time setting and studied the connection between the newly defined dropout rate and the parameters of NRSDE. Moreover, we discussed how to scale the outputs of NRSDE during the test phase, which is crucial for the practical implementation of dropout. We validated the effectiveness compared to existing regularization methods for NDEs and the universal applicability of NRSDE through extensive experiments on real-world time-series and image classification. NRSDE significantly improves the performance of all the NDEs under consideration, particularly achieving state-of-the-art results on time-series datasets. However, NRSDE has limitations in terms of computational cost. First, unlike traditional discrete dropout, which has a single hyperparameter $p$, NRSDE involves two hyperparameters, $p$ and $m$. Moreover, as Monte Carlo simulation is used to estimate the scaling factor, it is necessary to investigate the impact of the number of samples used in Monte Carlo simulation on the performance of NRSDE across a wide range of datasets.

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

## A    OVERVIEW OF ALTERNATING RENEWAL PROCESS

The renewal process is a concept generalized from the Poisson process, where interarrival times follow not only the exponential distribution but also more general distributions. This process is commonly used for analyzing purchase cycles, replacement times, and similar phenomena and has been extensively studied under the name renewal theory. Among these, we focus on the alternating renewal process, which is a renewal process where a system alternates between two states, 'on' and 'off'. In this appendix, we provide a brief overview of renewal theory. For more details, we refer to (Ross, 1995; Cox, 1962; Pham-Gia and Turkkan, 1999).

Let $F$ denote the common distribution of the independent interarrival times $\{T_n\}_{n \geq 1}$,

$$F(t) = \mathbb{P}(T_1 \leq t), \quad t \in [0, \infty).$$

And let $\mu$ be the mean time of $T_1$, i.e.,

$$\mu = \mathbb{E}[T_1] = \int_0^\infty t \, \mathrm{d}F(t).$$

Define $S_n$ as the time of the $n$-th arrival, given by

$$S_0 = 0, \quad S_n = \sum_{i=1}^n T_i, \quad n \geq 1.$$

For $n \in \mathbb{N}$, let $F_n$ denote the distribution of $S_n$,

$$F_n(t) = \mathbb{P}(S_n \leq t), \quad t \in [0, \infty).$$

Let $N(t)$ denote the number of arrivals in time $[0, t]$,

$$N(t) = \sum_{n=1}^\infty \mathbf{1}_{\{S_n \leq t\}} = \arg \sup_{n \geq 0} \{S_n \leq t\}$$

**Definition A.1.** *The counting process $\{N(t)\}_{t \geq 0}$ is called a renewal process.*

The distribution of $N(t)$ can be obtained as

$$\begin{aligned}
\mathbb{P}(N(t) = n) &= \mathbb{P}(N(t) \geq n) - \mathbb{P}(N(t) \geq n + 1) \\
&= \mathbb{P}(S_n \leq t) - \mathbb{P}(S_{n+1} \leq t) \\
&= F_n(t) - F_{n+1}(t).
\end{aligned}$$

The renewal function $m(t)$ is defined as the expected value of $N(t)$, i.e., $m(t) = \mathbb{E}[N(t)]$, is closely related to most renewal theory and determines the properties of the renewal process.

**Theorem A.2.** *Let $m(t)$ be the renewal function of $N(t)$, defined as $m(t) = \mathbb{E}[N(t)]$. Then, $m(t)$ is given in terms of the arrival distribution function by*

$$m(t) = \sum_{n=1}^\infty F_n(t).$$

We summarize some important theorems related to the renewal function $m(t)$.

**Theorem A.3.** *(The Renewal Equation) Let the distribution $F$ has the density $f$, then*

$$m(t) = F(t) + \int_0^t f(t - \tau) m(\tau) \, \mathrm{d}\tau,$$

**Theorem A.4.** *(The Elementary Renewal Theorem) For finite $\mu$,*

$$\frac{m(t)}{t} \to \frac{1}{\mu} \quad as \ t \to \infty.$$

**Theorem A.5.** *(The Key Renewal Theorem) Suppose that the renewal process is aperiodic and* $g : [0, \infty) \to [0, \infty)$ *is directly Riemann integrable, then*

$$\int_0^t g(t - \tau) \, \mathrm{d}m(\tau) \to \frac{1}{\mu} \int_0^\infty g(\tau) \, \mathrm{d}\tau \quad as \ t \to \infty.$$

Now, we define an alternating renewal process. Consider a system that alternates two states, 'on' and 'off'. It starts with the state 'on' for a time $X_1$, and then transitions to the state 'off' for a time $Y_1$. This pattern continues with $X_2$ in the 'on' state, followed by $Y_2$ in the 'off' state, and so forth. Suppose that $(X_n, Y_n)$ for $n \geq 1$ are independent random vectors with identically distributed. Let, $\{X_n\}_{n \geq 1}$ have a common distribution $G$, $\{Y_n\}_{n \geq 1}$ have a common distribution $H$ and the cycle lengths $\{X_n + Y_n\}_{n \geq 1}$ have a common distribution $F$. Notice that $X_n$ and $Y_m$ are independent for any $n \neq m$, but $X_n$ and $Y_n$ can be dependent for $n \geq 1$. In this setting, interarrival times $\{T_n\}_{n \geq 1}$ can be expressed as:

$$T_n = \begin{cases} X_{\frac{n+1}{2}} & \text{if } n \text{ is odd,} \\ Y_{\frac{n}{2}} & \text{if } n \text{ is even.} \end{cases}$$

Thus, it can be seen that an alternating renewal process is a type of renewal process. Note that in a renewal process, events, arrivals, and renewals are used interchangeably. However, in alternating renewal processes, a renewal is defined as the alternation between an active phase and an inactive phase. To avoid confusion, we distinguish and use $S_{2n}$ to denote the time of the $n$-th renewal and the $(2n)$-th event and arrival. Furthermore, the focus is more on renewals, thus $N(t)$ represents the number of renewals in the interval $[0, t]$.

Define instantaneous availability $A(t)$ as

$$A(t) = \mathbb{P}\{\text{system is on at time } t\}.$$

Then, we have the following asymptotic formula for $A(t)$.

**Theorem A.6.** *If* $\mathbb{E}[X_n + Y_n] < \infty$ *and the alternating renewal process is aperiodic then*

$$\lim_{t \to \infty} A(t) = \frac{\mathbb{E}[X_n]}{\mathbb{E}[X_n] + \mathbb{E}[Y_n]}.$$

Theorem A.6 describes the ratio of 'on' and 'off' states in the limit for a system that follows an alternating renewal process. We called this limit the steady-state availability, which is generally more useful than instantaneous availability for analytical analysis in alternating renewal processes.

# B  PROOFS

For a function $u$, we denote the Laplace transform of $u(t)$ by

$$\mathcal{L}\{u\}(s) = \int_0^\infty e^{-st} u(t) \, \mathrm{d}t.$$

*Proof of Theorem 3.1.* From the definition of the dropout rate in continuous time, we have $p = 1 - A(T)$ for given $p \in (0, 1)$ where $A(T) := \mathbb{P}(\{z^{(i)}(T) \text{ is in the active state}\})$[6].

Recall that $\{X_n^{(i)}\}_{n \geq 1}$ have a common distribution $\text{Exp}(\lambda_1)$, $\{Y_n^{(i)}\}_{n \geq 1}$ have a common distribution $\text{Exp}(\lambda_2)$ and the cycle lengths $\{X_n^{(i)} + Y_n^{(i)}\}_{n \geq 1}$ have a common distribution $F$. Denote the density of $F$ as $f$. Then, using the convolution theorem, the Laplace transform of $f$ is given by

$$\mathcal{L}\{f\}(s) = \int_0^\infty e^{-st} \left(\lambda_1 e^{-\lambda_1 t}\right) \mathrm{d}t \int_0^\infty e^{-st} \left(\lambda_2 e^{-\lambda_2 t}\right) \mathrm{d}t$$

$$= \frac{\lambda_1}{s + \lambda_1} \frac{\lambda_2}{s + \lambda_2}. \tag{8}$$

---

[6]As each $z^{(i)}(t)$ has common distributions for $\{X_n^{(i)}\}_{n \geq 1}$ and $\{Y_n^{(i)}\}_{n \geq 1}$, we have $\mathbb{P}(\{z^{(1)}(T) \text{ is in the active state}\}) = \cdots = \mathbb{P}(\{z^{(d_z)}(T) \text{ is in the active state}\})$.

Moreover, $A(t)$ can be further expressed by conditioning on the value of $S_2^{(i)} = X_1^{(i)} + Y_1^{(i)}$

$$
\begin{aligned}
A(t) &= \mathbb{P}(\{z^{(i)}(t) \text{ is in the active state}\}) \\
&= \mathbb{P}(\{z^{(i)}(t) \text{ is in the active state}\}, S_2^{(i)} > t) + \mathbb{P}(\{z^{(i)}(t) \text{ is in the active state}\}, S_2^{(i)} \le t) \\
&= \mathbb{P}\left(S_1^{(i)} > t\right) + \int_0^t \mathbb{P}\left(\{z^{(i)}(t) \text{ is in the active state}\} \mid S_2^{(i)} = \tau\right) f(\tau)\,\mathrm{d}\tau \\
&= \mathbb{P}\left(X_1^{(i)} > t\right) + \int_0^t \mathbb{P}(\{z^{(i)}(t - \tau) \text{ is in the active state}\})f(\tau)\,\mathrm{d}\tau \\
&= e^{-\lambda_1 t} + \int_0^t A(t - \tau)f(\tau)\,\mathrm{d}\tau, \quad\quad\quad\quad (9)
\end{aligned}
$$

where we have used the renewal property of $z^{(i)}(t)$ for the fourth equality. Taking the Laplace transform on both sides of Equation 9 yields,

$$
\mathcal{L}\{A\}(s) = \frac{1}{s + \lambda_1} + \mathcal{L}\{A\}(s)\,\mathcal{L}\{f\}(s).
$$

Thus, we have using Equation 8

$$
\begin{aligned}
\mathcal{L}\{A\}(s) &= \frac{1}{(1 - \mathcal{L}\{f\}(s))(s + \lambda_1)} \\
&= \frac{s + \lambda_2}{s(s + \lambda_1 + \lambda_2)} \\
&= \frac{\lambda_2}{\lambda_1 + \lambda_2}\frac{1}{s} + \frac{\lambda_1}{\lambda_1 + \lambda_2}\frac{1}{s + \lambda_1 + \lambda_2}.
\end{aligned}
$$

Then, $A(t)$ is uniquely determined as

$$
A(t) = \frac{\lambda_2}{\lambda_1 + \lambda_2} + \frac{\lambda_1}{\lambda_1 + \lambda_2}e^{-(\lambda_1 + \lambda_2)t}.
$$

Thus, we have the desired result:

$$
\begin{aligned}
p &= 1 - A(T) \\
&= \frac{\lambda_1}{\lambda_1 + \lambda_2}\left(1 - e^{-(\lambda_1 + \lambda_2)T}\right).
\end{aligned}
$$

$\square$

*Proof of Corollary 3.2.* Let $m^{(i)}(t) = \mathbb{E}[N^{(i)}(t)]$. Then, by the Renewal equation in Theorem A.3,

$$
m^{(i)}(t) = F(t) + \int_0^t f(t - \tau)m^{(i)}(\tau)\,\mathrm{d}\tau. \quad\quad\quad\quad (10)
$$

By taking the Laplace transform on both sides of Equation 10, we obtain

$$
\mathcal{L}\{m^{(i)}\}(s) = \mathcal{L}\{F\}(s) + \mathcal{L}\{f\}(s)\,\mathcal{L}\{m^{(i)}\}(s).
$$

where $F$ and $f$ are distribution and density functions of $\{X_n^{(i)} + Y_n^{(i)}\}_{n \ge 1}$, which yields

$$
\begin{aligned}
\mathcal{L}\{m^{(i)}\}(s) &= \frac{\mathcal{L}\{F\}(s)}{1 - \mathcal{L}\{f\}(s)} \\
&= \frac{\lambda_1 \lambda_2}{s^2(s + \lambda_1 + \lambda_2)} \\
&= \frac{\lambda_1 \lambda_2}{\lambda_1 + \lambda_2}\frac{1}{s^2} - \frac{\lambda_1 \lambda_2}{(\lambda_1 + \lambda_2)^2}\left(\frac{1}{s} - \frac{1}{s + \lambda_1 + \lambda_2}\right).
\end{aligned}
$$

Therefore, $m^{(i)}(t)$ is given by

$$
m^{(i)}(t) = \frac{\lambda_1 \lambda_2}{\lambda_1 + \lambda_2}t - \frac{\lambda_1 \lambda_2}{(\lambda_1 + \lambda_2)^2}\left(1 - e^{-(\lambda_1 + \lambda_2)t}\right).
$$

From the definition of $m = \mathbb{E}[N^{(i)}(T)] = m^{(i)}(T)$ and Theorem 3.1, we have for given $p \in (0, 1)$ and $m > 0$

$$p = \frac{\lambda_1}{\lambda_1 + \lambda_2} \left( 1 - e^{-(\lambda_1 + \lambda_2)T} \right),$$

$$m = \frac{\lambda_1 \lambda_2}{\lambda_1 + \lambda_2} T - \frac{\lambda_1 \lambda_2}{(\lambda_1 + \lambda_2)^2} \left( 1 - e^{-(\lambda_1 + \lambda_2)T} \right).$$

Furthermore, from Theorem A.6, $\lim_{T \to \infty} A(T)$ can be easily computed as follows:

$$\lim_{T \to \infty} A(T) = \frac{\mathbb{E}[X_n^{(i)}]}{\mathbb{E}[X_n^{(i)}] + \mathbb{E}[Y_n^{(i)}]} = \frac{1/\lambda_1}{1/\lambda_1 + 1/\lambda_2} = \frac{\lambda_2}{\lambda_1 + \lambda_2}.$$

In other words, we have the following asymptotic relation: for sufficiently large $T$,

$$p = 1 - A(T)$$

$$\approx 1 - \frac{\lambda_2}{\lambda_1 + \lambda_2}. \tag{11}$$

Furthermore, using Theorem A.4, we get for sufficiently large $T$

$$\frac{m}{T} = \frac{m^{(i)}(T)}{T}$$

$$\approx \frac{\lambda_1 \lambda_2}{\lambda_1 + \lambda_2}. \tag{12}$$

From Equation 11 and Equation 12, $\lambda_1$ and $\lambda_2$ can be approximated by for large $T$

$$\lambda_1 \approx \frac{m}{(1-p)\,T}, \quad \lambda_2 \approx \frac{m}{p\,T}.$$

$\square$

## C  DESCRIPTION OF DATASETS

**SmoothSubspace.**   The SmoothSubspace dataset (Huang et al., 2016) is designed to test algorithms' ability to identify and classify smooth trajectories in a high-dimensional space. This dataset consists of 150 simulated univariate time series, each represented in a 15-dimensional space. The time series are generated such that they lie on or near a smooth subspace within this higher-dimensional space. Each series can belong to one of three classes, with the class indicating the particular subspace configuration that the series aligns with.

**ArticularyWordRecognition.**   The ArticularyWordRecognition dataset (Wang et al., 2013) uses an Electromagnetic Articulograph (EMA) to measure tongue and lip movements during speech. It includes data from multiple native English speakers producing 25 words. The dataset features recordings from twelve sensors capturing **x**, **y**, and **z** positions at a 200 Hz sampling rate. Sensors are positioned on the forehead, tongue (T1 to T4), lips, and jaw. Out of 36 possible dimensions, this dataset includes 9.

**ERing.**   The ERing dataset (Wilhelm et al., 2015) is collected using a prototype finger ring known as eRing, which detects hand and finger gestures through electric field sensing. The dataset includes six classes of finger postures involving the thumb, index finger, and middle finger. Each data series is four-dimensional with 65 observations, representing measurements from electrodes sensitive to the distance from the hand.

**RacketSports.**   The RacketSports dataset consists of data collected from university students playing badminton or squash while wearing a Smart Watch. The watch recorded **x**, **y**, and **z** coordinates from both the accelerometer and gyroscope, which were transmitted to a phone and stored in an Attribute-Relation File Format (ARFF) file. The dataset includes accelerometer and gyroscope measurements in the order: accelerometer **x**, **y**, **z**, followed by gyroscope **x**, **y**, **z**. The data was collected at 10 Hz over 3 seconds during either forehand/backhand strokes in squash or clear/smash strokes in badminton.

**Speech Commands.** The Speech Commands dataset (Warden, 2018) features a comprehensive set of one-second long audio clips that include spoken words and ambient sounds. This dataset contains 34,975 time-series entries, each corresponding to one of 35 different words. Ten specific words were chosen as categories: 'yes', 'no', 'up', 'down', 'left', 'right', 'on', 'off', 'stop', and 'go'. The audio clips are pre-processed by computing the Mel-frequency cepstral coefficients, which serve as features to capture the audio characteristics more effectively, thereby enhancing the accuracy of the applied machine learning models. Each audio sample is represented by a time series of 161 frames, with each frame consisting of 20 feature dimensions.

**PhysioNet Sepsis.** The 2019 PhysioNet Sepsis challenge (Reyna et al., 2019) focuses on predicting sepsis, a critical condition caused by blood infections leading to numerous fatalities. This challenge uses a dataset with 40,335 ICU patient records, featuring 34 time-dependent indicators like heart rate and body temperature. Researchers aim to determine the presence of sepsis in patients based on the sepsis-3 criteria. This dataset presents challenges due to its irregular time-series nature—only 10% of data points are timestamped for each patient. To tackle this, we employ two strategies for time-series classification: (i) using observation intensity (OI), which gauges the severity of the patient's condition, and (ii) without using OI. Each method adjusts for the dataset's imbalance by evaluating performance through the Area Under the Receiver Operating Characteristic curve (AUROC) score.

**CIFAR-100 and CIFAR-10.** These datasets, introduced by Krizhevsky and Hinton (2009), contain 60,000 32x32 pixel RGB images that are split into two groups: CIFAR-100, with 100 different fine-grained classes, and CIFAR-10, with 10 broader categories. Each dataset is evenly divided, with 50,000 images allocated for training and 10,000 for testing. The images depict a wide variety of objects, including animals, vehicles, and everyday scenes. For example, CIFAR-100 includes detailed classes such as maple trees, shrews, and pickup trucks, while CIFAR-10 contains broader categories such as airplanes, automobiles, dogs, and cats.

**STL-10.** The STL-10 dataset (Coates et al., 2011) consists of 13,000 color images, each with a resolution of 96x96 pixels, categorized into 10 classes. The images primarily capture animals, objects, and outdoor scenes. For instance, some of the categories include airplanes, birds, cars, cats, deer, dogs, horses, monkeys, ships, and trucks.

**SVHN.** The Street View House Numbers (SVHN) dataset (Netzer et al., 2011) consists of over 600,000 digit images extracted from real-world scenes, specifically house numbers captured by Google's Street View. The images typically have a resolution of around 32x32 pixels and contain digits from '0' to '9'. Each image often includes multiple digits, but the dataset focuses on identifying individual digits.

## D    DETAILS OF EXPERIMENTAL SETTINGS

We followed the recommended pipelines for each dataset and model. We conducted experiments repeated for five iterations to ensure robustness in all settings. For time series classification experiments, we adhered to experimental protocols provided in the GitHub repositories of each model. For image classification experiments, as there were no publicly available protocols for NDEs, we established our own. Details of the protocols we devised are elaborated in Section D.4. Our experiments were performed using a server on Ubuntu 20.04.6 LTS, equipped with an Intel(R) Core(TM) i9-10980XE CPU and four NVIDIA GeForce RTX 4090 GPUs. The source code can be accessed at https://bit.ly/4gMUDQk.

### D.1    DETAILED NRSDE ALGORITHM

We implemented the NRSDE algoritm based on the commonly used SDE solver, torchsde[7] (Li et al., 2020). The algorithm 2 shows the detailed NRSDE algorithm. Please refer the stochastic adjoint sensitivity method in the original paper (Li et al., 2020).

---

[7]https://github.com/google-research/torchsde

---

**Algorithm 2** Detailed NRSDE Algorithm

---

**Subroutine**: NRSDE_Solver $(\gamma_\theta, \sigma_\theta, \mathbf{z}(0), [0, T], (p, m))$

**Input**: Drift function $\gamma_\theta$, diffusion function $\sigma_\theta$, initial condition $\mathbf{z}(0)$, time interval $[0, T]$, hyperparameters $(p, m) \in [0, 1) \times \mathbb{R}^+$

1: **if** $p = 0$ **then**
2:     **return** SDE_Solver$(\gamma_\theta, \sigma_\theta, \mathbf{z}(0), [0, T])$
3: **else**
4:     Compute $\lambda_1$ and $\lambda_2$ using $(p, m, T)$ with Equation 7
5:     Generate event times $\mathbf{S} = \{\mathbf{S}_n, \mathbf{S}_n \leq T\}$ using Event_Times$(\lambda_1, \lambda_2, [0, T])$
6:     Initialize $\mathbf{z}_{\text{curr}} = \mathbf{z}(0)$
7:     **for** each time step $t_k$ in discretization of $[0, T]$ **do**
8:         Identify interval $[\mathbf{S}_i, \mathbf{S}_{i+1}]$ in $\mathbf{S}$ containing $t_k$
9:         **if** $i$ is even ($i \bmod 2 = 0$) **then**
10:            $\mathbf{z}_{\text{next}} = \text{Step}(\gamma_\theta, \sigma_\theta, [t_k, t_{k+1}], \mathbf{z}_{\text{curr}})$              $\triangleright$ Active phase
11:        **else**
12:            $\mathbf{z}_{\text{next}} = \mathbf{z}_{\text{curr}}$              $\triangleright$ Inactive phase
13:        **end if**
14:        Update $\mathbf{z}_{\text{curr}} = \mathbf{z}_{\text{next}}$
15:     **end for**
16:     **return** $\mathbf{z}_{\text{curr}}$
17: **end if**

**Subroutine**: Event_Times$(\lambda_1, \lambda_2, [0, T])$

1: Initialize $\mathbf{S} = \mathbf{0}, \mathbf{t} = \mathbf{0}, i = 1$
2: **while** $\mathbf{t} < T$ **do**
3:     Sample event time $\mathbf{t}$ by alternating renewal process using $\lambda_1$ and $\lambda_2$
4:     **if** $\mathbf{t} \leq T$ **then**
5:         Append $\mathbf{t}$ to $\mathbf{S}$
6:     **end if**
7:     Increment $i \mathrel{+}= 1$
8: **end while**
9: **return** $\mathbf{S}$

**Subroutine**: Step$(\gamma_\theta, \sigma_\theta, [t_k, t_{k+1}], \mathbf{z}_{\text{curr}})$

1: Compute $\Delta t = t_{k+1} - t_k$
2: Sample $\Delta \mathbf{W} \sim \mathcal{N}(0, \Delta t)$
3: $\mathbf{z}_{\text{next}} = \mathbf{z}_{\text{curr}} + \gamma(t_k, \mathbf{z}_{\text{curr}}; \theta_\gamma)\Delta t + \sigma(t_k, \mathbf{z}_{\text{curr}}; \theta_\sigma)\Delta \mathbf{W}$              $\triangleright$ e.g., Euler-Maruyama Method
4: **return** $\mathbf{z}_{\text{next}}$

**Subroutine**: SDE_Solver$(\gamma_\theta, \sigma_\theta, \mathbf{z}(0), [0, T])$

1: Initialize $\mathbf{z}_{\text{curr}} = \mathbf{z}(0)$
2: **for** each time step $t_k$ in discretization of $[0, T]$ **do**
3:     $\mathbf{z}_{\text{next}} = \text{Step}(\gamma_\theta, \sigma_\theta, [t_k, t_{k+1}], \mathbf{z}_{\text{curr}})$
4:     Update $\mathbf{z}_{\text{curr}} = \mathbf{z}_{\text{next}}$
5: **end for**
6: **return** $\mathbf{z}_{\text{curr}}$

**Gradient Computation** using Stochastic Adjoint Sensitivity Method

1: **Define** augmented state $\mathbf{a}(t) = [\mathbf{z}(t), \mathbf{q}(t)]$
2: Initialize adjoint variable $\mathbf{q}(T) = \nabla_{\mathbf{z}(T)}\mathcal{L}$
3: **for** time $t$ from $T$ to $0$ **do**
4:     Solve the **Stochastic Adjoint SDE** backward in time:

$$d\mathbf{q}(t) = -\left(\frac{\partial \gamma}{\partial \mathbf{z}}^\top \mathbf{q}(t)\right) dt + \left(\frac{\partial \sigma}{\partial \mathbf{z}}^\top \mathbf{q}(t)\right) d\mathbf{W}(t)$$

5:     Accumulate gradients w.r.t parameters:

$$\nabla_\theta \mathcal{L} \mathrel{+}= \left(\frac{\partial \gamma}{\partial \theta}^\top \mathbf{q}(t)\right) dt + \left(\frac{\partial \sigma}{\partial \theta}^\top \mathbf{q}(t)\right) d\mathbf{W}(t)$$

6: **end for**

---

## D.2 Various Regularization Methods

In 'Dropout for Drift Network' (Srivastava et al., 2014), we used $p = [0.1, 0.2, 0.3, 0.4, 0.5]$ for tuning the dropout rate. For 'Dropout of Liu et al. (2020)', since they did not provide a reproducible code implementation, we referred to their Github repository[8] to devise our own pipeline for the experiments. We employed tuning methods for typical dropout rates; however, since we could not observe significant performance improvements, we utilized a broader tuning grid of $p = [10^{-5}, 10^{-4}, 10^{-3}, 10^{-2}, 0.1, 0.2, 0.3, 0.4, 0.5]$. For 'STEER', we followed the pipeline outlined by Ghosh et al. (2020) and the GitHub repository[9]. For further details, please refer to the original paper.

## D.3 Time Series Classification

In 'SmoothSubspace', 'ArticularyWordRecognition', 'ERing' and 'RacketSports', we adhered to the experimental protocol using the pipeline outlined by Oh et al. (2024) and Github repository[10]. We utilized a 70%:15%:15% split for train, validation, and test due to the unconventional split ratios in the original datasets, as recommended by Oh et al. (2024). Regarding hyperparameter tuning, we employed the Python library `ray`[11] (Moritz et al., 2018; Liaw et al., 2018), as suggested by Oh et al. (2024). For further details, please refer to the original paper.

In 'Speech Commands' and 'PhysioNet Sepsis', we followed the experimental protocol using publicly available pipelines for each model. For ANCDE, we used the pipeline outlined by Jhin et al. (2023) and the Github repository[12]. For the other models, we followed the pipeline outlined by Oh et al. (2024) and the Github repository[10]. However, for the ANCDE model using the 'Speech Commands' dataset, the hyperparameter settings for the architecture were not disclosed. Therefore, we conducted experiments using the number of layers $n_l = 4$, the hidden vector dimensions $n_h = 128$, and the hidden vector dimensions for attention $n_{attention} = 20$, and we reported the performance based on these settings. We performed NRSDE hyperparameter tuning for two architecture (optimal and complex) for Neural CDE, Neural LSDE, Neural LNSDE, and Neural GSDE. For Neural CDE, contrary to previous approaches, we utilized grid search to optimize the model hyperparameters. We evaluated test performance using $n_l$ from the set $\{1, 2, 3, 4\}$ and $n_h$ from $\{16, 32, 64, 128\}$, selecting both the **optimal** architecture and the most **complex** architecture. Note that in certain models, optimal hyperparameters might be the same as complex hyperparameters. Table 9 records the performance for each set of architecture hyperparameters. For Neural LSDE, Neural LNSDE, and Neural GSDE, we used the optimal architecture hyperparameters reported by Oh et al. (2024). The following are the definitions of each model:

**GRU-ODE.** De Brouwer et al. (2019) introduced GRU-ODE, a concept of combination of Neural ODE and GRU as the solution of the following ordinary differential equation:

$$\frac{d\mathbf{z}_0(t)}{dt} = (1 - \mathbf{u}_{gate}(t)) \circ (\mathbf{u}_{vector}(t) - \mathbf{z}_0(t)).$$

Here, $\mathbf{u}_{gate}(t)$ and $\mathbf{u}_{vector}(t)$ represent the update gate and update vector of the GRU, respectively. They control how much of the new state information should be retained and how much of the previous state should be forgotten.

The reset gate $\mathbf{u}_{reset}(t)$ and update gate $\mathbf{u}_{gate}(t)$ are defined as:

$$\mathbf{u}_{reset}(t) = \text{sigmoid}(\mathbf{W}_r\mathbf{x}(t) + \mathbf{V}_r\mathbf{z}_0(t) + \mathbf{b}_r),$$
$$\mathbf{u}_{gate}(t) = \text{sigmoid}(\mathbf{W}_g\mathbf{x}(t) + \mathbf{V}_g\mathbf{z}_0(t) + \mathbf{b}_g),$$

where $\mathbf{W}_r$, $\mathbf{V}_r$, $\mathbf{W}_g$ and $\mathbf{V}_g$ are weight matrices, $\mathbf{b}_r$ and $\mathbf{b}_g$ are bias vectors, and $\mathbf{x}(t)$ is the input data at time $t$.

The update vector $\mathbf{u}_{vector}(t)$ is given by:

$$\mathbf{u}_{vector}(t) = \tanh(\mathbf{W}_v\mathbf{x}(t) + \mathbf{V}_v(\mathbf{u}_{reset}(t) \circ \mathbf{z}_0(t)) + \mathbf{b}_v),$$

where $\mathbf{W}_v$ and $\mathbf{V}_v$ are weight matrices, and $\mathbf{b}_v$ is a bias vector.

---

[8] https://github.com/xuanqing94/NeuralSDE
[9] https://github.com/arnabgho/steer
[10] https://github.com/yongkyung-oh/Stable-Neural-SDEs
[11] https://github.com/ray-project/ray
[12] https://github.com/sheoyon-jhin/ANCDE

**ODE-RNN.**  The ODE-RNN, proposed by Rubanova et al. (2019), combines Neural ODEs and RNNs. The latent process $\mathbf{z}_0(t)$ is obtained by numerically solving the ODE and then undergoing a standard RNN update process:

$$\bar{\mathbf{z}}_0(t_i) = \text{ODE\_Solver}(\gamma(\cdot; \cdot; \theta_\gamma), \mathbf{z}_0(t_{i-1}), [t_{i-1}, t_i]),$$

$$\mathbf{z}_0(t_i) = \text{RNNCell}(\bar{\mathbf{z}}_0(t_i), x^{(i)}) \quad \text{with } \mathbf{z}_0(t_0) = 0,$$

where the drift function $\gamma(\cdot; \cdot; \theta_\gamma)$ is a neural network with parameter $\theta_\gamma$.

**Neural CDE.**  Kidger et al. (2020) proposed Neural Controlled Differential Equation (Neural CDE), an extension of RNN to a continuous-time setting. It addresses the limitation of the initial condition determining the solution in the existing Neural ODE by introducing the concept of a controlled path $\mathbf{X}(t)$, which incorporates data arriving later. The model is formulated as follows:

$$\mathbf{z}_0(t) = \mathbf{z}_0(0) + \int_0^t \gamma(s, \mathbf{z}_0(s); \theta_\gamma) \, \mathrm{d}\mathbf{X}(s) \quad \text{with } \mathbf{z}_0(0) = \zeta(x^{(1)}; \theta_\zeta),$$

where the drift function $\gamma(\cdot; \cdot; \theta_\gamma)$ is a neural network with parameter $\theta_\gamma$ and the integral is the Riemann-Stieltjes integral. A controlled path $\mathbf{X}(t)$ can be any continuous function of bounded variation, but we have chosen natural cubic spline of $\mathbf{x}$ in our experiments.

**ANCDE.**  Jhin et al. (2023) proposed the Attentive Neural Controlled Differential Equation (ANCDE), where they adopt two Neural CDEs: the bottom Neural CDE for attention $\mathbf{z}_{attention}(t)$ and the top Neural CDE for latent process $\mathbf{z}_0(t)$:

$$\mathbf{z}_{attention}(t) = \mathbf{z}_{attention}(0) + \int_0^t \gamma_1(s, \mathbf{z}_{attention}(s); \theta_{\gamma_1}) \, \mathrm{d}\mathbf{X}(s),$$

$$\mathbf{z}_0(t) = \mathbf{z}_0(0) + \int_0^t \gamma_2(s, \mathbf{z}_0(s); \theta_{\gamma_2}) \, \mathrm{d}\mathbf{Y}(s),$$

where $\mathbf{Y}(t) = \text{sigmoid}(\mathbf{z}_{attention}(t)) \circ \mathbf{X}(t)$ and the drift functions $\gamma_1(\cdot; \cdot; \theta_{\gamma_1})$ and $\gamma_2(\cdot; \cdot; \theta_{\gamma_2})$ are neural networks with parameter $\theta_{\gamma_1}$ and $\theta_{\gamma_2}$, respectively.

Oh et al. (2024) proposed three classes of Neural SDEs: Neural Langevin-type SDE (LSDE), Neural Linear Noise SDE (LNSDE), and Neural Geometric SDE (GSDE). These models incorporate controlled paths into well-established SDEs, effectively capturing sequential observations like time series data and achieving recent state-of-the-art performance.

**Neural LSDE.**  Neural LSDE is a class of Langevin SDE, defined as follows:

$$\mathbf{z}_0(t) = \mathbf{z}_0(0) + \int_0^t \gamma(\mathbf{z}_0(s); \theta_\gamma) \, \mathrm{d}s + \int_0^t \sigma(s; \theta_\sigma) \, \mathrm{d}\mathbf{W}(s) \quad \text{with } \mathbf{z}_0(0) = \zeta(\mathbf{x}; \theta_\zeta),$$

where the drift function $\gamma(\cdot; \theta_\gamma)$ is a neural network with parameter $\theta_\gamma$ and the diffusion function $\sigma(\cdot; \theta_\sigma)$ is a neural network with parameter $\theta_\sigma$.

**Neural LNSDE.**  Neural LNSDE is an SDE with linear multiplicative noise, defined as follows:

$$\mathbf{z}_0(t) = \mathbf{z}_0(0) + \int_0^t \gamma(s, \mathbf{z}_0(s); \theta_\gamma) \, \mathrm{d}s + \int_0^t \sigma(s; \theta_\sigma) \mathbf{z}_0(s) \, \mathrm{d}\mathbf{W}(s) \quad \text{with } \mathbf{z}_0(0) = \zeta(\mathbf{x}; \theta_\zeta),$$

where the drift function $\gamma(\cdot; \cdot; \theta_\gamma)$ is a neural network with parameter $\theta_\gamma$ and the diffusion function $\sigma(\cdot; \theta_\sigma)$ is a neural network with parameter $\theta_\sigma$.

**Neural GSDE.**  Neural GSDE is motivated by Geometric Brownian motion (GBM) and is defined as follows:

$$\mathbf{z}_0(t) = \mathbf{z}_0(0) + \int_0^t \gamma(s, \mathbf{z}_0(s); \theta_\gamma) \mathbf{z}_0(s) \, \mathrm{d}s + \int_0^t \sigma(s; \theta_\sigma) \mathbf{z}_0(s) \, \mathrm{d}\mathbf{W}(s) \quad \text{with } \mathbf{z}_0(0) = \zeta(\mathbf{x}; \theta_\zeta),$$

where the drift function $\gamma(\cdot; \cdot; \theta_\gamma)$ is a neural network with parameter $\theta_\gamma$ and the diffusion function $\sigma(\cdot; \theta_\sigma)$ is a neural network with parameter $\theta_\sigma$.

## D.4 IMAGE CLASSIFICATION

For the dataset preprocessing, we adhered to the original train-test split provided with the datasets. From the training set, 20% was reserved for validation purposes. For data augmentation in the training set, we employed random resizing, cropping, and flipping techniques. Image normalization was carried out using the original mean and standard deviation values from each dataset, and the images were used at their default sizes without any resizing. This approach helped maintain the integrity of the original data while enhancing the model's ability to generalize from augmented variations.

In our experiments, the batch size was set to 128 for the CIFAR-100, CIFAR-10, and SVHN datasets. For the STL-10 dataset, a smaller batch size of 64 was utilized due to its limited number of samples. The models were optimized using stochastic gradient descent (SGD) with an initial learning rate of 0.1 over a course of 100 epochs. To address potential stalls in training progress, the learning rate was halved if there was no change in the loss over two consecutive epochs. Additionally, an early-stopping mechanism was implemented, terminating the training process if there was no improvement in the loss for ten consecutive epochs. This strategy helps in preventing overfitting and ensures efficient training by curtailing unnecessary computation once performance plateaus.

Specifically, the model architecture consists of the following components:

- The input layer of the image classification model is similar to conventional CNN models, where multiple convolutional operations are applied to the input image to extract basic features. An average pooling layer is used to extract the input feature as a vector $\mathbf{z}$, with a vector size of 1024 obtained through convolutional layers.

- The feature vector $\mathbf{z}$ is fed into the neural differential equation module, which estimates the hidden state $\mathbf{z}(T)$ from the initial input feature $\mathbf{z}$, where $T$ represents the depth. The last value of $\mathbf{z}(T)$ is estimated by solving neural differential equations.

- The differential equation module consists of a vector field that can be learned through backpropagation. In the case of Neural SDE, the drift term and diffusion term are represented by neural networks. The complexity of the vector field is controlled to ensure that the total number of parameters is similar to ResNet18.

- The last value $\mathbf{z}(T)$ obtained from the neural differential equation module is fed into the classifier to identify the class of the input. The parameters for all three modules (feature extractor, neural differential equation module, and classifier) are optimized simultaneously using backpropagation and the adjoint-sensitive algorithm.

We defined the complexity of the model to have a number of parameters comparable to that of ResNet18. Precisely, ResNet18 has 11.18 million parameters for 10 classes and 11.23 million for 100 classes. Our Neural ODE models possess 10.49 million and 10.59 million parameters for 10 and 100 classes, respectively. Meanwhile, the Neural SDE models, both additive and multiplicative, contain 12.59 million and 12.69 million parameters for 10 and 100 classes, respectively. We used depth $T = 10$ in the experiments. We employed the Euler method (Euler-Maruyama method in the case of SDEs) for the numerical solving of the differential equations, using a time step size of $\Delta t = 0.1$.

Neural SDE with additive and Neural SDE with multiplicative noise are defined as follows:

**Neural SDE with Additive Noise.**

$$\mathbf{z}_0(t) = \mathbf{z}_0(0) + \int_0^t \gamma(s, \mathbf{z}_0(s); \theta_\gamma) \, \mathrm{d}s + \int_0^t \sigma(s; \theta_\sigma) \, \mathrm{d}\mathbf{W}(s) \quad \text{with } \mathbf{z}_0(0) = \zeta(\mathbf{x}; \theta_\zeta),$$

where the noise term does not depend on the latent process $\mathbf{z}_0(t)$ and only depends on time $t$.

**Neural SDE with Multiplicative Noise.**

$$\mathbf{z}_0(t) = \mathbf{z}_0(0) + \int_0^t \gamma(s, \mathbf{z}_0(s); \theta_\gamma) \, \mathrm{d}s + \int_0^t \sigma(s; \theta_\sigma) \, \mathbf{z}_0(s) \, \mathrm{d}\mathbf{W}(s) \quad \text{with } \mathbf{z}_0(0) = \zeta(\mathbf{x}; \theta_\zeta),$$

where the noise term indicates an interaction between the diffusion function and the latent process $\mathbf{z}_0(t)$.

# E    DETAILED RESULTS OF BENCHMARK DATASETS

## E.1    VARIOUS REGULARIZATION METHODS

Table 7: Results of hyperparameter tuning for various regularization methods in time series datasets

### (a) Dropout for Drift Network

| $p$ | SmoothSubspace | ArticularyWordRecognition | ERing | RacketSports |
|---|---|---|---|---|
| 0 | 0.569 (0.040) | 0.859 (0.005) | 0.839 (0.018) | 0.565 (0.065) |
| 0.1 | 0.589 (0.046) | 0.862 (0.026) | 0.839 (0.048) | **0.598 (0.045)** |
| 0.2 | 0.594 (0.024) | 0.862 (0.022) | 0.844 (0.042) | 0.582 (0.028) |
| 0.3 | 0.583 (0.043) | 0.856 (0.021) | 0.844 (0.052) | 0.592 (0.056) |
| 0.4 | 0.589 (0.056) | **0.862 (0.014)** | **0.844 (0.031)** | 0.582 (0.094) |
| 0.5 | **0.594 (0.016)** | 0.851 (0.043) | 0.833 (0.019) | 0.582 (0.068) |

### (b) Dropout of Liu et al. (2020)

| $p$ | SmoothSubspace | ArticularyWordRecognition | ERing | RacketSports |
|---|---|---|---|---|
| 0 | 0.569 (0.040) | 0.859 (0.005) | 0.839 (0.018) | 0.565 (0.065) |
| $10^{-5}$ | 0.594 (0.048) | **0.871 (0.054)** | 0.844 (0.050) | 0.571 (0.018) |
| $10^{-4}$ | **0.617 (0.043)** | 0.862 (0.043) | **0.861 (0.064)** | 0.592 (0.032) |
| $10^{-3}$ | 0.606 (0.043) | 0.862 (0.014) | 0.850 (0.036) | **0.609 (0.031)** |
| $10^{-2}$ | 0.594 (0.036) | 0.856 (0.035) | 0.850 (0.055) | 0.592 (0.056) |
| 0.1 | 0.600 (0.057) | 0.859 (0.021) | 0.844 (0.057) | 0.609 (0.061) |
| 0.2 | 0.611 (0.011) | 0.853 (0.029) | 0.844 (0.065) | 0.576 (0.033) |
| 0.3 | 0.589 (0.060) | 0.862 (0.049) | 0.856 (0.040) | 0.582 (0.071) |
| 0.4 | 0.594 (0.024) | 0.856 (0.014) | 0.856 (0.056) | 0.576 (0.036) |
| 0.5 | 0.600 (0.068) | 0.862 (0.017) | 0.850 (0.033) | 0.569 (0.041) |

### (c) Dropout of Liu et al. (2020)+TTN

| $p$ | SmoothSubspace | ArticularyWordRecognition | ERing | RacketSports |
|---|---|---|---|---|
| 0 | 0.569 (0.040) | 0.859 (0.005) | 0.839 (0.018) | 0.565 (0.065) |
| $10^{-5}$ | 0.583 (0.036) | 0.874 (0.029) | 0.872 (0.010) | 0.587 (0.080) |
| $10^{-4}$ | 0.594 (0.024) | **0.876 (0.025)** | **0.878 (0.025)** | 0.576 (0.024) |
| $10^{-3}$ | 0.583 (0.024) | 0.874 (0.022) | 0.867 (0.031) | **0.598 (0.076)** |
| $10^{-2}$ | 0.594 (0.040) | 0.874 (0.031) | 0.861 (0.018) | 0.576 (0.059) |
| 0.1 | **0.606 (0.018)** | 0.871 (0.015) | 0.872 (0.033) | 0.560 (0.060) |
| 0.2 | 0.594 (0.065) | 0.862 (0.021) | 0.872 (0.036) | 0.587 (0.060) |
| 0.3 | 0.600 (0.057) | 0.868 (0.031) | 0.850 (0.024) | 0.572 (0.051) |
| 0.4 | 0.606 (0.048) | 0.871 (0.033) | 0.861 (0.033) | 0.571 (0.056) |
| 0.5 | 0.600 (0.010) | 0.862 (0.020) | 0.844 (0.016) | 0.578 (0.039) |

### (d) NRSDE (ours)

| $p$ | $m$ | SmoothSubspace | ArticularyWordRecognition | ERing | RacketSports |
|---|---|---|---|---|---|
| 0 | - | 0.569 (0.040) | 0.859 (0.005) | 0.839 (0.018) | 0.565 (0.065) |
| | 5 | 0.583 (0.029) | 0.874 (0.029) | 0.844 (0.035) | 0.576 (0.024) |
| 0.1 | 10 | 0.589 (0.011) | 0.865 (0.034) | 0.867 (0.031) | 0.576 (0.070) |
| | 50 | 0.611 (0.046) | 0.868 (0.013) | 0.867 (0.042) | 0.603 (0.071) |
| | 100 | 0.606 (0.040) | 0.871 (0.013) | 0.861 (0.043) | 0.571 (0.018) |
| | 5 | 0.594 (0.033) | 0.868 (0.026) | 0.844 (0.027) | 0.576 (0.024) |
| 0.2 | 10 | **0.639 (0.018)** | 0.859 (0.019) | 0.844 (0.050) | 0.598 (0.088) |
| | 50 | 0.600 (0.067) | 0.851 (0.031) | 0.881 (0.029) | 0.582 (0.120) |
| | 100 | 0.594 (0.055) | 0.871 (0.005) | **0.884 (0.025)** | 0.598 (0.011) |
| | 5 | 0.589 (0.056) | 0.862 (0.042) | 0.828 (0.018) | 0.592 (0.064) |
| 0.3 | 10 | 0.600 (0.016) | 0.868 (0.042) | 0.850 (0.024) | 0.592 (0.062) |
| | 50 | 0.594 (0.040) | 0.859 (0.030) | 0.844 (0.043) | 0.598 (0.085) |
| | 100 | 0.578 (0.054) | 0.876 (0.025) | 0.850 (0.010) | 0.592 (0.076) |
| | 5 | 0.583 (0.036) | **0.882 (0.040)** | 0.833 (0.040) | 0.571 (0.028) |
| 0.4 | 10 | 0.622 (0.042) | 0.868 (0.034) | 0.856 (0.025) | 0.587 (0.043) |
| | 50 | 0.594 (0.043) | 0.876 (0.026) | 0.850 (0.018) | 0.592 (0.042) |
| | 100 | 0.583 (0.033) | 0.865 (0.043) | 0.844 (0.052) | **0.625 (0.044)** |
| | 5 | 0.578 (0.031) | 0.868 (0.026) | 0.828 (0.024) | 0.576 (0.064) |
| 0.5 | 10 | 0.594 (0.024) | 0.856 (0.026) | 0.833 (0.033) | 0.587 (0.053) |
| | 50 | 0.600 (0.047) | 0.862 (0.027) | 0.839 (0.040) | 0.609 (0.034) |
| | 100 | 0.589 (0.033) | 0.868 (0.037) | 0.828 (0.043) | 0.592 (0.094) |

Table 8: Results of hyperparameter tuning for other regularization methods in image datasets

(a) Dropout for Drift Network

| $p$ | CIFAR-100 | CIFAR-10 | STL-10 | SVHN |
|---|---|---|---|---|
| 0 | 74.475 (1.181) | 73.870 (0.820) | 70.650 (0.688) | 91.348 (0.440) |
| 0.1 | **75.850 (0.367)** | 74.013 (1.431) | 70.784 (0.219) | **91.671 (0.353)** |
| 0.2 | 75.547 (0.321) | **74.865 (1.710)** | 70.672 (0.486) | 90.870 (0.319) |
| 0.3 | 73.650 (1.450) | 72.685 (2.041) | 70.628 (0.219) | 91.374 (0.294) |
| 0.4 | 75.055 (1.380) | 73.855 (2.137) | **70.787 (0.197)** | 91.587 (0.554) |
| 0.5 | 74.345 (1.074) | 73.280 (1.931) | 70.659 (0.411) | 91.467 (0.348) |

(b) Dropout of Liu et al. (2020)

| $p$ | CIFAR-100 | CIFAR-10 | STL-10 | SVHN |
|---|---|---|---|---|
| 0 | 74.475 (1.181) | 73.870 (0.820) | 70.650 (0.688) | 91.348 (0.440) |
| $10^{-5}$ | 74.852 (1.085) | **74.987 (0.350)** | **71.097 (0.242)** | 91.388 (0.348) |
| $10^{-4}$ | **76.083 (0.502)** | 74.370 (1.689) | 71.084 (0.215) | 90.906 (0.422) |
| $10^{-3}$ | 73.873 (1.973) | 74.377 (1.303) | 70.912 (0.436) | 91.520 (0.450) |
| $10^{-2}$ | 75.098 (0.530) | 74.600 (1.165) | 70.409 (0.370) | 91.194 (0.556) |
| 0.1 | 75.415 (0.415) | 73.570 (1.788) | 70.966 (0.192) | 91.272 (0.367) |
| 0.2 | 74.975 (0.394) | 72.927 (1.782) | 70.931 (0.230) | 91.539 (0.238) |
| 0.3 | 74.782 (0.608) | 72.727 (1.437) | 70.938 (0.648) | 91.478 (0.146) |
| 0.4 | 74.900 (1.401) | 72.282 (1.413) | 70.903 (0.364) | **91.568 (0.413)** |
| 0.5 | 74.535 (0.231) | 73.850 (2.184) | 70.778 (0.204) | 91.480 (0.135) |

(c) Dropout of Liu et al. (2020)+TTN

| $p$ | CIFAR-100 | CIFAR-10 | STL-10 | SVHN |
|---|---|---|---|---|
| 0 | 74.475 (1.181) | 73.870 (0.820) | 70.650 (0.688) | 91.348 (0.440) |
| $10^{-5}$ | **76.013 (0.276)** | 73.742 (1.646) | 70.747 (0.479) | **91.730 (0.518)** |
| $10^{-4}$ | 75.743 (0.307) | 74.305 (2.027) | 70.716 (0.282) | 91.366 (0.110) |
| $10^{-3}$ | 75.227 (0.261) | 74.755 (1.500) | 70.409 (0.390) | 91.323 (0.566) |
| $10^{-2}$ | 74.430 (1.637) | 74.710 (1.307) | 70.544 (0.177) | 91.088 (0.309) |
| 0.1 | 75.170 (0.213) | **75.015 (0.503)** | 70.775 (0.259) | 91.509 (0.107) |
| 0.2 | 75.692 (0.392) | 73.403 (1.190) | **70.931 (0.286)** | 91.561 (0.494) |
| 0.3 | 74.992 (1.470) | 73.375 (2.269) | 70.819 (0.455) | 91.267 (0.360) |
| 0.4 | 74.553 (1.220) | 73.915 (1.692) | 70.741 (0.206) | 91.423 (0.401) |
| 0.5 | 75.502 (1.176) | 73.160 (1.698) | 70.722 (0.411) | 91.238 (0.606) |

## E.2 TIME SERIES CLASSIFICATION

Table 9: Results of architecture hyperparameter tuning of Neural CDE for
'Speech Commands' (left) and 'PhysioNet Sepsis' (right)

| $n_l$ | $n_h$ | Test Accuracy |
|---|---|---|
| | 16 | 0.390 (0.053) |
| 1 | 32 | 0.603 (0.043) |
| | 64 | 0.782 (0.021) |
| | 128 | 0.830 (0.013) |
| | 16 | 0.753 (0.013) |
| 2 | 32 | 0.826 (0.008) |
| | 64 | 0.864 (0.003) |
| | 128 | 0.895 (0.003) |
| | 16 | 0.786 (0.011) |
| 3 | 32 | 0.853 (0.006) |
| | 64 | 0.882 (0.006) |
| | 128 | 0.906 (0.004) |
| | 16 | 0.789 (0.013) |
| 4 | 32 | 0.869 (0.004) |
| | 64 | 0.887 (0.003) |
| | 128 | **0.910 (0.005)** |

| $n_l$ | $n_h$ | Test AUROC | |
|---|---|---|---|
| | | OI | No OI |
| | 16 | 0.879 (0.007) | 0.786 (0.014) |
| 1 | 32 | 0.875 (0.009) | 0.786 (0.015) |
| | 64 | 0.879 (0.009) | 0.799 (0.012) |
| | 128 | 0.886 (0.006) | 0.808 (0.010) |
| | 16 | 0.895 (0.007) | 0.804 (0.010) |
| 2 | 32 | 0.883 (0.003) | 0.808 (0.013) |
| | 64 | 0.896 (0.008) | 0.821 (0.009) |
| | 128 | 0.895 (0.007) | 0.826 (0.022) |
| | 16 | 0.899 (0.005) | 0.813 (0.008) |
| 3 | 32 | 0.893 (0.006) | 0.834 (0.004) |
| | 64 | 0.901 (0.009) | 0.834 (0.005) |
| | 128 | **0.909 (0.006)** | 0.840 (0.011) |
| | 16 | 0.900 (0.005) | 0.822 (0.002) |
| 4 | 32 | 0.903 (0.007) | 0.836 (0.004) |
| | 64 | 0.908 (0.005) | 0.840 (0.008) |
| | 128 | 0.907 (0.004) | **0.841 (0.007)** |

Table 10: Results of hyperparameter tuning for 'Speech Commands'

| $p$ | $m$ | ANCDE |
|---|---|---|
| 0 | - | 0.760 (0.003) |
| 0.1 | 5 | 0.777 (0.002) |
| 0.1 | 10 | 0.780 (0.018) |
| 0.1 | 50 | **0.794 (0.007)** |
| 0.1 | 100 | 0.782 (0.009) |
| 0.2 | 5 | 0.776 (0.024) |
| 0.2 | 10 | 0.770 (0.028) |
| 0.2 | 50 | 0.763 (0.016) |
| 0.2 | 100 | 0.775 (0.011) |
| 0.3 | 5 | 0.753 (0.029) |
| 0.3 | 10 | 0.748 (0.033) |
| 0.3 | 50 | 0.763 (0.018) |
| 0.3 | 100 | 0.764 (0.015) |
| 0.4 | 5 | 0.672 (0.065) |
| 0.4 | 10 | 0.699 (0.056) |
| 0.4 | 50 | 0.697 (0.076) |
| 0.4 | 100 | 0.690 (0.012) |
| 0.5 | 5 | 0.572 (0.101) |
| 0.5 | 10 | 0.615 (0.088) |
| 0.5 | 50 | 0.690 (0.061) |
| 0.5 | 100 | 0.708 (0.060) |

| Models | $p$ | $m$ | Test Accuracy | | Models | $p$ | $m$ | Test Accuracy | |
|---|---|---|---|---|---|---|---|---|---|
| | | | Optimal | Complex | | | | Optimal | Complex |
| | 0 | - | 0.910 (0.005) | 0.910 (0.005) | | 0 | - | 0.924 (0.000) | 0.912 (0.004) |
| | 0.1 | 5 | 0.940 (0.002) | 0.940 (0.002) | | 0.1 | 5 | 0.913 (0.002) | 0.931 (0.002) |
| | 0.1 | 10 | 0.941 (0.001) | 0.941 (0.001) | | 0.1 | 10 | 0.914 (0.001) | 0.930 (0.001) |
| | 0.1 | 50 | 0.942 (0.000) | 0.942 (0.000) | | 0.1 | 50 | 0.912 (0.005) | 0.930 (0.002) |
| | 0.1 | 100 | 0.940 (0.003) | 0.940 (0.003) | | 0.1 | 100 | 0.915 (0.003) | 0.928 (0.001) |
| | 0.2 | 5 | 0.939 (0.001) | 0.939 (0.001) | | 0.2 | 5 | 0.895 (0.015) | 0.927 (0.002) |
| | 0.2 | 10 | 0.942 (0.001) | 0.942 (0.001) | | 0.2 | 10 | 0.908 (0.004) | 0.928 (0.003) |
| | 0.2 | 50 | 0.943 (0.002) | 0.943 (0.002) | | 0.2 | 50 | 0.905 (0.008) | 0.930 (0.002) |
| | 0.2 | 100 | **0.945 (0.001)** | **0.945 (0.001)** | | 0.2 | 100 | 0.910 (0.007) | **0.932 (0.002)** |
| Neural CDE | 0.3 | 5 | 0.937 (0.005) | 0.937 (0.005) | Neural LNSDE | 0.3 | 5 | 0.901 (0.007) | 0.904 (0.005) |
| | 0.3 | 10 | 0.938 (0.003) | 0.938 (0.003) | | 0.3 | 10 | 0.870 (0.005) | 0.916 (0.002) |
| | 0.3 | 50 | 0.940 (0.001) | 0.940 (0.001) | | 0.3 | 50 | 0.916 (0.004) | 0.929 (0.004) |
| | 0.3 | 100 | 0.941 (0.002) | 0.941 (0.002) | | 0.3 | 100 | 0.913 (0.002) | 0.926 (0.001) |
| | 0.4 | 5 | 0.932 (0.003) | 0.932 (0.003) | | 0.4 | 5 | 0.881 (0.013) | 0.890 (0.011) |
| | 0.4 | 10 | 0.932 (0.002) | 0.932 (0.002) | | 0.4 | 10 | 0.878 (0.004) | 0.910 (0.008) |
| | 0.4 | 50 | 0.935 (0.005) | 0.935 (0.005) | | 0.4 | 50 | 0.913 (0.006) | 0.920 (0.002) |
| | 0.4 | 100 | 0.932 (0.005) | 0.932 (0.005) | | 0.4 | 100 | 0.917 (0.003) | 0.928 (0.003) |
| | 0.5 | 5 | 0.927 (0.004) | 0.927 (0.004) | | 0.5 | 5 | 0.861 (0.016) | 0.867 (0.006) |
| | 0.5 | 10 | 0.920 (0.008) | 0.920 (0.008) | | 0.5 | 10 | 0.891 (0.005) | 0.893 (0.002) |
| | 0.5 | 50 | 0.926 (0.003) | 0.926 (0.003) | | 0.5 | 50 | 0.905 (0.003) | 0.915 (0.003) |
| | 0.5 | 100 | 0.920 (0.002) | 0.920 (0.002) | | 0.5 | 100 | 0.897 (0.007) | 0.928 (0.004) |
| | 0 | - | 0.927 (0.004) | 0.926 (0.005) | | 0 | - | 0.913 (0.001) | 0.899 (0.005) |
| | 0.1 | 5 | 0.930 (0.001) | 0.933 (0.002) | | 0.1 | 5 | **0.930 (0.002)** | 0.923 (0.004) |
| | 0.1 | 10 | 0.931 (0.000) | 0.930 (0.003) | | 0.1 | 10 | 0.927 (0.003) | 0.926 (0.003) |
| | 0.1 | 50 | **0.933 (0.001)** | 0.930 (0.002) | | 0.1 | 50 | 0.927 (0.001) | 0.924 (0.004) |
| | 0.1 | 100 | 0.929 (0.004) | 0.933 (0.004) | | 0.1 | 100 | 0.926 (0.001) | 0.926 (0.002) |
| | 0.2 | 5 | 0.922 (0.004) | 0.924 (0.002) | | 0.2 | 5 | 0.922 (0.001) | 0.905 (0.012) |
| | 0.2 | 10 | 0.928 (0.004) | 0.930 (0.001) | | 0.2 | 10 | 0.918 (0.005) | 0.921 (0.005) |
| | 0.2 | 50 | 0.933 (0.003) | 0.928 (0.002) | | 0.2 | 50 | 0.927 (0.003) | 0.916 (0.002) |
| | 0.2 | 100 | 0.933 (0.002) | 0.933 (0.004) | | 0.2 | 100 | 0.929 (0.003) | 0.922 (0.001) |
| Neural LSDE | 0.3 | 5 | 0.917 (0.010) | 0.912 (0.006) | Neural GSDE | 0.3 | 5 | 0.903 (0.008) | 0.888 (0.007) |
| | 0.3 | 10 | 0.909 (0.009) | 0.911 (0.009) | | 0.3 | 10 | 0.911 (0.003) | 0.914 (0.001) |
| | 0.3 | 50 | 0.931 (0.003) | 0.929 (0.003) | | 0.3 | 50 | 0.923 (0.002) | 0.921 (0.007) |
| | 0.3 | 100 | 0.932 (0.002) | 0.932 (0.002) | | 0.3 | 100 | 0.924 (0.001) | 0.926 (0.002) |
| | 0.4 | 5 | 0.896 (0.013) | 0.888 (0.008) | | 0.4 | 5 | 0.890 (0.004) | 0.894 (0.015) |
| | 0.4 | 10 | 0.913 (0.003) | 0.904 (0.003) | | 0.4 | 10 | 0.903 (0.005) | 0.881 (0.010) |
| | 0.4 | 50 | 0.929 (0.002) | 0.927 (0.005) | | 0.4 | 50 | 0.922 (0.002) | 0.915 (0.006) |
| | 0.4 | 100 | 0.929 (0.001) | 0.926 (0.002) | | 0.4 | 100 | 0.924 (0.003) | 0.912 (0.004) |
| | 0.5 | 5 | 0.864 (0.015) | 0.871 (0.016) | | 0.5 | 5 | 0.869 (0.018) | 0.853 (0.021) |
| | 0.5 | 10 | 0.889 (0.004) | 0.879 (0.017) | | 0.5 | 10 | 0.891 (0.008) | 0.862 (0.018) |
| | 0.5 | 50 | 0.925 (0.002) | 0.921 (0.007) | | 0.5 | 50 | 0.911 (0.003) | 0.903 (0.010) |
| | 0.5 | 100 | 0.924 (0.002) | 0.924 (0.004) | | 0.5 | 100 | 0.920 (0.002) | 0.915 (0.003) |

Table 11: Results of hyperparameter tuning for 'PhysioNet Sepsis' with observation intensity

| $p$ | $m$ | GRU-ODE | ODE-RNN | ANCDE |
|---|---|---|---|---|
| 0 | - | 0.852 (0.010) | 0.874 (0.016) | 0.900 (0.002) |
| 0.1 | 5 | 0.873 (0.007) | 0.891 (0.009) | 0.905 (0.004) |
| | 10 | 0.854 (0.019) | 0.883 (0.013) | 0.907 (0.005) |
| | 50 | 0.870 (0.005) | 0.889 (0.011) | **0.908 (0.004)** |
| | 100 | 0.827 (0.035) | 0.881 (0.006) | 0.904 (0.003) |
| 0.2 | 5 | 0.846 (0.030) | 0.880 (0.008) | 0.897 (0.002) |
| | 10 | 0.870 (0.006) | 0.893 (0.009) | 0.905 (0.004) |
| | 50 | 0.866 (0.006) | 0.890 (0.004) | 0.902 (0.003) |
| | 100 | **0.877 (0.004)** | 0.889 (0.010) | 0.901 (0.008) |
| 0.3 | 5 | 0.858 (0.006) | 0.872 (0.009) | 0.902 (0.006) |
| | 10 | 0.858 (0.014) | 0.894 (0.002) | 0.900 (0.007) |
| | 50 | 0.854 (0.021) | 0.891 (0.004) | 0.904 (0.007) |
| | 100 | 0.871 (0.004) | 0.887 (0.002) | 0.905 (0.008) |
| 0.4 | 5 | 0.828 (0.015) | 0.885 (0.008) | 0.895 (0.003) |
| | 10 | 0.841 (0.006) | 0.890 (0.003) | 0.899 (0.003) |
| | 50 | 0.855 (0.016) | 0.890 (0.004) | 0.900 (0.006) |
| | 100 | 0.863 (0.015) | 0.894 (0.005) | 0.904 (0.004) |
| 0.5 | 5 | 0.835 (0.007) | 0.889 (0.004) | 0.881 (0.001) |
| | 10 | 0.843 (0.010) | 0.893 (0.010) | 0.889 (0.004) |
| | 50 | 0.847 (0.010) | 0.889 (0.006) | 0.898 (0.004) |
| | 100 | 0.866 (0.008) | **0.895 (0.003)** | 0.902 (0.005) |

| Models | $p$ | $m$ | Test AUROC Optimal | Complex | Models | $p$ | $m$ | Test AUROC Optimal | Complex |
|---|---|---|---|---|---|---|---|---|---|
| | 0 | - | 0.909 (0.006) | 0.907 (0.004) | | 0 | - | 0.911 (0.002) | 0.900 (0.004) |
| | 0.1 | 5 | 0.911 (0.002) | 0.910 (0.001) | | 0.1 | 5 | 0.908 (0.009) | 0.924 (0.005) |
| | | 10 | 0.910 (0.005) | 0.903 (0.004) | | | 10 | 0.914 (0.005) | 0.922 (0.003) |
| | | 50 | 0.908 (0.004) | 0.909 (0.003) | | | 50 | 0.923 (0.008) | 0.925 (0.003) |
| | | 100 | 0.906 (0.001) | 0.906 (0.004) | | | 100 | 0.921 (0.003) | 0.923 (0.004) |
| | 0.2 | 5 | 0.901 (0.005) | 0.907 (0.004) | | 0.2 | 5 | 0.917 (0.011) | 0.914 (0.009) |
| | | 10 | 0.907 (0.003) | 0.909 (0.001) | | | 10 | 0.919 (0.007) | 0.925 (0.004) |
| | | 50 | 0.910 (0.002) | 0.908 (0.004) | | | 50 | 0.915 (0.002) | 0.924 (0.003) |
| | | 100 | 0.906 (0.003) | 0.907 (0.004) | | | 100 | 0.909 (0.006) | **0.930 (0.001)** |
| Neural CDE | 0.3 | 5 | 0.893 (0.004) | 0.907 (0.002) | Neural LNSDE | 0.3 | 5 | 0.911 (0.007) | 0.913 (0.005) |
| | | 10 | 0.901 (0.002) | 0.899 (0.004) | | | 10 | 0.908 (0.003) | 0.915 (0.003) |
| | | 50 | 0.904 (0.003) | **0.912 (0.005)** | | | 50 | 0.920 (0.007) | 0.923 (0.003) |
| | | 100 | 0.907 (0.003) | 0.909 (0.003) | | | 100 | 0.914 (0.007) | 0.922 (0.001) |
| | 0.4 | 5 | 0.886 (0.007) | 0.901 (0.004) | | 0.4 | 5 | 0.901 (0.004) | 0.903 (0.004) |
| | | 10 | 0.895 (0.008) | 0.904 (0.003) | | | 10 | 0.914 (0.004) | 0.903 (0.004) |
| | | 50 | 0.902 (0.007) | 0.910 (0.006) | | | 50 | 0.911 (0.007) | 0.916 (0.003) |
| | | 100 | 0.904 (0.006) | 0.906 (0.001) | | | 100 | 0.909 (0.004) | 0.920 (0.006) |
| | 0.5 | 5 | 0.890 (0.007) | 0.886 (0.006) | | 0.5 | 5 | 0.890 (0.007) | 0.898 (0.004) |
| | | 10 | 0.900 (0.001) | 0.895 (0.007) | | | 10 | 0.897 (0.008) | 0.901 (0.007) |
| | | 50 | 0.895 (0.005) | 0.903 (0.006) | | | 50 | 0.909 (0.002) | 0.916 (0.003) |
| | | 100 | 0.903 (0.003) | 0.902 (0.002) | | | 100 | 0.909 (0.011) | 0.914 (0.003) |
| | 0 | - | 0.909 (0.004) | 0.900 (0.002) | | 0 | - | 0.909 (0.001) | 0.905 (0.001) |
| | 0.1 | 5 | 0.923 (0.002) | **0.928 (0.003)** | | 0.1 | 5 | 0.919 (0.002) | **0.928 (0.002)** |
| | | 10 | 0.914 (0.006) | 0.927 (0.002) | | | 10 | 0.923 (0.002) | 0.927 (0.002) |
| | | 50 | 0.919 (0.001) | 0.924 (0.004) | | | 50 | 0.917 (0.005) | 0.926 (0.004) |
| | | 100 | 0.921 (0.007) | 0.926 (0.003) | | | 100 | 0.923 (0.004) | 0.923 (0.007) |
| | 0.2 | 5 | 0.919 (0.006) | 0.921 (0.004) | | 0.2 | 5 | 0.914 (0.005) | 0.907 (0.008) |
| | | 10 | 0.917 (0.005) | 0.924 (0.002) | | | 10 | 0.919 (0.002) | 0.918 (0.003) |
| | | 50 | 0.917 (0.010) | 0.923 (0.002) | | | 50 | 0.919 (0.004) | 0.926 (0.004) |
| | | 100 | 0.921 (0.003) | 0.925 (0.003) | | | 100 | 0.922 (0.004) | 0.923 (0.001) |
| Neural LSDE | 0.3 | 5 | 0.904 (0.004) | 0.910 (0.003) | Neural GSDE | 0.3 | 5 | 0.908 (0.005) | 0.914 (0.001) |
| | | 10 | 0.908 (0.008) | 0.920 (0.007) | | | 10 | 0.914 (0.005) | 0.912 (0.008) |
| | | 50 | 0.917 (0.003) | 0.927 (0.003) | | | 50 | 0.913 (0.003) | 0.918 (0.004) |
| | | 100 | 0.920 (0.009) | 0.921 (0.002) | | | 100 | 0.912 (0.007) | 0.920 (0.000) |
| | 0.4 | 5 | 0.907 (0.007) | 0.906 (0.003) | | 0.4 | 5 | 0.901 (0.012) | 0.903 (0.002) |
| | | 10 | 0.903 (0.002) | 0.915 (0.003) | | | 10 | 0.900 (0.010) | 0.909 (0.002) |
| | | 50 | 0.917 (0.005) | 0.920 (0.006) | | | 50 | 0.904 (0.014) | 0.920 (0.003) |
| | | 100 | 0.909 (0.005) | 0.915 (0.001) | | | 100 | 0.916 (0.005) | 0.921 (0.004) |
| | 0.5 | 5 | 0.886 (0.004) | 0.893 (0.006) | | 0.5 | 5 | 0.891 (0.011) | 0.899 (0.005) |
| | | 10 | 0.897 (0.005) | 0.902 (0.004) | | | 10 | 0.897 (0.009) | 0.907 (0.004) |
| | | 50 | 0.909 (0.004) | 0.912 (0.005) | | | 50 | 0.911 (0.002) | 0.915 (0.005) |
| | | 100 | 0.915 (0.003) | 0.912 (0.003) | | | 100 | 0.909 (0.004) | 0.915 (0.004) |

Table 12: Results of hyperparameter tuning for 'PhysioNet Sepsis' without observation intensity

| $p$ | $m$ | GRU-ODE | ODE-RNN | ANCDE |
|---|---|---|---|---|
| 0 | - | 0.771 (0.024) | 0.833 (0.020) | 0.823 (0.003) |
| 0.1 | 5 | 0.787 (0.006) | 0.830 (0.001) | **0.843 (0.007)** |
| | 10 | 0.803 (0.002) | 0.831 (0.001) | 0.833 (0.013) |
| | 50 | **0.805 (0.009)** | 0.835 (0.004) | 0.833 (0.013) |
| | 100 | 0.792 (0.007) | 0.831 (0.001) | 0.831 (0.014) |
| 0.2 | 5 | 0.790 (0.014) | 0.838 (0.003) | 0.838 (0.005) |
| | 10 | 0.784 (0.015) | 0.825 (0.001) | 0.833 (0.012) |
| | 50 | 0.783 (0.014) | 0.832 (0.004) | 0.825 (0.018) |
| | 100 | 0.769 (0.027) | 0.824 (0.003) | 0.829 (0.015) |
| 0.3 | 5 | 0.788 (0.008) | 0.833 (0.001) | 0.826 (0.001) |
| | 10 | 0.731 (0.027) | 0.830 (0.003) | 0.822 (0.009) |
| | 50 | 0.787 (0.017) | 0.839 (0.001) | 0.818 (0.012) |
| | 100 | 0.784 (0.016) | **0.842 (0.002)** | 0.824 (0.010) |
| 0.4 | 5 | 0.756 (0.023) | 0.832 (0.003) | 0.802 (0.009) |
| | 10 | 0.752 (0.046) | 0.829 (0.002) | 0.801 (0.024) |
| | 50 | 0.782 (0.011) | 0.838 (0.002) | 0.807 (0.017) |
| | 100 | 0.766 (0.009) | 0.837 (0.003) | 0.814 (0.017) |
| 0.5 | 5 | 0.731 (0.026) | 0.834 (0.005) | 0.784 (0.006) |
| | 10 | 0.752 (0.020) | 0.836 (0.013) | 0.792 (0.008) |
| | 50 | 0.779 (0.010) | 0.827 (0.001) | 0.799 (0.019) |
| | 100 | 0.780 (0.008) | 0.834 (0.005) | 0.798 (0.015) |

| Models | $p$ | $m$ | Test AUROC | | Models | $p$ | $m$ | Test AUROC | |
|---|---|---|---|---|---|---|---|---|---|
| | | | Optimal | Complex | | | | Optimal | Complex |
| | 0 | - | 0.841 (0.007) | 0.841 (0.007) | | 0 | - | 0.881 (0.002) | 0.859 (0.005) |
| | 0.1 | 5 | 0.855 (0.003) | 0.855 (0.003) | | 0.1 | 5 | 0.879 (0.002) | 0.884 (0.003) |
| | | 10 | 0.833 (0.021) | 0.833 (0.021) | | | 10 | 0.878 (0.004) | 0.886 (0.003) |
| | | 50 | 0.838 (0.003) | 0.838 (0.003) | | | 50 | 0.884 (0.002) | 0.886 (0.006) |
| | | 100 | 0.842 (0.001) | 0.842 (0.001) | | | 100 | 0.881 (0.006) | 0.882 (0.004) |
| | 0.2 | 5 | 0.848 (0.007) | 0.848 (0.007) | | 0.2 | 5 | 0.877 (0.002) | 0.886 (0.003) |
| | | 10 | 0.853 (0.004) | 0.853 (0.004) | | | 10 | 0.883 (0.004) | 0.883 (0.004) |
| | | 50 | 0.856 (0.006) | 0.856 (0.006) | | | 50 | 0.879 (0.004) | 0.882 (0.003) |
| | | 100 | 0.852 (0.008) | 0.852 (0.008) | | | 100 | 0.886 (0.002) | 0.880 (0.006) |
| Neural CDE | 0.3 | 5 | 0.851 (0.001) | 0.851 (0.001) | Neural LNSDE | 0.3 | 5 | 0.885 (0.010) | 0.870 (0.011) |
| | | 10 | 0.849 (0.009) | 0.849 (0.009) | | | 10 | 0.875 (0.007) | 0.880 (0.002) |
| | | 50 | 0.851 (0.009) | 0.851 (0.009) | | | 50 | 0.884 (0.005) | **0.890 (0.005)** |
| | | 100 | 0.852 (0.006) | 0.852 (0.006) | | | 100 | 0.884 (0.006) | 0.884 (0.004) |
| | 0.4 | 5 | 0.833 (0.004) | 0.833 (0.004) | | 0.4 | 5 | 0.876 (0.007) | 0.876 (0.008) |
| | | 10 | 0.838 (0.011) | 0.838 (0.011) | | | 10 | 0.885 (0.004) | 0.875 (0.002) |
| | | 50 | 0.854 (0.003) | 0.854 (0.003) | | | 50 | 0.879 (0.000) | 0.881 (0.000) |
| | | 100 | **0.860 (0.001)** | **0.860 (0.001)** | | | 100 | 0.884 (0.002) | 0.886 (0.002) |
| | 0.5 | 5 | 0.814 (0.014) | 0.814 (0.014) | | 0.5 | 5 | 0.856 (0.024) | 0.863 (0.006) |
| | | 10 | 0.829 (0.012) | 0.829 (0.012) | | | 10 | 0.875 (0.006) | 0.877 (0.008) |
| | | 50 | 0.843 (0.003) | 0.843 (0.003) | | | 50 | 0.885 (0.003) | 0.884 (0.008) |
| | | 100 | 0.829 (0.011) | 0.829 (0.011) | | | 100 | 0.878 (0.003) | 0.884 (0.004) |
| | 0 | - | 0.879 (0.008) | 0.866 (0.006) | | 0 | - | 0.884 (0.002) | 0.875 (0.003) |
| | 0.1 | 5 | **0.894 (0.005)** | 0.878 (0.007) | | 0.1 | 5 | 0.883 (0.006) | 0.885 (0.001) |
| | | 10 | 0.886 (0.004) | 0.880 (0.004) | | | 10 | 0.881 (0.002) | 0.884 (0.002) |
| | | 50 | 0.887 (0.001) | 0.882 (0.004) | | | 50 | 0.878 (0.010) | 0.884 (0.002) |
| | | 100 | 0.887 (0.006) | 0.885 (0.005) | | | 100 | 0.877 (0.009) | 0.888 (0.003) |
| | 0.2 | 5 | 0.875 (0.001) | 0.882 (0.002) | | 0.2 | 5 | 0.880 (0.005) | 0.883 (0.002) |
| | | 10 | 0.879 (0.005) | 0.873 (0.003) | | | 10 | **0.890 (0.002)** | 0.876 (0.006) |
| | | 50 | 0.885 (0.006) | 0.882 (0.002) | | | 50 | 0.880 (0.002) | 0.883 (0.004) |
| | | 100 | 0.885 (0.007) | 0.883 (0.003) | | | 100 | 0.886 (0.001) | 0.884 (0.005) |
| Neural LSDE | 0.3 | 5 | 0.875 (0.005) | 0.874 (0.007) | Neural GSDE | 0.3 | 5 | 0.883 (0.010) | 0.878 (0.001) |
| | | 10 | 0.880 (0.003) | 0.879 (0.003) | | | 10 | 0.876 (0.002) | 0.882 (0.006) |
| | | 50 | 0.883 (0.003) | 0.886 (0.008) | | | 50 | 0.879 (0.006) | 0.886 (0.006) |
| | | 100 | 0.888 (0.002) | 0.875 (0.003) | | | 100 | 0.887 (0.002) | 0.880 (0.003) |
| | 0.4 | 5 | 0.877 (0.005) | 0.874 (0.011) | | 0.4 | 5 | 0.876 (0.004) | 0.873 (0.009) |
| | | 10 | 0.882 (0.007) | 0.873 (0.006) | | | 10 | 0.882 (0.002) | 0.884 (0.011) |
| | | 50 | 0.882 (0.004) | 0.882 (0.009) | | | 50 | 0.888 (0.006) | 0.885 (0.003) |
| | | 100 | 0.887 (0.006) | 0.875 (0.007) | | | 100 | 0.879 (0.004) | 0.879 (0.004) |
| | 0.5 | 5 | 0.866 (0.006) | 0.872 (0.011) | | 0.5 | 5 | 0.874 (0.014) | 0.865 (0.008) |
| | | 10 | 0.872 (0.006) | 0.888 (0.001) | | | 10 | 0.882 (0.003) | 0.874 (0.006) |
| | | 50 | 0.880 (0.005) | 0.885 (0.004) | | | 50 | 0.882 (0.006) | 0.878 (0.004) |
| | | 100 | 0.885 (0.003) | 0.887 (0.004) | | | 100 | 0.878 (0.005) | 0.880 (0.003) |

### E.3 IMAGE CLASSIFICATION

Table 13: Results of hyperparameter tuning in image classification, with top 5 accuracy for CIFAR-100 and top 1 accuracy for CIFAR-10, STL-10, and SVHN

(a) CIFAR-100

| $p$ | $m$ | Neural ODE | Neural SDE (additive) | Neural SDE (multiplicative) |
|---|---|---|---|---|
| 0 | - | 74.475 (0.581) | 74.878 (0.328) | 75.317 (0.338) |
| 0.1 | 5 | 75.515 (0.077) | 75.941 (0.677) | 76.206 (0.519) |
| | 10 | 75.728 (0.235) | 75.371 (0.445) | 75.993 (0.664) |
| | 50 | 75.493 (0.341) | 75.716 (0.345) | 76.864 (0.610) |
| | 100 | 75.870 (0.124) | 76.283 (0.787) | 76.119 (0.354) |
| 0.2 | 5 | 75.725 (0.540) | 76.234 (0.227) | 76.676 (0.521) |
| | 10 | 75.787 (0.530) | 75.577 (0.817) | 76.111 (0.455) |
| | 50 | 75.547 (0.174) | **76.872 (0.442)** | 76.140 (0.745) |
| | 100 | 76.155 (0.509) | 75.808 (0.421) | 76.048 (0.666) |
| 0.3 | 5 | 75.692 (0.221) | 76.624 (0.664) | 76.298 (0.640) |
| | 10 | 75.740 (0.275) | 76.145 (0.497) | 76.759 (0.523) |
| | 50 | 75.833 (0.291) | 76.269 (0.474) | **76.947 (0.458)** |
| | 100 | 76.108 (0.203) | 76.014 (0.860) | 75.863 (0.381) |
| 0.4 | 5 | 75.845 (0.178) | 75.076 (0.552) | 76.321 (0.835) |
| | 10 | 76.058 (0.279) | 76.051 (0.708) | 76.542 (0.470) |
| | 50 | **76.470 (0.480)** | 75.860 (0.854) | 76.251 (0.397) |
| | 100 | 75.938 (0.509) | 76.232 (0.364) | 76.253 (0.999) |
| 0.5 | 5 | 76.008 (0.472) | 76.058 (0.283) | 76.370 (0.532) |
| | 10 | 76.052 (0.316) | 75.952 (0.289) | 75.966 (0.699) |
| | 50 | 76.205 (0.202) | 75.681 (0.564) | 76.558 (0.482) |
| | 100 | 76.240 (0.478) | 76.291 (0.395) | 76.587 (0.475) |

(b) CIFAR-10

| $p$ | $m$ | Neural ODE | Neural SDE (additive) | Neural SDE (multiplicative) |
|---|---|---|---|---|
| 0 | - | 73.870 (0.820) | 74.457 (0.910) | 74.670 (0.875) |
| 0.1 | 5 | 75.295 (1.636) | 75.862 (0.612) | 75.778 (0.979) |
| | 10 | 75.970 (1.333) | 76.206 (0.714) | 75.897 (0.294) |
| | 50 | 75.270 (2.477) | 75.635 (0.993) | 76.232 (0.536) |
| | 100 | 75.413 (2.044) | 76.157 (1.103) | 76.125 (0.704) |
| 0.2 | 5 | 75.585 (1.226) | **77.102 (0.225)** | 75.977 (0.778) |
| | 10 | 76.665 (1.933) | 76.326 (0.645) | **76.654 (0.696)** |
| | 50 | 75.385 (1.651) | 76.601 (0.457) | 76.054 (0.544) |
| | 100 | 74.825 (1.066) | 76.304 (0.474) | 76.520 (0.628) |
| 0.3 | 5 | 74.805 (1.826) | 76.362 (0.877) | 76.420 (0.962) |
| | 10 | 75.108 (2.974) | 75.519 (0.699) | 75.857 (0.913) |
| | 50 | 76.325 (2.046) | 76.257 (0.553) | 75.983 (0.526) |
| | 100 | 76.480 (2.315) | 76.031 (1.166) | 75.968 (0.546) |
| 0.4 | 5 | 74.600 (1.881) | 76.102 (0.697) | 75.807 (1.085) |
| | 10 | 74.190 (0.679) | 76.151 (0.749) | 75.967 (0.928) |
| | 50 | 76.060 (0.815) | 76.214 (0.879) | 76.562 (0.679) |
| | 100 | 75.957 (1.706) | 76.277 (0.662) | 76.496 (0.687) |
| 0.5 | 5 | **76.877 (0.615)** | 75.804 (0.745) | 76.050 (0.850) |
| | 10 | 75.145 (2.001) | 75.563 (0.765) | 75.997 (0.800) |
| | 50 | 75.145 (1.730) | 75.651 (0.848) | 76.283 (0.592) |
| | 100 | 75.562 (0.620) | 76.338 (0.791) | 76.427 (0.908) |

(c) STL-10

| $p$ | $m$ | Neural ODE | Neural SDE (additive) | Neural SDE (multiplicative) |
|---|---|---|---|---|
| 0 | - | 70.650 (0.688) | 70.666 (0.354) | 70.463 (0.485) |
| 0.1 | 5 | 70.894 (0.559) | 71.471 (0.260) | 70.990 (0.429) |
| | 10 | 71.069 (0.211) | 71.061 (0.247) | 70.892 (0.180) |
| | 50 | 70.250 (0.533) | 71.248 (0.475) | 71.099 (0.656) |
| | 100 | 70.319 (0.214) | 70.651 (0.142) | 71.177 (0.311) |
| 0.2 | 5 | 70.997 (0.437) | 70.856 (0.211) | 71.099 (0.209) |
| | 10 | 70.947 (0.340) | 70.670 (0.461) | 71.052 (0.459) |
| | 50 | 71.022 (0.615) | 71.153 (0.312) | 70.950 (0.234) |
| | 100 | 70.941 (0.477) | 71.539 (0.215) | 70.712 (0.583) |
| 0.3 | 5 | 70.950 (0.310) | 70.794 (0.385) | **71.484 (0.591)** |
| | 10 | 71.328 (0.313) | 71.296 (0.610) | 70.966 (0.829) |
| | 50 | 70.912 (0.182) | **71.650 (0.253)** | 71.265 (0.409) |
| | 100 | 70.909 (0.560) | 71.176 (0.759) | 71.250 (0.396) |
| 0.4 | 5 | 70.825 (0.376) | 71.351 (0.217) | 71.225 (0.188) |
| | 10 | 70.913 (0.438) | 70.997 (0.607) | 70.719 (0.586) |
| | 50 | 70.872 (0.322) | 71.390 (0.745) | 71.031 (0.255) |
| | 100 | 70.825 (0.260) | 71.288 (0.373) | 71.226 (0.470) |
| 0.5 | 5 | **71.833 (0.334)** | 70.670 (0.206) | 71.280 (0.787) |
| | 10 | 71.588 (0.166) | 71.336 (0.306) | 71.090 (0.216) |
| | 50 | 71.584 (0.445) | 71.631 (0.172) | 71.135 (0.703) |
| | 100 | 70.506 (0.080) | 71.619 (0.532) | 71.167 (0.129) |

(d) SVHN

| $p$ | $m$ | Neural ODE | Neural SDE (additive) | Neural SDE (multiplicative) |
|---|---|---|---|---|
| 0 | - | 91.348 (0.440) | 91.839 (0.255) | 91.419 (0.577) |
| 0.1 | 5 | 92.199 (0.095) | 92.256 (0.124) | 92.183 (0.155) |
| | 10 | 92.174 (0.106) | 92.214 (0.132) | 92.231 (0.144) |
| | 50 | 92.157 (0.132) | 92.139 (0.139) | 91.977 (0.146) |
| | 100 | 92.262 (0.124) | 92.114 (0.199) | 92.250 (0.124) |
| 0.2 | 5 | 92.185 (0.165) | 91.914 (0.123) | 92.355 (0.044) |
| | 10 | 92.174 (0.106) | 92.261 (0.186) | 92.249 (0.258) |
| | 50 | 92.327 (0.201) | **92.418 (0.153)** | 92.022 (0.119) |
| | 100 | 92.262 (0.124) | 92.366 (0.196) | 92.355 (0.209) |
| 0.3 | 5 | 92.332 (0.170) | 92.260 (0.144) | 92.311 (0.098) |
| | 10 | 92.241 (0.135) | 92.053 (0.068) | 92.332 (0.108) |
| | 50 | 92.156 (0.209) | 92.122 (0.145) | 92.127 (0.107) |
| | 100 | 92.277 (0.082) | 92.221 (0.111) | **92.434 (0.149)** |
| 0.4 | 5 | 92.331 (0.146) | 92.306 (0.072) | 92.055 (0.153) |
| | 10 | 92.317 (0.194) | 91.882 (0.113) | 92.349 (0.173) |
| | 50 | 92.310 (0.273) | 92.264 (0.092) | 91.958 (0.182) |
| | 100 | 92.172 (0.086) | 92.181 (0.094) | 92.176 (0.156) |
| 0.5 | 5 | 92.237 (0.125) | 92.155 (0.131) | 92.044 (0.139) |
| | 10 | 92.240 (0.142) | 92.072 (0.181) | 92.291 (0.186) |
| | 50 | **92.381 (0.083)** | 92.155 (0.115) | 92.367 (0.154) |
| | 100 | 92.225 (0.022) | 92.066 (0.198) | 92.014 (0.092) |

Figure 5 illustrates the impact of dropout on generalization loss, which is assessed on the validation set during the training process. As anticipated, the implementation of dropout helps to mitigate overfitting, thereby enhancing overall model performance in test set.

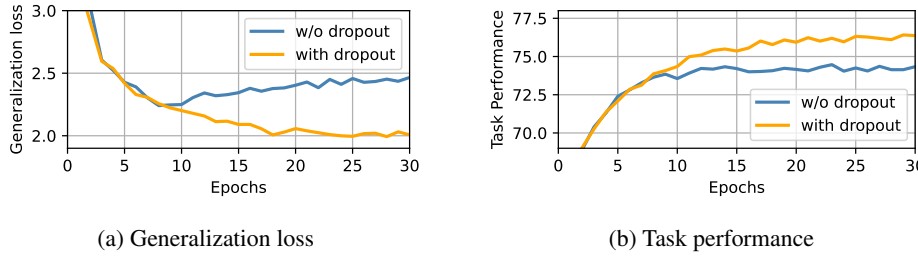

(a) Generalization loss        (b) Task performance

Figure 5: Monitoring Neural ODE training on CIFAR-100

Figures 6–9 illustrate how the performance of NRSDEs for Neural ODE and Neural SDEs varies with different hyperparameters $p$ and $m$, showing that the proposed dropout scheme robustly enhances performance regardless of the choice of $p$ and $m$. Black dashed line and gray area indicate the mean and standard deviation of method without dropout.

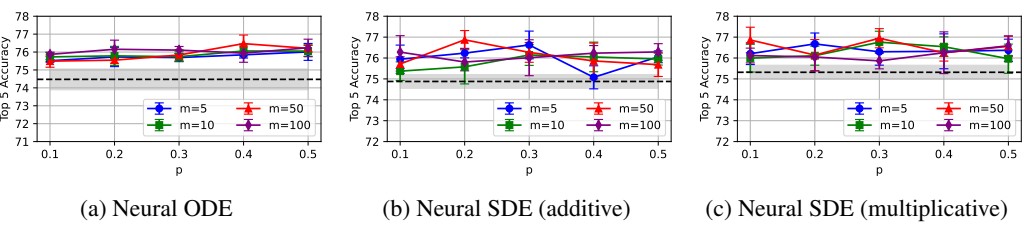

(a) Neural ODE      (b) Neural SDE (additive)      (c) Neural SDE (multiplicative)

Figure 6: Performance comparison with different hyperparameters on CIFAR-100

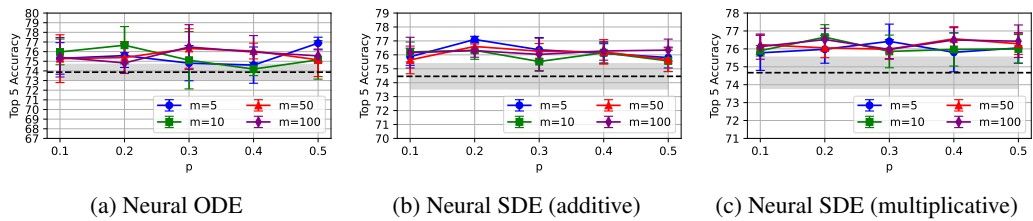

(a) Neural ODE      (b) Neural SDE (additive)      (c) Neural SDE (multiplicative)

Figure 7: Performance comparison with different hyperparameters on CIFAR-10

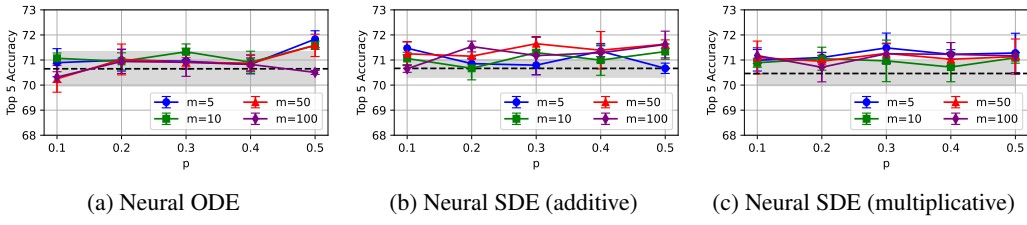

(a) Neural ODE      (b) Neural SDE (additive)      (c) Neural SDE (multiplicative)

Figure 8: Performance comparison with different hyperparameters on STL-10

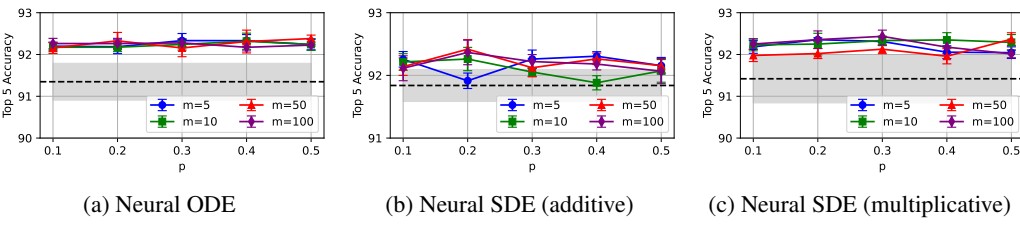

(a) Neural ODE      (b) Neural SDE (additive)      (c) Neural SDE (multiplicative)

Figure 9: Performance comparison with different hyperparameters on SVHN

## E.4 EXTENDED EXPERIMENTS ON TIME SERIES DATASETS

We conducted a comprehensive evaluation of the proposed dropout technique using 30 diverse datasets from the University of East Anglia (UEA) and University of California Riverside (UCR) Time Series Classification Repository[13] (Bagnall et al., 2018; Dau et al., 2019). This analysis was facilitated by the Python library `sktime`[14] (Löning et al., 2019). We followed the experiment protocol and code repository based on Oh et al. (2024).

Table 14: Data description for extended experiments

| Dataset | Total number of samples | Number of classes | Dimension of time series | Length of time series |
|---|---|---|---|---|
| ArrowHead | 211 | 3 | 1 | 251 |
| Car | 120 | 4 | 1 | 577 |
| Coffee | 56 | 2 | 1 | 286 |
| GunPoint | 200 | 2 | 1 | 150 |
| Herring | 128 | 2 | 1 | 512 |
| Lightning2 | 121 | 2 | 1 | 637 |
| Lightning7 | 143 | 7 | 1 | 319 |
| Meat | 120 | 3 | 1 | 448 |
| OliveOil | 60 | 4 | 1 | 570 |
| Rock | 70 | 4 | 1 | 2844 |
| SmoothSubspace | 300 | 3 | 1 | 15 |
| ToeSegmentation1 | 268 | 2 | 1 | 277 |
| ToeSegmentation2 | 166 | 2 | 1 | 343 |
| Trace | 200 | 4 | 1 | 275 |
| Wine | 111 | 2 | 1 | 234 |
| ArticularyWordRecognition | 575 | 25 | 9 | 144 |
| BasicMotions | 80 | 4 | 6 | 100 |
| CharacterTrajectories | 2858 | 20 | 3 | 60-180 |
| Cricket | 180 | 12 | 6 | 1197 |
| Epilepsy | 275 | 4 | 3 | 206 |
| ERing | 300 | 6 | 4 | 65 |
| EthanolConcentration | 524 | 4 | 3 | 1751 |
| EyesOpenShut | 98 | 2 | 14 | 128 |
| FingerMovements | 416 | 2 | 28 | 50 |
| Handwriting | 1000 | 26 | 3 | 152 |
| JapaneseVowels | 640 | 9 | 12 | 7-26 |
| Libras | 360 | 15 | 2 | 45 |
| NATOPS | 360 | 6 | 24 | 51 |
| RacketSports | 303 | 4 | 6 | 30 |
| SpokenArabicDigits | 8798 | 10 | 13 | 4-93 |

Table 15 presents a comprehensive overview of the performance improvements achieved by our proposed dropout technique (NRSDE) across various methods. The results demonstrate significant enhancements in classification accuracy across the 30 datasets examined. Notably, the impact was more pronounced for Neural ODE and Neural CDE models compared to Neural SDE-based approaches.

Table 15: Comprehensive performance analysis on extended datasets (Results are averaged across 30 diverse datasets. Values in parentheses represent the mean of individual standard deviations. Improvement and percentage change are reported as average $\pm$ standard error of the mean.)

| Method | Baseline | Proposed | Improvement | % Change |
|---|---|---|---|---|
| Neural ODE | 0.521 (0.065) | 0.568 (0.054) | 0.047 ± 0.006 | 11.56 ± 2.02 |
| Neural CDE | 0.709 (0.061) | 0.781 (0.048) | 0.072 ± 0.012 | 13.39 ± 2.70 |
| Neural SDE | 0.526 (0.068) | 0.571 (0.065) | 0.045 ± 0.008 | 9.69 ± 1.87 |
| Neural LSDE | 0.717 (0.056) | 0.741 (0.061) | 0.024 ± 0.010 | 4.19 ± 1.65 |
| Neural LNSDE | 0.727 (0.047) | 0.761 (0.056) | 0.034 ± 0.007 | 5.93 ± 1.38 |
| Neural GSDE | 0.716 (0.065) | 0.752 (0.063) | 0.036 ± 0.009 | 6.00 ± 1.50 |

Tables 16 and 17 offer a detailed decomposition of our experimental outcomes, stratifying the 30 datasets into two equal subsets: 15 univariate and 15 multivariate time series. Our proposed dropout technique demonstrates heightened efficacy when applied to univariate datasets. The contrasting results between these two categories underscore the importance of considering data dimensionality when applying proposed dropout method to neural differential equation models.

---

[13]http://www.timeseriesclassification.com/
[14]https://github.com/sktime/sktime

Table 16: Comprehensive performance analysis on 15 univariate datasets

| Method | Baseline | Proposed | Improvement | % Change |
|---|---|---|---|---|
| **Neural ODE** | 0.535 (0.073) | 0.593 (0.068) | 0.058 ± 0.009 | 11.73 ± 1.93 |
| **Neural CDE** | 0.628 (0.079) | 0.725 (0.065) | 0.098 ± 0.020 | 19.33 ± 4.63 |
| **Neural SDE** | 0.535 (0.086) | 0.597 (0.082) | 0.062 ± 0.013 | 12.99 ± 3.23 |
| **Neural LSDE** | 0.636 (0.082) | 0.683 (0.089) | 0.047 ± 0.013 | 8.12 ± 2.35 |
| **Neural LNSDE** | 0.665 (0.065) | 0.718 (0.078) | 0.053 ± 0.010 | 9.41 ± 2.19 |
| **Neural GSDE** | 0.649 (0.097) | 0.711 (0.096) | 0.062 ± 0.015 | 10.30 ± 2.50 |

Table 17: Comprehensive performance analysis on 15 multivariate datasets

| Method | Baseline | Proposed | Improvement | % Change |
|---|---|---|---|---|
| **Neural ODE** | 0.507 (0.056) | 0.544 (0.041) | 0.037 ± 0.009 | 11.38 ± 3.63 |
| **Neural CDE** | 0.790 (0.044) | 0.837 (0.031) | 0.047 ± 0.010 | 7.45 ± 1.93 |
| **Neural SDE** | 0.516 (0.049) | 0.545 (0.048) | 0.028 ± 0.007 | 6.38 ± 1.57 |
| **Neural LSDE** | 0.798 (0.031) | 0.800 (0.032) | 0.002 ± 0.013 | 0.27 ± 1.88 |
| **Neural LNSDE** | 0.789 (0.030) | 0.805 (0.034) | 0.016 ± 0.007 | 2.45 ± 1.18 |
| **Neural GSDE** | 0.783 (0.034) | 0.794 (0.030) | 0.011 ± 0.005 | 1.71 ± 0.69 |

Figure 10 illustrates the performance variations across different hyperparameter configurations. The baseline performance without dropout is represented by a black dashed horizontal line. Our results consistently demonstrate that the proposed dropout method enhances classification performance relative to this baseline across various hyperparameter settings. This robust improvement highlights the method's effectiveness in bolstering generalization capabilities and its versatility in addressing diverse time series classification tasks, further validating its practical utility in real-world problems.

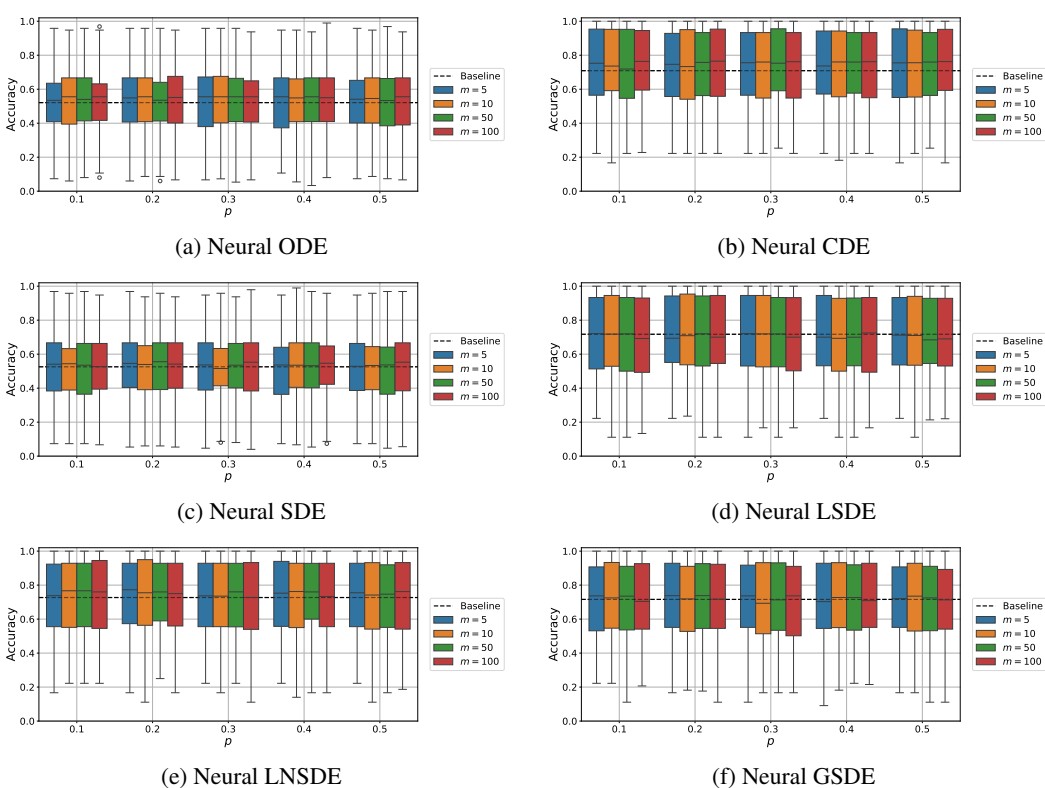

(a) Neural ODE

(b) Neural CDE

(c) Neural SDE

(d) Neural LSDE

(e) Neural LNSDE

(f) Neural GSDE

Figure 10: Performance comparison with different hyperparameters on extended datasets

# F FURTHER ANALYSIS OF NRSDE

## F.1 COMPREHENSIVE ANALYSIS OF COMPUTATIONAL EFFICIENCY

We performed comprehensive experiments to evaluate the computational overhead of our proposed NRSDE method with varying dropout rates $p$ and expected renewal counts $m$. Table 18 presents the average computation times per epoch on Speech Commands dataset. The results indicate that while the NRSDE method generally increases computation time compared to the baseline, the impact of different $p$ and $m$ values is relatively minor. Also, since Monte Carlo simulation is only conducted during the test phase, we observe that it does not significantly increase the computational overhead. In particular, with Monte Carlo simulation using 5 samples, the computational overhead remains almost negligible, showing only a marginal increase compared to the baseline.

Table 18: Computation time comparison on Speech Commands (time in seconds per epoch)

(a) Monte Carlo simulation with 5 samples

| $p$ | $m$ | Neural CDE | ANCDE | Neural LSDE | Neural LNSDE | Neural GSDE |
|-----|-----|-----------|-------|-------------|--------------|-------------|
| 0 | - | 25.560 (0.259) | 53.264 (0.157) | 19.416 (0.147) | 19.532 (0.109) | 19.776 (0.052) |
| 0.1 | 5 | 28.056 (0.262) | 60.208 (0.292) | 22.086 (0.158) | 22.499 (0.223) | 23.095 (0.233) |
|  | 10 | 28.182 (0.258) | 59.914 (0.199) | 22.076 (0.171) | 22.207 (0.173) | 22.807 (0.112) |
|  | 50 | 28.014 (0.245) | 59.339 (0.307) | 22.157 (0.150) | 22.213 (0.161) | 23.178 (0.246) |
|  | 100 | 28.375 (0.508) | 60.100 (0.340) | 22.332 (0.190) | 22.128 (0.122) | 22.939 (0.131) |
| 0.3 | 5 | 27.180 (0.366) | 59.148 (0.408) | 21.576 (0.155) | 21.668 (0.171) | 22.496 (0.176) |
|  | 10 | 27.226 (0.305) | 59.156 (0.367) | 21.610 (0.135) | 21.632 (0.144) | 22.460 (0.216) |
|  | 50 | 27.064 (0.248) | 59.062 (0.373) | 21.589 (0.214) | 21.646 (0.199) | 22.482 (0.183) |
|  | 100 | 27.145 (0.255) | 59.093 (0.276) | 21.580 (0.214) | 21.740 (0.131) | 22.436 (0.174) |
| 0.5 | 5 | 26.052 (0.255) | 58.108 (0.229) | 21.045 (0.094) | 20.790 (0.174) | 21.818 (0.089) |
|  | 10 | 26.393 (0.276) | 57.643 (0.136) | 21.058 (0.097) | 21.185 (0.091) | 21.893 (0.088) |
|  | 50 | 25.688 (0.219) | 57.646 (0.407) | 21.025 (0.135) | 21.176 (0.120) | 21.931 (0.116) |
|  | 100 | 26.144 (0.314) | 57.905 (0.265) | 21.005 (0.104) | 21.098 (0.144) | 21.960 (0.084) |

(b) Monte Carlo simulation with 10 samples

| $p$ | $m$ | Neural CDE | ANCDE | Neural LSDE | Neural LNSDE | Neural GSDE |
|-----|-----|-----------|-------|-------------|--------------|-------------|
| 0 | - | 25.560 (0.259) | 53.264 (0.157) | 19.416 (0.147) | 19.532 (0.109) | 19.776 (0.052) |
| 0.1 | 5 | 33.485 (0.346) | 76.222 (0.245) | 30.518 (0.171) | 30.358 (0.099) | 32.352 (0.249) |
|  | 10 | 33.195 (0.245) | 76.630 (0.251) | 30.736 (0.186) | 30.502 (0.066) | 32.255 (0.284) |
|  | 50 | 33.430 (0.252) | 76.202 (0.245) | 30.393 (0.107) | 30.467 (0.102) | 31.895 (0.194) |
|  | 100 | 33.944 (0.511) | 76.638 (0.366) | 30.642 (0.173) | 30.283 (0.058) | 32.064 (0.219) |
| 0.3 | 5 | 32.252 (0.372) | 75.468 (0.285) | 30.382 (0.154) | 30.084 (0.112) | 31.256 (0.296) |
|  | 10 | 32.152 (0.370) | 75.500 (0.215) | 30.084 (0.148) | 30.046 (0.115) | 31.201 (0.356) |
|  | 50 | 32.078 (0.327) | 75.448 (0.176) | 30.208 (0.193) | 30.039 (0.130) | 31.276 (0.208) |
|  | 100 | 32.108 (0.376) | 75.603 (0.238) | 30.193 (0.108) | 30.187 (0.152) | 31.251 (0.198) |
| 0.5 | 5 | 30.652 (0.368) | 74.664 (0.123) | 29.545 (0.098) | 29.633 (0.077) | 30.092 (0.162) |
|  | 10 | 31.356 (0.192) | 74.390 (0.243) | 29.446 (0.144) | 29.614 (0.073) | 30.255 (0.179) |
|  | 50 | 30.688 (0.287) | 74.645 (0.250) | 29.458 (0.151) | 29.601 (0.102) | 30.308 (0.215) |
|  | 100 | 30.722 (0.299) | 74.457 (0.244) | 29.463 (0.149) | 29.598 (0.099) | 30.351 (0.188) |

## F.2 VALIDATION OF SCALING FACTOR c ESTIMATION

We designed experiments to validate the scaling factor $\mathbf{c}$ estimation during the test phase. Table 19 and Table 20 show the optimal performance based on the number of Monte Carlo simulation samples (MC) on Speech Commands and CIFAR-100, respectively. We considered MC $= [0, 1, 3, 5, 10, 20]$, where MC $= 0$ indicates that no scaling is applied during the test phase, which corresponds to performing inference using $\mathbf{z}_0(t)$. We observed consistent performance improvements with only 5–10 samples, which supports the accurate estimation of $\mathbf{c}$. While 5 samples provide sufficient performance, this paper uses 10 samples as the default for experiments to achieve more stable results.

## F.3 SENSITIVITY ANALYSIS OF HYPERPARAMETERS

We conducted a sensitivity analysis to provide guidelines for selecting the hyperparameters $p$ and $m$. We evaluated the accuracy of Neural CDE successfully trained on Speech Commands dataset.

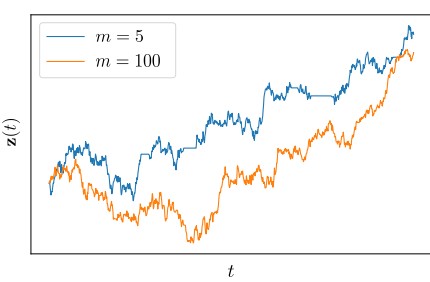
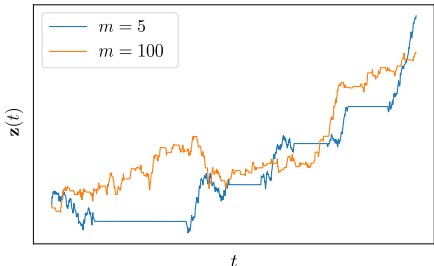

(a) NRSDE $\mathbf{z}(t)$ with dropout rate $p = 0.1$      (b) NRSDE $\mathbf{z}(t)$ with dropout rate $p = 0.5$

Figure 11: Illustration of dropout in NDEs with different hyperparameters $p$ and $m$

Table 19: Accuracy on Speech Commands with different numbers of Monte Carlo simulation samples

| Dropout | MC | Neural CDE | ANCDE | Neural LSDE | Neural LNSDE | Neural GSDE |
|---------|-----|------------|-------|-------------|--------------|-------------|
| X | - | 0.910 (0.005) | 0.760 (0.003) | 0.927 (0.004) | 0.923 (0.001) | 0.913 (0.001) |
| O | 0 | 0.934 (0.000) | 0.776 (0.003) | 0.932 (0.000) | 0.930 (0.003) | 0.920 (0.005) |
| | 1 | 0.908 (0.002) | 0.762 (0.002) | 0.931 (0.003) | 0.925 (0.002) | 0.919 (0.006) |
| | 3 | 0.929 (0.002) | 0.782 (0.006) | 0.932 (0.001) | 0.930 (0.000) | 0.927 (0.002) |
| | 5 | 0.940 (0.001) | **0.794 (0.003)** | 0.932 (0.000) | **0.932 (0.001)** | 0.927 (0.001) |
| | 10 | **0.945 (0.001)** | 0.793 (0.007) | **0.933 (0.001)** | 0.932 (0.002) | 0.930 (0.002) |
| | 20 | 0.943 (0.003) | 0.793 (0.005) | 0.932 (0.001) | 0.932 (0.002) | **0.930 (0.001)** |

Table 20: Performance on CIFAR-100 with different numbers of Monte Carlo simulation samples

| Dropout | MC | Neural ODE | Neural SDE (additive) | Neural SDE (multiplicative) |
|---------|-----|------------|-----------------------|------------------------------|
| X | - | 74.475 (0.581) | 74.878 (0.328) | 75.317 (0.338) |
| O | 0 | 75.822 (0.314) | 75.754 (0.390) | 75.925 (0.286) |
| | 1 | 76.365 (0.342) | 76.390 (0.263) | 76.006 (0.426) |
| | 3 | **76.848 (0.254)** | 76.547 (0.398) | 76.182 (0.522) |
| | 5 | 76.213 (0.238) | **76.958 (0.415)** | 76.787 (0.420) |
| | 10 | 76.470 (0.480) | 76.872 (0.442) | **76.947 (0.458)** |
| | 20 | 76.374 (0.298) | 76.704 (0.252) | 76.927 (0.404) |

To observe the performance variations with respect to $p$ and $m$, we set $p \in [0.1, 0.2, 0.3, 0.4, 0.5]$ and $m \in [3, 5, 10, 50, 100]$. Table 21 shows the performance for each combination of $p$ and $m$, and we highlighted in red the cases that exhibit lower performance than the without dropout. We observed a significant performance decline at high dropout rates when $m = 3$. This is because a smaller $m$ increases the variance of NRSDE $\mathbf{z}(t)$, hindering stable training. Consequently, we limited our experiments to $m \in [5, 10, 50, 100]$ in this paper. This choice robustly improves performance regardless of hyperparameters $p$ and $m$. Additionally, performance with respect to the choice of hyperparameters for various datasets and models can be found in Figures 6–10, so please refer to them for further insights.

Table 21: Comparison of average accuracy from the sensitivity analysis

| $p$ | $m$ | | | | |
|-----|-----|-----|-----|-----|-----|
| | 3 | 5 | 10 | 50 | 100 |
| 0.1 | 0.936 (0.003) | 0.940 (0.002) | 0.941 (0.001) | 0.942 (0.000) | 0.940 (0.003) |
| 0.2 | 0.939 (0.002) | 0.939 (0.001) | 0.942 (0.001) | 0.943 (0.002) | **0.945 (0.001)** |
| 0.3 | 0.923 (0.004) | 0.937 (0.005) | 0.938 (0.003) | 0.940 (0.001) | 0.941 (0.002) |
| 0.4 | 0.915 (0.008) | 0.932 (0.003) | 0.932 (0.002) | 0.935 (0.005) | 0.932 (0.005) |
| 0.5 | 0.908 (0.013) | 0.927 (0.004) | 0.920 (0.008) | 0.926 (0.003) | 0.920 (0.002) |

## G  LIMITATIONS OF JUMP DIFFUSION DROPOUT

In this section, we examine more thoroughly the limitations of the jump diffusion dropout mentioned in the main text. Specifically, Appendix G.1 analyzes the theoretical issues of jump diffusion dropout,

while Appendix G.2 demonstrates, through experiments, why jump diffusion dropout cannot be universally applicable to various variants of NDEs.

## G.1 DISCRETIZATION

Liu et al. (2020) claimed that

$$\mathbf{Z}(k+1) = \mathbf{Z}(k) + \gamma(\mathbf{Z}(k); \theta_\gamma) \circ \xi$$
$$= \mathbf{Z}(k) + \frac{1}{2}\gamma(\mathbf{Z}(k); \theta_\gamma) + \frac{1}{2}\gamma(\mathbf{Z}(k); \theta_\gamma) \circ \Xi \tag{13}$$

with $k = 0, 1, \ldots, N-1$, $\mathbb{P}(\xi^{(i)} = 0) = 1 - \mathbb{P}(\xi^{(i)} = 1) = p$ for $i = 1, \ldots, d_z$ and $\Xi = 2\xi - 1$ is a discrete version of the following jump diffusion process: for $0 \le t \le T$,

$$\mathbf{z}(t) = \mathbf{z}(0) + \int_0^t \frac{1}{2}\gamma(\mathbf{z}(\tau); \theta_\gamma)\, \mathrm{d}\tau + \int_0^t \frac{1}{2}\gamma(\mathbf{z}(\tau); \theta_\gamma) \circ \Xi_{N_\tau}\, \mathrm{d}N_\tau, \tag{14}$$

where $N_\tau$ is a Poisson counting process.

However, that claim is not only incomplete in defining equation 13, but even if we were to correct it properly, our analysis reveals theoretical inconsistencies in this approach. More specifically, the time step size $\Delta t$ has not been considered in equation 13, which is essential for it to be a valid discrete approximation of the continuous jump diffusion process. In fact, a correct Euler discretization of equation 14 is given by

$$\mathbf{Z}(k+1) = \mathbf{Z}(k) + \frac{1}{2}\gamma(\mathbf{Z}(k); \theta_\gamma)\Delta t + \frac{1}{2}\gamma(\mathbf{Z}(k); \theta_\gamma) \circ \Xi_{N_k} \Delta N_k$$
$$= \mathbf{Z}(k) + \begin{cases} \frac{1}{2}\gamma(\mathbf{Z}(k); \theta_\gamma)\Delta t + \frac{1}{2}\gamma(\mathbf{Z}(k); \theta_\gamma) \circ \Xi, & \text{if } \Delta N_k = 1, \\ \frac{1}{2}\gamma(\mathbf{Z}(k); \theta_\gamma)\Delta t, & \text{if } \Delta N_k = 0, \end{cases}$$
$$\ne \mathbf{Z}(k) + \gamma(\mathbf{Z}(k); \theta_\gamma) \circ \xi,$$

where $\Delta N_k = N_{\frac{T}{N}k} - N_{\frac{T}{N}k^-} \in \{0, 1\}$.

In contrast, our proposed dropout scheme based on the alternating renewal processes (NRSDE) accurately extends the following discrete-time equation to the continuous-time process:

$$\mathbf{Z}(k+1) = \mathbf{Z}(k) + \gamma(\mathbf{Z}(k); \theta_\gamma)\Delta t \circ \xi_k.$$

## G.2 UNIVERSAL APPLICABILITY

We discuss why jump diffusion dropout is not universally applicable, unlike the proposed dropout method (NRSDE). Liu et al. (2020) introduced a new type of Neural SDE by adding a stochastic dropout term to Neural ODE. However, this approach is limited to ODE-based models. Nonetheless, from an engineering perspective, we compared the performance of NRSDE with the model, which combines jump diffusion dropout with the CDE-based model. Additionally, in the case of SDE-based models, the existing diffusion network dominates the jump diffusion dropout term, and thus, these models are not considered.

Table 22 presents the performance of jump diffusion dropout and the proposed dropout method (NRSDE), using Neural CDE as the baseline. When jump diffusion dropout is applied, performance either declines or, if it improves, the improvement is not statistically significant. However, our method demonstrates statistically significant and highly successful performance improvements, experimentally proving that it is the only universally applicable dropout method.

Table 22: Performance of various dropout methods on Speech Commands

| Dropout Methods | Test Accuracy |
|---|---|
| Baseline (Neural CDE) | 0.910 (0.005) |
| Dropout of Liu et al. (2020) | 0.917 (0.009) |
| Dropout of Liu et al. (2020)+TTN | 0.906 (0.013) |
| NRSDE (ours) | **0.945 (0.001)**[**] |

