# OpenReview forum: "Neural Regenerative Stochastic Differential Equation: Dropout Scheme for Neural Differential Equations"
_ICLR.cc/2025/Conference — Submitted to ICLR 2025_

### Official Review · Reviewer_jDuQ · 2024-10-19

**Soundness:** 3
**Presentation:** 4
**Contribution:** 3
**Rating:** 6
**Confidence:** 5

**Summary:**

This paper is about taking dropout from feedforward networks and generalizing to continuous networks defined by Neural Differential Equations.

The paper identifies how to take dropout from the discrete to the continuous regime, as well as adapting test time behavior as is done in standard dropout.

The paper is well motivated, with a mixture of theory and more intuitive explanations. The evaluation is extensive and supports the use of this method over other Neural DE regularization schemes.

**Strengths:**

- The paper is very nicely written, there is a good mix of intuitive explanation with diagrams and theoretical insights.
- The contribution of generalizing dropout to continuous time networks is a valuable one.
- The evaluation is extensive and the results are convincing.

**Weaknesses:**

- Adjusting dropout to work at test time is an important part of this method. However, the approach is computationally expensive. My understanding is that around 5 additional forward passes of the NDE are required to estimate $c$. Is this required for every test point, or does this only need to be carried out once? Table 17 in the appendix would be improved by considering 0 samples so we can see the effect of including this correction term. The table would be further improved with a larger sensitivity analysis, including 1 sample, 2 samples etc. to see exactly when $c$ becomes useful.

**Questions:**

- I'm curious how vanilla dropout works on these datasets, where dropout is not applied to the output of the dynamics function, only the hidden layers. That way there are still full dynamics, but the dynamics function is still regularized. Is it possible to give numbers for this vanilla baseline?

---

> ### Author Response · Authors · 2024-11-25
>
> Thank you for your valuable comments. We have carefully studied your comments and have made every effort to address your concerns. Detailed responses to each comment are provided below.
>
> **Response to Weaknesses (W)**
>
> > **W1.**
>
> Thank you for your comment. You are correct that we use 5 additional forward passes of the NDE for each test point to estimate the scaling factor via Monte Carlo simulation. As you pointed out, this is one of the main limitations of our method.
>
> To defend against this limitation, it is worth noting that this computation occurs during the inference phase, not during training, which is typically the more time-intensive part of the process. Additionally, using 5 Monte Carlo samples introduces an overhead of approximately 6.1% to 13.6% in computational cost (see Table 6.). We believe this overhead remains within a reasonable range, particularly given the substantial performance improvements achieved over state-of-the-art methods. This trade-off makes our approach practical and advantageous for many applications.
>
> > **W2.**
>
> Thank you for the suggestion. We have conducted additional experiments comparing performance with different numbers of Monte Carlo (MC) simulation samples, including $\text{MC} = [0, 1, 3, 5, 10, 20]$. Here, $\text{MC} = 0$ means no scaling was applied during the test phase, effectively using $\mathbf{z}_0(t)$ for inference.
>
> These experiments were performed on both the Speech Commands (time-series dataset) and CIFAR-100 (image dataset). The results, presented in Section 4.3 and Appendix F.3, Tables 19 and 20 of the updated paper, show that performance converges after using 5-10 MC samples. This indicates that 5-10 samples are sufficient to accurately estimate the scaling factor $\mathbf{c}$, with no additional benefit observed beyond this range.
>
>
> > **Response to Question**
>
> Thank you for your interesting suggestion. In the current experiments, the results labeled as "Dropout for Drift Network" already report the performance of vanilla dropout applied to the dynamics function (drift network). Following your suggestion, we additionally conducted experiments by applying dropout only to the hidden layers of the MLP classifier. We refer to "Dropout for MLP Classifier".
>
> The results, presented in Tables 1 and 2 of the updated paper, indicate that the performance improvement with vanilla dropout was not significant. This suggests that applying conventional dropout to the drift function or MLP classifier does not lead to meaningful improvements in the performance of NDEs, highlighting the need for a novel dropout technique specifically designed for NDEs.
>
>
> Table 1. Accuracy of various regularization methods on time series classification
>
> | | SmoothSubspace            | ArticularyWordRecognition | ERing          | RacketSports    |
> |---------------------------|---------------------------|----------------|-----------------|---------------|
> | Baseline (Neural ODE)     | 0.569 (0.040)            | 0.859 (0.005)  | 0.839 (0.018)   | 0.565 (0.065)  |
> | Dropout for Drift Network| 0.594 (0.016)            | 0.862 (0.014)  | 0.844 (0.031)   | 0.598 (0.045)  |
> | Dropout for MLP Classifer | 0.600 (0.067)            | 0.860 (0.006)  | 0.856 (0.078)   | 0.554 (0.076)  |
> | NRSDE (Ours)              | 0.639 (0.018)            | 0.882 (0.040)  | 0.884 (0.025)   | 0.625 (0.044)  |
>
> Table 2. Performance of various regularization methods on image classification
>
> || CIFAR-100                 | CIFAR-10               | STL-10                | SVHN                 |
> |---------------------------|------------------------|-----------------------|----------------------|---------------|
> | Baseline (Neural ODE)     | 74.475 (1.181)        | 73.870 (0.820)       | 70.650 (0.688)      | 91.348 (0.440)      |
> | Dropout for Drift Network| 75.850 (0.367)        | 74.865 (1.710)       | 70.787 (0.197)      | 91.71 (0.353) |
> | Dropout for MLP Classifer| 75.005 (0.575)        | 75.095 (0.685)       | 70.723 (0.376)      | 92.327 (0.098)      |
> | NRSDE (Ours)              | 76.470 (0.480)        | 76.877 (0.615)       | 71.833 (0.334)      | 92.381 (0.083)      |

---

> > ### Comment · Reviewer_jDuQ · 2024-11-25
> > **Many thanks for the response**
> >
> > Many thanks for the detailed response. Regarding the individual points:
> >
> > **Monte-Carlo samples.** Thank you for this response, just to check again, do these samples have to be taken for every single inference point, including in deployment if we were to get new test points?
> >
> > **Sensitivity to number of samples.** Thank you for including this table, for me it makes the benefit of estimating $c$ more clear.
> >
> > **Additional dropout baseline.** My apologies for not understanding the dropout for drift network before.

---

> > > ### Author Response · Authors · 2024-11-26
> > >
> > > Thank you for your very constructive feedback. It has been invaluable in improving various aspects of our paper, including areas we might have overlooked.
> > >
> > > Regarding Monte Carlo samples: Yes, these samples are required for each inference point, including during deployment for new test points.

---

> > > > ### Comment · Reviewer_jDuQ · 2024-11-26
> > > > **Thanks for the response**
> > > >
> > > > Great many thanks for the response. Ultimately this could be made better in future research by finding a way to calculate this $c$ once, as is done in ordinary dropout.

---

### Official Review · Reviewer_gfhy · 2024-10-21

**Soundness:** 3
**Presentation:** 4
**Contribution:** 2
**Rating:** 6
**Confidence:** 3

**Summary:**

This paper investigates regularization techniques for NDEs, with a particular emphasis on dropout methods. The authors introduce a novel NRSDE approach, grounded in alternating renewal processes that characterize dropout dynamics through transitions between active and inactive states. The paper includes a rigorous theoretical analysis, providing guidelines for selecting dropout rates and scaling outputs during the testing phase. Extensive experiments conducted across various systems demonstrate that the proposed method outperforms existing baseline models.

**Strengths:**

1. The focus on regularization for NDEs is somewhat interesting.
2. The study is presented with a high level of completeness, as the proposed method, along with the hyperparameter settings and scaling mechanisms, is thoroughly developed.
3. The manuscript is well-structured and clearly written, facilitating easy comprehension for readers.

**Weaknesses:**

1. My primary concern is regard the motivation for developing dropout techniques for NDEs in a way just like the ``true’’ dropout mechanism for conventional neural networks. The aim/effect of dropout is to mitigate overfitting via randomly disabling a subset of neurons. In this context, the dropout strategies proposed by Liu et al. (2020) appear to be adequate. In other words, I don't know why we should maintain the dynamics during inactive states.
2. In Section 3.3, the addition of $\mathbf{c}$ for scaling requires further explanation.
3. Determining the value of $\mathbf{c}$ through Monte Carlo simulations during the testing phase could hinder the practical applicability of the proposed method. The experimental section would benefit from including an experiment that clearly demonstrates that the observed improvements are not simply attributable to test-time optimization.

**Questions:**

Please refer to weaknesses

---

> ### Author Response · Authors · 2024-11-25
>
> Thank you for your valuable comments. We have carefully studied your comments and have made every effort to address your concerns. Detailed responses to each comment are provided below.
>
> **Response to Weaknesses (W)**
>
> > **W1.**
>
> Thank you very much for the opportunity to address this important comment, which is crucial for understanding the contributions of our paper.
>
> As you mentioned, both Liu et al. and our work aim to model the mechanism of dropout—randomly pausing neurons—in a continuous learning representation for Neural Differential Equations (NDEs).  A shared goal is to develop a continuous stochastic process that aligns with the discrete dropout dynamics in neural networks, described by: (Equation (4) in Liu’s paper)
>
> $$
> \mathbf{Z}(k+1) = \mathbf{Z}(k) + \gamma(\mathbf{Z}(k); \theta_\gamma) \circ \xi_k, \quad k = 0, 1, \ldots, N-1,
> $$
> where $\xi_k^{(i)} \sim \text{Bernoulli}(1-p)$.
>
> Liu et al. claim that the continuous version of this equation is a jump diffusion process and propose the Neural Jump Diffusion Process based on this assumption. However, as we demonstrate in Section 3.4 and Appendix G.1 of our paper, this claim is mathematically incorrect. The misinterpretation arises from an improper discretization of the jump diffusion process, leading to errors in their theoretical analysis. These mathematical errors prevent their technique from correctly analyzing key elements of conventional dropout, such as dropout rate and scaling.
>
> In contrast, we accurately model the dropout dynamics as an **alternating renewal process**, which allows us to establish a precise connection between the concept of continuous dropout and existing notions like dropout rate and scaling. This ensures that our method aligns more closely with the conventional dropout mechanism and provides a robust, theoretically grounded framework for modeling dropout in both training and inference.
>
> > **W2.**
>
> Thank you for your comment, which helps improve the clarity of our paper. To provide more clarity, we have added an additional explanation of the scaling factor $\mathbf{c}$ in Section 3.3 of the updated paper as follows:
>
> "We explicitly note that the scaling factor is not a tunable or learnable parameter but rather a fixed value specific to each data point. Since  $\mathbf{c}$ cannot be obtained analytically, we employ Monte Carlo simulations, which may seem to introduce additional computational overhead."
>
>
> > **W3.**
>
> Thank you for bringing up this very important point. We realize that some readers might misunderstand the scaling factor $\mathbf{c}$ as being optimized during the test phase to improve performance. To clarify, $\mathbf{c}$ is a fixed value that represents the difference between $\mathbb{E}[\mathbf{z}_0(T)]$ and $\mathbb{E}[\mathbf{z}(T)]$. We simply estimate this value using Monte Carlo (MC) simulations because it cannot be obtained analytically. That is, $\mathbf{c}$ is neither a learnable parameter nor a tunable hyperparameter. While we currently compute $\mathbf{c}$ during the test phase for demonstration purposes, it can also be precomputed during the training phase without affecting results.
>
> To address concerns about the scaling factor further, we conducted additional experiments to analyze the impact of the number of Monte Carlo (MC) samples used to estimate $\mathbf{c}$. Specifically, we tested with $\text{MC} = [0, 1, 3, 5, 10, 20]$, where $\text{MC} = 0$ means no scaling was applied, effectively using $\mathbf{z}_0(t)$ for inference. As shown in Tables 19 and 20 of the updated paper, there was no significant performance improvement when increasing the number of samples beyond 5-10.
>
> These results confirm that 5-10 MC samples are sufficient to accurately estimate the scaling factor $\mathbf{c}$ and demonstrate that our method does not rely on test-time optimization.

---

> > ### Comment · Reviewer_gfhy · 2024-11-26
> >
> > Thank you for your comprehensive responses. I have thoroughly reviewed them and concur with the authors' insights. Most of my concerns have been addressed and I have decided to raise my score.

---

> > > ### Author Response · Authors · 2024-11-26
> > >
> > > Thank you for taking the time to thoroughly review our responses and for your thoughtful feedback.  We truly appreciate your decision to raise your score.

---

### Official Review · Reviewer_jq8a · 2024-10-30

**Soundness:** 2
**Presentation:** 2
**Contribution:** 2
**Rating:** 5
**Confidence:** 4

**Summary:**

This paper introduces an extension to neural differential equations that functions similarly to the classical Dropout layer in standard neural networks. The extension is based on a renewal process defined by two hyperparameters: (i) the expected number of renewals, or on-off states, within a given time interval, and (ii) the probability of an "on" state after the interval. The authors demonstrate the utility of their method across different classification tasks.

**Strengths:**

The use of a renewal process is a novel concept, to the best of my knowledge.

The method's applicability is shown for several baseline models.

**Weaknesses:**

The paper's primary weakness lies in its motivation and presentation. While the authors position neural differential equations as an ideal tool for modeling continuous-time systems, they limit their experiments to classification problems rather than exploring continuous dynamical systems. Addressing this issue is essential for recommending acceptance.

**Questions:**

I'm unsure on the scaling aspect with your dropout layer. Typically, scaling is applied during training, but you've chosen to apply it only during inference. Could you elaborate on this design choice?

You reference Liu et al.'s work, "How Does Noise Help Robustness? Explanation and Exploration under the Neural SDE Framework," as the closest comparable method, but I noticed it’s missing from the comparison table. Could it be added for completeness?

Could you explain in more detail the difference between the Jump Diffusion process proposed by Liu et al. in "How Does Noise Help Robustness? Explanation and Exploration under the Neural SDE Framework", Dropout as described by Look et al. in "A Deterministic Approximation to Neural SDEs", and Dropout as proposed by you?

Would it be possible to extend your algorithm to use a non-fixed-step-size solver?

Could you elaborate on your point that SDEs serve as a regularizer for NDEs? To my knowledge, SDEs are not regularizers but are used for modeling different types of dynamical systems.

Could you discuss the effect of applying dropout to weights rather than inputs?

---

> ### Author Response · Authors · 2024-11-25
>
> Thank you for your valuable comments. We have carefully studied your comments and have made every effort to address your concerns. Detailed responses to each comment are provided below.
>
> > **Response to Weakness**
>
> As you noted, NDEs are highly versatile tools, applicable not only to continuous representation learning for neural networks, which is our focus, but also to tasks such as time series modeling. Our method introduces the ability to model regenerative processes—dynamics involving recurring active and inactive states—which are not well captured by existing variants of NDEs like NSDEs, Neural temporal point processes, or Neural jump processes. This capability opens up applications in fields such as quality management, reliability analysis, and operations research, where such processes frequently arise.
>
> Despite its broader potential, we focused on demonstrating NRSDE's ability to capture continuous dropout representation learning in neural networks. This novel and inherently complex concept required prioritizing clarity and depth to effectively convey our idea.
>
> On the other hand, we agree with your opinion. To address your concern, we have added experiments related to continuous-time modeling in Section 4.4 of the updated paper. Specifically, we designed a prediction task using a dataset with periods of inactivity. Traditional Neural SDE methods often struggle with such datasets due to their instability and inability to effectively capture constant values over time. In contrast, our method, NRSDE, demonstrates significantly improved stability and performance in modeling these dynamics.
>
> **Response to Questions (Q)**
>
> > **Q1.**
>
> Thank you for your question. Conventional dropout applies scaling during the test phase [1], and our method follows the same principle. By adopting scaling during inference, we aim to accurately represent the behavior of conventional dropout in a continuous-time setting.
>
> Of course, from an implementation perspective, applying the reciprocal of the scaling factor during training instead of inference would yield the same results. Similarly, our method could also apply scaling during the training phase instead of the test phase without affecting the outcome.
>
> [1] Nitish Srivastava et al. Dropout: A Simple Way to Prevent Neural Networks from Overfitting. JMLR 2014.
>
> > **Q2.**
>
> Thank you for your comment. We would like to point out that our paper already includes experiments comparing our method with various regularization techniques, including Liu et al.'s dropout approach, as presented in Section 4.1 (see Tables 1 and 2). These results were provided to highlight the relative performance of our method in detail.
>
> If the concern is regarding the absence of Liu et al.'s method in Tables 3 and 4, we would like to clarify the distinction. Liu's method can be understood as a specific variant of NDEs incorporating regularization effects. In contrast, our method is a universally applicable dropout technique that can be integrated with any type of NDE. To emphasize this broader applicability, we focused on reporting results across diverse NDE models with our method. As such, a direct comparison with Liu’s method in those tables is not applicable.

---

> > ### Comment · Reviewer_jq8a · 2024-11-26
> > **New Experiments**
> >
> > I appreciate your responses, and agree that most of my concerns have been adressed.
> > However, I am still struggling with the motivation of the paper, as noted by the major weakness in my initial review.
> > I acknowledge the new experiment in 4.4, however it is missing details and a  proper discussion of the results.
> >
> > I will raise my score to 5, but cannot recommend acceptance after rebuttal.

---

> > > ### Author Response · Authors · 2024-11-27
> > >
> > > We are happy to hear that most of your concerns have been addressed, and we sincerely appreciate you raising your score.
> > >
> > > Regarding the motivation, we understand your perspective. While NRSDE can indeed be applied to modeling regenerative processes, as demonstrated in the new experiment in Section 4.4, we found it particularly intriguing that NRSDE could address a more urgent challenge in NDEs: the concept of continuous dropout. This connection between NDEs and dropout motivated us to focus on developing and presenting the novel concept of continuous dropout for NDEs. Additionally, we believe our experiments align with this focus by emphasizing the modeling of continuous latent processes (learning representation) in neural networks.
> > >
> > >
> > > That said, we fully respect your opinion and recognize that preferences in presentation can vary.

---

> ### Author Response · Authors · 2024-11-25
>
> > **Q3.**
>
> Thank you for your excellent question. Understanding these differences is key to appreciating our contribution.
>
> **1. Liu et al. [1]**
>
> Both Liu et al. and our work share the same objective: developing a dropout technique for Neural Differential Equations (NDEs). Specifically, this involves finding a suitable continuous stochastic process corresponding to the hidden state of a ResNet with dropout at the $k$-th layer, expressed as:
>
> $$\mathbf{Z}(k+1) = \mathbf{Z}(k) + \gamma(\mathbf{Z}(k); \theta_\gamma) \circ \xi_k, \quad k = 0, 1, \ldots, N-1,$$
>
> where $\xi_k^{(i)} \sim \text{Bernoulli}(1-p)$.
>
> Liu et al. claim that the continuous version of this equation is a jump diffusion process and propose a Neural Jump Diffusion Process based on this assumption. However, this claim is mathematically incorrect because they misinterpret the discretization of the jump diffusion process as being analogous to the above equation. (See Section 3.4. and Appendix G.1.) The mathematical errors prevent their dropout technique from correctly analyzing important elements of conventional dropout, such as dropout rate and scaling.
>
> In contrast, we accurately model the above ResNet dropout evolution as an **alternating renewal process**. This approach allows us to precisely connect the new concept of continuous dropout with existing notions such as dropout rate and scaling. Our method, therefore, aligns more closely with the conventional dropout mechanism and provides a robust, theoretically grounded framework for modeling dropout in both training and inference.
>
> **2. Look et al. [2]**
>
> Look et al. do not propose a novel dropout technique tailored for NDEs. Instead, they study the mathematical implications of applying existing dropout to the drift function. While their work provides valuable insights into the theoretical role of dropout in moment matching and uncertainty quantification, it does not offer a fundamental solution for addressing overfitting in NDEs.
>
> **Summary**
>
> In summary, while Liu et al. attempt to propose a dropout method for NDEs, their approach suffers from mathematical errors that affect both theoretical analysis and empirical performance. Look et al. provide theoretical insights but do not introduce a novel dropout technique for NDEs. Our method stands out by accurately modeling continuous dropout as an alternating renewal process, ensuring both mathematical rigor and practical alignment with the conventional dropout mechanism.
>
> [1] Liu et al. How Does Noise Help Robustness? Explanation and Exploration under the Neural SDE Framework. CVPR 2020.
>
> [2] Look et al. A Deterministic Approximation to Neural SDEs. IEEE Transactions on Pattern Analysis and Machine Intelligence 2022.
>
>
> > **Q4.**
>
> An adapted time step solver can be applied for our method. Specifically, during the active phase, the adapted time step is applied as usual, while in the inactive phase, the time step is extended to last until the end of the inactive phase. This approach enhances computational efficiency while retaining the benefits of adaptive solvers, such as the ability to handle stiff dynamics.
>
> Furthermore, it is important to highlight that our method is compatible with any solver. In response to a request from the reviewer eeNJ, we applied NRSDE to the advanced solver MALI [1], which utilizes an adapted time step. For this experiment, we implemented the aforementioned approach for integrating NRSDE with an adapted time step solver.
>
> The experimental results demonstrate that while the advanced solver MALI already improves performance, combining it with NRSDE yields even better results. This empirically validates the successful extension of NRSDE to adaptive time step solvers.
>
> | Solver          | CIFAR-100         | CIFAR-10          |
> |-----------------|-------------------|-------------------|
> | Euler           | 74.475 (0.581)   | 73.870 (0.820)   |
> | MALI            | 75.560 (0.397)   | 75.733 (0.505)   |
> | MALI + NRSDE    | **76.227 (0.423)** | **78.070 (0.317)** |
>
> [1] Zhuang, Juntang, et al. MALI: A memory efficient and reverse accurate integrator for Neural ODEs. ICLR 2021.
>
>
> > **Q5.**
>
> Neural ODEs  are designed to model deterministic processes, which can lead to overfitting when the data contains noise or follows a stochastic process. By introducing a diffusion term, Neural SDEs account for this noise, effectively acting as a regularizer by preventing overfitting to a single path or noisy observations.
>
> From a representation learning perspective, adding stochastic noise to hidden layers can be interpreted as Neural SDEs. In this sense, Neural SDEs can be seen as an extension of NODEs that incorporates a regularization effect through stochasticity.
>
> At the same time, Neural SDEs are naturally suited for modeling stochastic dynamical systems. These two interpretations—regularization and stochastic modeling—are complementary and depend on the specific application or context.

---

> ### Author Response · Authors · 2024-11-25
>
> > **Q6**
>
> Thank you for your question. It seems there may have been some misunderstanding of our paper, so we would like to clarify.
>
> Our method does not focus on applying dropout to inputs. Instead, we propose a framework where the **continuous learning representation** of a neural network with dropout is modeled as an NRSDE. This modeling approach is significant because simply applying dropout to the drift or diffusion networks of NDEs does not result in meaningful improvements in performance or robustness (See Table 1 for the results labeled "dropout for drift network"). Therefore, we developed a systematic and theoretically grounded framework for **continuous dropout techniques** specifically tailored for NDEs, which fills this gap.
>
> Given that you indicated a confidence level of 4 in your review, we assume you have a solid understanding of our paper. In this context, the question you raised is somewhat unexpected. If your question was intended to address a different aspect or explore another angle, please let us know.

---

### Official Review · Reviewer_eeNJ · 2024-10-30

**Soundness:** 3
**Presentation:** 3
**Contribution:** 2
**Rating:** 6
**Confidence:** 4

**Summary:**

This paper addresses a significant gap in the field of Neural Differential Equations (NDEs) models by focusing on regularization techniques. The authors identify a lack of effective regularization methods for NDEs and propose a novel approach to apply dropout in this context.

**Strengths:**

1. Innovative application of dropout: The authors have successfully adapted the dropout technique, which is commonly used in traditional neural networks, to Neural Differential Equations (NDEs). This adaptation shows a deep understanding of both NDEs and regularization techniques, bridging the gap between these two areas of machine learning.

2. Simplicity and versatility: The proposed method is remarkably straightforward, making it highly accessible and practical. Its simplicity does not compromise its effectiveness, and importantly, it can be easily applied to a wide range of NDE-based networks. This versatility enhances the method's potential impact across various applications and research areas within the field of continuous-time deep learning.

3. Excellent readability: The paper is exceptionally well-written, making it accessible to a broad audience. The authors have succeeded in presenting complex concepts in a clear and comprehensible manner, which is crucial for the dissemination and adoption of new ideas in the scientific community.

**Weaknesses:**

This paper lies in its narrow focus on dropout as a regularization technique for Neural Differential Equations (NDEs). While the direct application of dropout to the network is valuable, other regularization approaches exist for NDEs, such as using solvers as regularizers. The paper concentrates too heavily on dropout, potentially overlooking these alternative methods.

**Questions:**

Could this dropout method also lead to improved performance in models using advanced solvers [1]? I would like to see the test accuracy performance on CIFAR-10 by applying dropout to the solver proposed in [1].

Also, although dropout acts as a regularizer, it shows a larger performance variance compared to other baseline models in most experiments. Why is that?

[1] MALI: A memory efficient and reverse accurate integrator for Neural ODEs

---

> ### Author Response · Authors · 2024-11-25
>
> Thank you for your valuable comments. We have carefully studied your comments and have made every effort to address your concerns. Detailed responses to each comment are provided below.
>
> > **Response to Weakness**
>
> Thank you for introducing regularization methods for NDEs that leverage solvers. We appreciate your perspective and want to clarify that our work does not focus solely on dropout-based regularization. For instance, STEER, one of the methods we evaluate, is not a dropout-based regularization technique. We have aimed to cover a broad spectrum of regularization methods for NDEs.
>
> To address your point about solver-based regularization effects, we conducted additional experiments to compare our approach. The results below show the performance on CIFAR-100 and CIFAR-10:
>
> | Solver          | CIFAR-100         | CIFAR-10          |
> |-----------------|-------------------|-------------------|
> | Euler           | 74.475 (0.581)   | 73.870 (0.820)   |
> | MALI            | 75.560 (0.397)   | 75.733 (0.505)   |
> | MALI + NRSDE    | **76.227 (0.423)** | **78.070 (0.317)** |
>
> These experiments highlight that MALI, as an advanced solver, already improves performance compared to simpler solvers like Euler. Furthermore, when combined with NRSDE, performance is further improved, demonstrating the complementary nature of NRSDE to solver-based regularization techniques.
>
> **Response to Questions (Q)**
>
> > **Q1.**
>
> Please refer to Response to Weakness.
>
> > **Q2.**
>
> Based on our experimental results, the variance of our method is not larger than that of the baselines. As shown in Tables 1–5, the variance of the results when applying our method is generally comparable to or even lower than that of the baseline models.

---

> > ### Comment · Reviewer_eeNJ · 2024-11-25
> >
> > Thank you for the authors' response. They have addressed my concerns. I will increase my score.

---

> > > ### Author Response · Authors · 2024-11-26
> > >
> > > We are very pleased to hear that your concerns have been fully addressed. Thank you for increasing your score!

---

### Official Review · Reviewer_5zJ2 · 2024-10-31

**Soundness:** 2
**Presentation:** 3
**Contribution:** 2
**Rating:** 5
**Confidence:** 3

**Summary:**

This paper explores how to use the well-known dropout technique for Neural Differential Equations (NDEs) to achieve robustness and generalizability.

**Strengths:**

The proposed NRSDE introduces a specialized dropout mechanism tailored for Neural Differential Equations (NDEs), which addresses a known limitation in current NDE frameworks. This regularization technique, similar to standard dropout, randomly deactivates neurons during training to enhance robustness and generalizability. The approach is with theoretical foundation. Experimental validation on multiple datasets and model types demonstrates performance improvements.

**Weaknesses:**

1.	The motivation of dropout-based regularization for NDEs is limited in the Introduction. For example, there are extensive regularization techniques. Why do the authors choose the dropout-based scheme for NDEs? The authors should provide stronger support from literature or experiments.
2.	The authors employ a key concept called alternating renewal processes. However, in the Introduction, the concept is not explained explicitly and there is no citation about this process.
3.	The following statement is unclear. “Assume that $X_n$ and $Y_m$ are independent for any $n\neq m$, but can be dependent for $n=m$.”
4.	Why do you focus on the exponential alternating renewable process? Are there other options?
5.	The authors should state the importance of the introduced baselines. Are they cutting-edge methods? Are they representative and comprehensive?
6.	As NRSDE is computationally tensive, the authors should report the computational time in the Experiments and future possible solutions to mitigate this issue in the Conclusion.

**Questions:**

1. What are other methods? Why do the authors choose the dropout-based scheme for NDEs?
2. What are the drawback to the alternating renewal processes?
3. Why do you focus on the exponential alternating renewable process? Are there other options?
4. Are the benchmark method cutting-edge?
5. How robust is the proposed method in low-data scenarios where overfitting is a primary concern?

---

> ### Author Response · Authors · 2024-11-25
>
> Thank you for your valuable comments. We have carefully studied your comments and have made every effort to address your concerns. Detailed responses to each comment are provided below.
>
> **Response to Weaknesses (W)**
>
> > **W1.**
>
> Thank you for giving us the opportunity to clarify the motivation behind our paper.
>
> Neural Differential Equations (NDEs) offer a new paradigm for modeling the complex dynamics of data. They have been widely used in applications such as representation learning and time series modeling, leading to the development of various NDE variants. However, their strength in capturing complex nonlinear relationships between input features also makes them highly prone to overfitting. This underscores the urgent need for effective regularization techniques specifically designed for NDEs.
>
> Despite this need, regularization methods for NDEs remain relatively underexplored compared to those for conventional neural networks. Additionally, many regularization techniques developed for standard neural networks are not directly applicable to NDEs. To the best of our knowledge, only a few examples exist, such as STEER [1] and Liu's paper [2].
>
> Dropout, in particular, is one of the most efficient and widely used regularization techniques in deep learning. While [2] attempted to propose dropout for NDEs, their method contains mathematical errors, preventing their dropout technique from correctly describing the process of conventional dropout (See Section 3.4 and Appendix G.1 for detailed discussions.). Therefore, the development of a theoretically well-established dropout technique for NDEs is both critical and timely, which we aim to address in this work.
>
> [1] Ghosh, Arnab, et al. STEER: Simple Temporal Regularization for Neural ODE. NeurIPS 2020.
>
> [2] Liu, Xuanqing, et al. How Does Noise Help Robustness? Explanation and Exploration under the Neural SDE Framework. CVPR 2020.
>
> > **W2.**
>
> We have added citations for alternating renewal processes in the Introduction of the updated paper. A detailed explanation of the concept can be found in Section 2.2.
>
> > **W3.**
>
> To clarify the statement, we have revised the updated paper as follows:
>
> "Assume that $X_n$ and $Y_m$ are independent for any $n \neq m$. However, $X_n$ and $Y_n$ (for the same index $n$) are not necessarily independent."
>
> > **W4.**
>
> While other options exist for modeling the durations of active and inactive periods using distributions other than the exponential distribution, we chose to focus on the exponential alternating renewal process for two primary reasons:
>
> 1. **Mimicking the Conventional Dropout Mechanism:**
>
>     In conventional dropout, the probability of a node being dropped is independent of the results of other nodes. In a continuous-time setting, this property translates to a memoryless characteristic, where dropout events occur according to an exponential distribution. This ensures consistency between our method and the behavior of conventional dropout.
>
> 2. **Computational Convenience:**
>
>    The exponential distribution offers analytical tractability, allowing us to compute key quantities of the process—such as the average number of repetitions of active and inactive states or instantaneous availability—easily. These computations are crucial for matching the behavior of our NRSDE with the conventional dropout rate, as shown in Theorem 3.1 and Corollary 3.2. Alternative forms of alternating renewal processes would lose these advantages, complicating both analysis and implementation.
>
>
> These reasons highlight why we focused on the exponential alternating renewal process in our work.
>
> > **W5.**
>
> We have added an explanation regarding the choice of baselines in the updated paper.
>
> The primary baseline, Neural ODE, is the foundational model for Neural Differential Equations (NDEs). Additionally, we have included several variants of Neural ODEs, including state-of-the-art methods such as Neural LSDE, GSDE, and LNSDE, as detailed in Section 4.2. Furthermore, we compared our method against regularization techniques specifically developed for NDEs, including STEER, Liu’s Neural SDE, and Neural jump diffusion processes, as described in Section 4.1. These baselines ensure that our evaluation is both representative and comprehensive.
>
> > **W6.**
>
> We have included the computational time for the experiments in the revised manuscript to provide a clear understanding of the computational cost associated with NRSDE. Additionally, we performed a sensitivity analysis by varying the number of MC samples to evaluate the proposed method. The results show that using 5-10 MC samples is sufficient to achieve stable performance across both time series and image datasets. For more details, please refer to Section 4.3 of the updated manuscript.

---

> ### Author Response · Authors · 2024-11-25
>
> **Response to Questions (Q)**
>
> > **Q1.**
>
> Please refer to Response to W1.
>
> > **Q2.**
>
> One potential drawback of alternating renewal processes is that they can introduce slight complexity in path generation compared to simpler stochastic processes. However, as explained in Response to W4, the exponential alternating renewal process offers significant advantages due to its memoryless property and analytical tractability.
>
> > **Q3.**
>
> Please refer to Response to W4.
>
> > **Q4.**
>
> Please refer to Response to W5.
>
> > **Q5.**
>
> The datasets used in Table 1 of Section 4.1 (SmoothSubspace, ArticularyWordRecognition, ERing, RacketSports) are relatively small, with total sample sizes of 300, 575, 300, and 303, respectively. These scenarios are particularly prone to overfitting due to the limited amount of data.
>
> Despite these challenges, our proposed method demonstrates robust performance, improving results by 2.3% to a maximum of 7% across these datasets. These improvements highlight the effectiveness of our approach in addressing overfitting, even in low-data scenarios.

---

> ### Author Response · Authors · 2024-11-27
>
> Once again, thank you for your valuable comments and insights.
>
> We would greatly appreciate it if you could provide feedback on our responses. If you have any remaining unaddressed concerns, please let us know so that we can address them further.

---

### Official Review · Reviewer_43ig · 2024-11-01

**Soundness:** 2
**Presentation:** 3
**Contribution:** 3
**Rating:** 6
**Confidence:** 4

**Summary:**

The paper addresses the lack of regularization methods for NDEs, by introducing a novel NRSDE method. Its main innovation is to incorporate the dropout mechanism into models that work in continuous time setting using a so-called alternating renewal process, a mathematical framework for modelling systems that alternate between two states over time. NRSDE is designed to be applicable to various types of NDEs. Authors discuss the relations between their approach and prior-art work. The method is extensively tested on various datasets and compared with prior-art methods. In addition, a study on sensitivity of NRSDE’s hyper-parameters is provided.

**Strengths:**

- Regularization techniques, such as dropout, are essential for modern AI models and thus the topic that the authors touch upon is very timely.
- The authors have provided extensive proves and mathematical details about their approach. From what I can tell, the math seems sound as well
- I appreciate that the source code is shared publicly

**Weaknesses:**

- The main weakness that I see with this paper is the overhead to compute the scaling factor. In machine learning workloads one would like to keep the inference (testing) phase as simple and fast as possible, as this phase is use extensively by many users and in many applications. Therefore, the requirement to compute the scaling factor with Monte Carlo Sampling (MCS) during the testphase for each and every new test sample is really not ideal. Furthermore:
    1. tuning of the scaling factor for the test phase. Line 317-320 and Appendix F.2 show that 5-10 samples are sufficient to estimate the scaling factor. This is done for a fairly simple time series dataset (google speech command). What would happen in case of image datasets?
    2. the scaling factor is one of NRSDE’s parameters. Hence, following the best practices of ML, it has to be determined (or learned) using the only training data. But, currently, as shown in Algorithm 1, it is done using test data. This issue can be possibly addressed by dedicating a subset of training data specifically for determining the scaling factor via MCS.
    3. from Table 17 in F.1, the overhead to compute the scaling factor is by no means small or negligible.
    4. from Table 18 it is unclear whether indeed 5 or 10 is sufficient. What happens for larger values?
- The choice of the tasks is suboptimal. The authors claim to address “continuous-time dynamical systems” and the second set of tasks revolves around static image classification. If one would be interested in the spatial domain of images, then at least one needs to consider videos, or some other ways to convert images into a temporal input sequence that has a continuous-time dynamical systems underlying.
Minor comments: The authors chose the same notation (.) to denote the time t and the layer k. This is an unfortunate choice and may confuse the reader.

**Questions:**

See the points above

---

> ### Author Response · Authors · 2024-11-25
>
> Thank you for your valuable comments. We have carefully studied your comments and have made every effort to address your concerns. Detailed responses to each comment are provided below.
>
> **Response to weaknesses (W)**
> > **W1. Estimation of scaling factor**
>
> Thank you very much for your insightful comment. We address your concerns regarding the overhead of computing the scaling factor and its implementation during the test phase as follows:
>
> 1. **Scaling Factor Estimation on Image Datasets**
>
> We have extended our experiments to include the CIFAR-100 dataset, an image dataset, to evaluate the performance of NRSDE with varying numbers of Monte Carlo (MC) samples. The results, which are available in Table 20 of Appendix F.2 in the updated paper, show that 5-10 samples remain sufficient to estimate the scaling factor $\mathbf{c}$ even in image-based datasets, aligning with the findings from the Speech Commands dataset.
>
> 2. **Mechanism of the scaling factor**
>
> The scaling factor $\mathbf{c}$ is neither a learnable parameter nor a tunable hyperparameter. It is a fixed value that reflects the difference between $\mathbb{E}[\mathbf{z}_0(T)]$ and $\mathbb{E}[\mathbf{z}(T)]$, and its computation relies on MC simulations. We emphasize that no tuning of the scaling factor is performed to enhance results. Furthermore, while we currently compute the scaling factor during the test phase for demonstration purposes, it is entirely feasible to design the method such that $\mathbf{c}$ is precomputed during the training phase.
>
> 3. **Computational Overhead**
>
> While the method introduces additional computational cost, we believe this overhead remains within a reasonable range given the robustness and significant benefits it provides. For instance, using 5 MC samples adds 6.1% to 13.6% additional computational cost. This modest increase is outweighed by the substantial performance improvements achieved over state-of-the-art methods, making the trade-off not only practical but also advantageous for many real-world applications where robust and reliable models are crucial for deployment.
>
> 4. **Impact of MC Sample Size**
>
> To further analyze the effect of the number of MC samples, we conducted additional experiments with $\text{MC} = [0, 1, 3, 5, 10, 20]$. Here, $\text{MC} = 0$ means no scaling is applied during the test phase, corresponding to inference using $\mathbf{z}_0(t)$. The results, shown in Tables 19 and 20 of Appendix F.2, reveal no significant performance gains when using 20 samples compared to 5-10 samples across different models and datasets. This confirms our conclusion that 5-10 samples are sufficient for estimating the scaling factor $\mathbf{c}$.
>
> > **W2. Additional Experiment for Continuous-Time Modeling**
>
> The "continuous-time dynamical systems" in the context of static image classification refers to modeling the learning representation in deep learning using Neural Differential Equations (NDEs) in a continuous manner, as opposed to discrete counterparts like ResNet. This follows the approaches proposed in previous NDE studies [1, 2, 3].
>
> However, as reviewer jq8a also suggested exploring tasks beyond classification to model continuous dynamical systems, we conducted an experiment using the Total Agent Call Time dataset (See Section 4.4 of the updated manuscript). This dataset represents the cumulative call time of multiple call center agents over time and is characterized by periods of inactivity (no calls). These time series data highlight the importance of accurately modeling idle (inactive) periods, such as breaks or times without calls, which recur in a regenerative manner. NRSDE proves highly suitable for modeling such time series, whereas existing variants of NDEs fail to adequately capture the characteristics of regenerative processes, underscoring their limitations for such tasks.
>
> [1] Ricky T. Q. et al. Neural Ordinary Differential Equations. NeurIPS 2018.
>
> [2] Xuanqing Liu et al. How Does Noise Help Robustness? Explanation and Exploration under the Neural SDE Framework. CVPR 2020.
>
> [3] Arnab Ghosh et al. STEER: Simple Temporal Regularization For Neural ODEs. NeurIPS 2020.
>
> > **W3. Notation**
>
> Thank you for highlighting this potential source of confusion. While it is common to use a slight abuse of notation, where the subscript (.) is used to denote discrete-time solutions, we opted against this approach. In our work, the subscript is already reserved for representing the original process $\mathbf{z}_0$. Using the same notation for the discrete layer index $k$ or time $t$ would risk further ambiguity. To address this, we adopted the current notation, $\mathbf{Z}_0(k)$, for the $k$-th layer of ResNet without dropout.

---

> > ### Comment · Reviewer_43ig · 2024-11-26
> > **Overhead of computing the scaling factor c**
> >
> > Thank you for the effort of providing answers to my concerns. My main concerns is still present, which is the calculation of the scaling factor c. It is debatable whether a 13.6% increase in runtime is negligible or not. However, most pressing is the comment that the authors made:
> >
> > > Furthermore, while we currently compute the scaling factor during the test phase for demonstration purposes, it is entirely feasible to design the method such that is precomputed during the training phase.
> >
> > Can you point me to an experiment where this is demonstrated? I guess that the scaling factor highly depends on the data and thus if the training data has different characteristics than the testing data, this way of computing the scaling factor may not work?

---

> > > ### Author Response · Authors · 2024-11-27
> > >
> > > Thank you for your detailed feedback and for carefully reviewing our paper. We appreciate your input and the opportunity to clarify our approach regarding the scaling factor $\mathbf{c}$.
> > >
> > > First, as you correctly mentioned, $\mathbf{c}$ should be calculated during the test phase, as its value depends on the test sample.  This makes it challenging to design a method where $\mathbf{c}$ is precomputed during the training phase. We appreciate this clarification. However, **we emphasize that our main messages regarding the scaling factor remain unchanged as follows:**
> > >
> > > - $\mathbf{c}$ is not a tunable or learnable hyperparameter.
> > > - A sample size of 5–10 is sufficient to achieve stable performance.
> > > - The additional computational cost incurred by using Monte Carlo simulations is 6.1%–13.6%.
> > >
> > > Second, regarding your point, "It is debatable whether a 13.6% increase in runtime is negligible or not," we totally understand your concern. To further support this discussion, we conducted additional experiments using the Speech Commands dataset, comparing NDE models with a transformer-based model.
> > >
> > > NDE-based models are known to be highly efficient for irregular time series tasks, consistently outperforming other deep learning models such as transformers in terms of computational efficiency and performance. As shown in Table 1, even with a modest computational cost increase (~10%), NDE models with our dropout technique, except for ANCDE, remain slightly more efficient than transformers in terms of computation time. Additionally, Table 2 highlights that NRSDE achieves superior accuracy compared to both transformer and baseline NDE models, further justifying the trade-off.
> > >
> > > Table 1: Computation time comparison on Speech Commands (time in seconds per epoch)
> > > | Dropout | MC   | Neural CDE     | ANCDE         | Neural LSDE     | Neural LNSDE    | Neural GSDE    | Transformer |
> > > |---------|------|----------------|---------------|-----------------|-----------------|----------------|----------------|
> > > | X       | -    | 25.560 (0.259) | 53.264 (0.157)| 19.416 (0.147)  | 19.532 (0.109)  | 19.776 (0.052) | 33.953 (0.423)
> > > | O       | 5    | 27.127 (0.293) | 58.943 (0.300)| 21.595 (0.163)  | 21.682 (0.169)  | 22.458 (0.154) | - |
> > > |         | 10   | 32.172 (0.329) | 75.486 (0.250)| 31.567 (0.147)  | 30.034 (0.096)  | 31.234 (0.227) | -|
> > >
> > > Table 2: Accuracy on Speech Commands with different numbers of Monte Carlo simulation samples
> > > | Dropout | Neural CDE     | ANCDE         | Neural LSDE     | Neural LNSDE    | Neural GSDE    | Transformer |
> > > |---------|----------------|---------------|-----------------|-----------------|----------------|----------------|
> > > | X       |  0.910 (0.005)  | 0.760 (0.003) | 0.927 (0.004)   | 0.923 (0.001)   | 0.913 (0.001)  | 0.878 (0.002)|
> > > | O       |  0.945 (0.001)  | 0.794 (0.003) | 0.933 (0.001)   | 0.932 (0.001)   | 0.930 (0.001)  |-|
> > >
> > > We hope these comparisons demonstrate that while NRSDE incurs a modest increase in computational cost, it remains efficient compared to transformers and provides significant performance improvements. Thank you again for your constructive feedback, which has allowed us to address these concerns more comprehensively.

---

> > > > ### Comment · Reviewer_43ig · 2024-11-27
> > > > **Confusing rebuttal**
> > > >
> > > > Thank you for the response to my comment. I understand your argumentation, still my main point remains. Actually, after the first response from the authors I was tentative to improve my score, because of this statement:
> > > >
> > > > > Furthermore, while we currently compute the scaling factor during the test phase for demonstration purposes, it is entirely feasible to design the method such that is precomputed during the training phase.
> > > >
> > > > However, after the second response the authors themselves withdrew from this statement by saying:
> > > >
> > > > > First, as you correctly mentioned, $\mathbf{c}$ should be calculated during the test phase, as its value depends on the test sample. This makes it challenging to design a method where $\mathbf{c}$ is precomputed during the training phase. We appreciate this clarification.
> > > >
> > > > This is really confusing to me, as it now seems that the precomputation of the scaling factor $\mathbf{c}$ is not possible and the authors contradict themselves. Therefore, at best I would like to keep my score.

---

> > > > > ### Author Response · Authors · 2024-11-27
> > > > >
> > > > > Thank you for your thoughtful comment.
> > > > >
> > > > > When we initially responded, we believed that it would be straightforward to move the computation of the scaling factor $\mathbf{c}$ to the training phase. However, as correctly mentioned in your first feedback, the scaling factor depends on the test data and should, therefore, be calculated during the test phase. This leds us to realize that designing a method to precompute $\mathbf{c}$ during the training phase is indeed a challenging task.
> > > > >
> > > > > In this regard, we totally agree with Reviewer jDuQ’s insightful observation that finding a way to calculate $\mathbf{c}$ in a manner similar to ordinary dropout represents an important future direction.
> > > > >
> > > > > **Summary:**
> > > > >
> > > > > - It is difficult to precompute $\mathbf{c}$ during the training phase to completely eliminate the computational cost of Monte Carlo simulations.
> > > > > - The scaling factor $\mathbf{c}$ should be computed during the test phase. Therefore, our primary argument is to empirically demonstrate that the additional cost incurred by Monte Carlo simulations is within an acceptable range.
> > > > >
> > > > >
> > > > > We hope this summary clarifies any remaining confusion.

---

### Meta-Review · Area_Chair_kt3U · 2024-12-19

**Metareview:**

This paper presents a dropout method for neural differential equations, which is specialized in modeling regenerative stochastic processes, e.g., queuing models. Different to the dropout in the discrete-time neural network, the authors first define an differential equation based on i.i.d. exponential processes to present their dropout concept. In the experiment section, they introduce several interesting results.

However, the authors fail to answer some of key questions (raised by the reviewers and additionally by me). For the regenerative stochastic process, we typically analyze the active period only. After removing the inactive period, we can obtain a set of time series, which consists of samples during the active period only. In many engineering fields, we are typically interested in those active period samples. In addition, there is a paper by Liu et al. for the dropout method. I think the authors still need to report the comparison with it in a very special condition where both methods are applicable.

**Additional Comments On Reviewer Discussion:**

The authors left rebuttal messages and the reviewers actively participated in the discussion. However, the reviewers are not satisfied with the rebuttal.

---

### Decision · Program_Chairs · 2025-01-22

Reject